# TIME IS ALL IT TAKES: SPIKE-RETIMING ATTACKS ON EVENT-DRIVEN SPIKING NEURAL NETWORKS

**Yi Yu[1], Qixin Zhang[2]\*, Shuhan Ye[1], Xun Lin[3], Qianshan Wei[4], Kun Wang[5],**
**Wenhan Yang[6], Dacheng Tao[2], Xudong Jiang[1]**
[1]ROSE Lab, School of Electrical and Electronic Engineering, Nanyang Technological University
[2]Generative AI Lab, College of Computing and Data Science, Nanyang Technological University
[3]CUHK  [4]Southeast University  [5]Nanyang Technological University  [6]Pengcheng Laboratory
{yu.yi,qixin.zhang,exdjiang}@ntu.edu.sg

## ABSTRACT

Spiking neural networks (SNNs) compute with discrete spikes and exploit temporal structure, yet most adversarial attacks change intensities or event counts instead of timing. We study a timing-only adversary that retimes existing spikes while preserving spike counts and amplitudes in event-driven SNNs, thus remaining rate-preserving. We formalize a capacity-1 spike-retiming threat model with a unified trio of budgets: per-spike jitter $\mathcal{B}_\infty$, total delay $\mathcal{B}_1$, and tamper count $\mathcal{B}_0$. Feasible adversarial examples must satisfy timeline consistency and non-overlap, which makes the search space discrete and constrained. To optimize such retimings at scale, we use projected-in-the-loop (PIL) optimization: shift-probability logits yield a differentiable soft retiming for backpropagation, and a strict projection in the forward pass produces a feasible discrete schedule that satisfies capacity-1, non-overlap, and the chosen budget at every step. The objective maximizes task loss on the projected input and adds a capacity regularizer together with budget-aware penalties, which stabilizes gradients and aligns optimization with evaluation. Across event-driven benchmarks (CIFAR10-DVS, DVS-Gesture, N-MNIST) and diverse SNN architectures, we evaluate under binary and integer event grids and a range of retiming budgets, and also test models trained with timing-aware adversarial training designed to counter timing-only attacks. For example, on DVS-Gesture the attack attains high success (over $90\%$) while touching fewer than $2\%$ of spikes under $\mathcal{B}_0$. Taken together, our results show that spike retiming is a practical and stealthy attack surface that current defenses struggle to counter, providing a clear reference for temporal robustness in event-driven SNNs. Code is available at `https://github.com/yuyi-sd/Spike-Retiming-Attacks`.

## 1 INTRODUCTION

Spiking Neural Networks (SNNs) compute with discrete spikes and temporal coding, in contrast to the continuous activations in ANNs, thereby enabling computation only when information arrives and, on dedicated neuromorphic processors, yielding low energy and short latency (Roy et al., 2019; Davies et al., 2018). Recent learning advances make deep SNNs trainable through spatio temporal backpropagation with surrogate gradients (Wu et al., 2018), and refined estimators further improve gradient quality and stability for spiking nonlinearities (Lian et al., 2023). Normalization and time aware objectives support few step inference under tight latency budgets while preserving accuracy (Kim & Panda, 2021; Duan et al., 2022; Rathi & Roy, 2023). SNNs also align with event driven sensing, where asynchronous streams naturally match the sparse and temporal nature of spiking computation and favor low power edge deployment (Lichtsteiner et al., 2008). These trends position SNNs as practical backbones for efficient real time perception and motivate a closer look at robustness in the time domain where spike timing carries much information (Neftci et al., 2019).

---

\*Corresponding author.

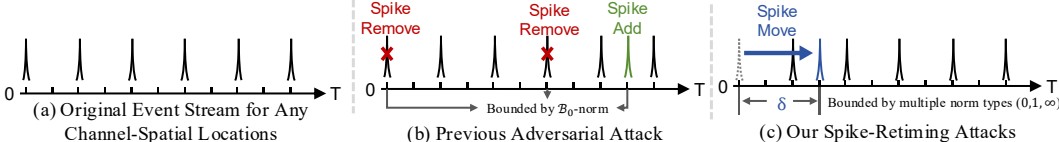

Figure 1: Attack overview. (a) Original event stream. (b) Previous attacks add/remove spikes under a 0-norm, limited to binary grids. (c) Ours move spikes on each event timeline, preserving counts and amplitudes, supports multiple norm types, and can be applied to both binary and integer grids.

Prior attacks on SNNs mostly inherit image domain strategies that modify intensities or event counts. On static image data or integer event grids, PGD (Madry et al., 2018) are applied with compatible gradients (Wu et al., 2018). Rate and timing-aware variants further sharpen gradients and raise success (*e.g.,* RGA (Bu et al., 2023) and HART (Hao et al., 2024) and their follow ups (Lun et al., 2025)). On event data, grid based methods operate on dense tensors using heuristic search, compatible gradients, or sparse rounding (Büchel et al., 2022; Marchisio et al., 2021a; Liang et al., 2023; Lun et al., 2025). Defenses include certified robustness and empirical strategies such as adversarial training with regularizers and robust objectives (Mukhoty et al., 2024; Ding et al., 2024a;b).

Despite recent progress, most attacks focus on *surrogate gradients* and mainly change *values or counts* (Lun et al., 2025), which can misalign with the *binary interface* in spike sparse regimes and tend to alter energy or rate statistics. Defenses in turn mostly regulate intensities, rates, or membrane dynamics rather than *input timing* (Ding et al., 2024a). Event cameras and other neuromorphic sensors naturally exhibit timestamp noise (jitter) and readout latency, and deployed SNN pipelines usually quantize events into discrete time bins. Under these conditions, a timing-only adversary that **retimes existing spikes while preserving spike counts and amplitudes** is realistic: it stays within the range of sensor timing uncertainty and does not change any frame-wise intensity or rate statistics. As a result, such perturbations are difficult to detect with standard intensity- or rate-based checks and directly stress the temporal computation that SNNs rely on.

In this paper, as shown in Fig. 1, we introduce **Spike-Retiming Attacks**, a timing-only adversary that retimes existing spikes along the time axis while preserving amplitudes and counts. We aim for a unified formulation that applies across event encodings, and budget types. We cast retiming as an assignment over spike timestamps with explicit feasibility: spikes remain on the timeline, each location-polarity line obeys capacity-1, and non-overlap holds within time bins. The framework supports three budgets, $\mathcal{B}_\infty$ for per-spike local jitter, $\mathcal{B}_1$ for total timing shift, and $\mathcal{B}_0$ for the number of tampered spikes. To optimize this discrete space at scale, we use projected-in-the-loop (PIL) optimization: shift-probability logits generate a differentiable soft retiming for backpropagation, and a strict projection in the forward pass enforces feasibility and the chosen budget at every step. We optimize the logits with a budget-aware objective that maximizes task loss on the projected input and adds capacity and budget penalties, which stabilizes gradients and keeps updates comparable across budget radii. The same formulation applies to binary and integer event grids without modification.

**Contributions:** 1) We formalize a timing-only threat with feasibility under budgets $\mathcal{B}_\infty/\mathcal{B}_1/\mathcal{B}_0$, establishing a unified protocol for temporal robustness. 2) Building on this, we develop projected-in-the-loop optimization, coupling a differentiable soft retiming with strict projection, yielding a scalable attack that enforces feasibility at each forward pass. 3) We further design a budget-aware objective that maximizes task loss and adds a capacity regularizer and budget penalties, yielding stable updates and aligning optimization with evaluation. 4) Finally, we conduct a comprehensive study across datasets, encodings (binary and integer), and budget regimes, and we evaluate adversarially trained models, revealing a temporal weakness in event-driven SNNs and two consistent patterns: integer grids are more robust, and polarity-specific shifts place positives later and negatives earlier.

## 2   RELATED WORK AND MOTIVATION

**Spiking Neural Networks (SNNs)** compute with discrete spikes and exploit temporal structure, offering favorable energy and latency on neuromorphic hardware (Merolla et al., 2014). Two routes dominate for high-performance SNNs. *ANN→SNN conversion* calibrates activations and sets the simulation length, and recent pipelines cut conversion error and required time steps while preserving accuracy (Rueckauer et al., 2017; Deng & Gu, 2021; Bu et al., 2022). *Direct training* learns SNNs

end to end with surrogate gradients. A key milestone is spatio-temporal backpropagation (STBP), which formalizes error propagation across layers and time and makes deep SNNs trainable from scratch (Wu et al., 2018). Later work refines surrogate gradient shape and smoothing to better match spike generation (Li et al., 2021; Wang et al., 2023), and time-aware normalization, such as tdBN (Zheng et al., 2021), supports deeper, few step models. Objectives that reward temporal efficiency, notably Temporal Efficient Training (TET) (Deng et al., 2022), improve generalization when the time step is small. Together, these pieces bring directly trained SNNs close to ANN level accuracy at low time steps while preserving the temporal computation that motivates spiking models.

**Adversarial Attacks.** SNNs are vulnerable to adversarial examples inherited from DNNs (Goodfellow et al., 2015; Miao et al., 2022; 2024a;b; Cheng et al., 2024; Cai et al., 2025), yet spikes and timing create distinct attack surfaces. For directly trained SNNs, spike-aligned white-box attacks are effective: RGA (Bu et al., 2023) uses firing-rate cues, and HART (Hao et al., 2024) fuses rate and temporal information to strengthen STBP gradients. For event-based inputs, DVS-Attacks (Marchisio et al., 2021a) search the event stream to fool SNN pipelines, SCG (Liang et al., 2023) makes continuous gradients spike-compatible to resolve gradient–input mismatch, SpikeFool (Büchel et al., 2022) adapts SparseFool by rounding sparse floating-point perturbations to binary values, and GSAttack (Yao et al., 2024) perturbs raw events via a Gumbel–Softmax parameterization. Building on this line, Lun et al. (2025) propose a Potential-Dependent Surrogate Gradient (PDSG) tied to run-time membrane distributions, and a Sparse Dynamic Attack (SDA) for binary dynamic frames.

**Existing Defenses** span certified and SNN-specific approaches. Certified robustness adapts interval bound propagation and randomized smoothing to spiking models (Mukhoty et al., 2024). Adversarial training is strengthened with weight or gradient regularization tailored to spikes (Ding et al., 2022; Liang et al., 2022). Biologically inspired mechanisms modify spiking dynamics, including stochastic gating and lateral inhibition, and DVS noise filtering provides input denoising (Ding et al., 2024b; Marchisio et al., 2021b). Inherent factors such as leakage, coding schemes, and firing thresholds also influence robustness (Sharmin et al., 2019; Zhang et al., 2023). Two recent defenses for directly trained SNNs are: Liu et al. (2024d) regularize input gradient sparsity, and Ding et al. (2024a) train for robust stability by minimizing membrane-potential perturbations in LIF dynamics.

**Motivation and Positioning of Our Attack.** SNNs process sparse event streams where information is carried by *when* spikes occur rather than by continuous intensities. Prior attacks mainly alter intensities or add/delete events, and many defenses certify or train against intensity or count changes rather than timing. We address this gap with a **timing-only** attack that retimes existing spikes while preserving amplitudes and counts, producing inputs that are physically realizable, energy-consistent, and aligned with realistic sensor jitter and latency. Focusing on timing can evade intensity-based checks and directly targets temporal computation, and it turns the search from addition to assignment on the time axis. We cast retiming into a **unified, norm-agnostic** formulation on spike timestamps that supports $\mathcal{B}_\infty$, $\mathcal{B}_1$, and $\mathcal{B}_0$ budgets, providing a protocol for evaluating timing robustness.

## 3 PRELIMINARY

### 3.1 NEURON DYNAMICS IN SPIKING NEURAL NETWORKS

**Discrete-time LIF neurons.** Following Yao et al. (2024); Lun et al. (2025), we adopt the standard leaky integrate-and-fire (LIF) neuron model (Izhikevich, 2004). Let $\boldsymbol{u}_i^{(l)}[t]$ be the membrane potential and $\boldsymbol{s}_i^{(l)}[t] \in \{0,1\}$ the spike of neuron $i$ at layer $l$ and time $t$. A common recurrence is

$$\boldsymbol{u}_i^{(l)}[t] = \tau\, \boldsymbol{u}_i^{(l)}[t-1]\big(1 - \boldsymbol{s}_i^{(l)}[t-1]\big) + \big(\boldsymbol{W}^{(l)}\boldsymbol{s}^{(l-1)}[t]\big)_i + \boldsymbol{b}_i^{(l)}, \quad \boldsymbol{s}_i^{(l)}[t] = H\big(\boldsymbol{u}_i^{(l)}[t] - V_{\text{th}}\big), \quad (1)$$

where $\tau$ is the leak, $\boldsymbol{W}^{(l)}$ and $\boldsymbol{b}^{(l)}$ are weights and bias, $V_{\text{th}}$ is the firing threshold, and $H(\cdot)$ is the Heaviside step. A hard reset sets $\boldsymbol{u}$ to zero on a spike. The LIF captures a neuron's spatiotemporal dynamics, and firing together with reset provides the nonlinearity to solve complex tasks.

**Data form & encoding.** SNNs take static images or event streams as inputs. A static input is an intensity frame $\boldsymbol{x}_{\text{s}} \in [0,1]^{C \times H \times W}$, unfolded over $T$ steps either by *direct* encoding ($\boldsymbol{x}[t] = \boldsymbol{x}_{\text{s}}$) or *rate* encoding (each pixel emits Bernoulli spikes so that $\mathbb{E}[\boldsymbol{x}[t]] = \boldsymbol{x}_{\text{s}}$ and $\boldsymbol{x} \in \{0,1\}^{T \times C \times H \times W}$). Event data are binned over fixed windows into dense grids, typically as *integer* grids that accumulate per-bin activity/energy ($\boldsymbol{x} \in \mathbb{Z}_{\geq 0}^{T \times C \times H \times W}$) or as *binary* grids that record presence ($\boldsymbol{x} \in \{0,1\}^{T \times C \times H \times W}$).

## 3.2 ADVERSARIAL ATTACKS FOR SNNS

**General formulation.** Given a classifier $f$ and label $y$, an adversarial example is $\boldsymbol{x}^{\mathrm{adv}} = \boldsymbol{x} + \boldsymbol{\delta}$ that maximizes the task loss $\mathcal{L}$ (*e.g.,* cross-entropy loss) under a $\mathcal{B}_p$-norm budget $\varepsilon$ (Madry et al., 2018; Yu et al., 2025; Xia et al., 2024a;b; Liu et al., 2024a;b;c; Li et al., 2024; 2025; Zhou et al., 2023; 2024a;b; 2025a;b):

$$\max_{\boldsymbol{\delta}} \; \mathcal{L}\big(f(\boldsymbol{x} + \boldsymbol{\delta}), y\big) \quad \text{s.t.} \quad \|\boldsymbol{\delta}\|_p \le \varepsilon. \tag{2}$$

For *frame-based* inputs (static images) or integer event grids, the common choice is $\mathcal{B}_\infty$ neighbor ($p = \infty$). For *binary* grids, sparsity is better captured by an $\mathcal{B}_0$ constraint.

**Frame-based attacks on SNNs.** Under an $\mathcal{B}_\infty$ budget, *FGSM* applies one signed step to the input:

$$\boldsymbol{x}^{\mathrm{adv}} = \mathrm{clip}_{[0,1]}\big(\boldsymbol{x} + \varepsilon \cdot \mathrm{sign}(\nabla_{\boldsymbol{x}} \mathcal{L}(f(\boldsymbol{x}), y))\big). \tag{3}$$

*PGD* repeats signed steps with projection onto the $\mathcal{B}_\infty$ ball around the clean input:

$$\boldsymbol{x}^{(k+1)} = \Pi_{\mathcal{B}_\infty(\boldsymbol{x}, \varepsilon)}\Big(\boldsymbol{x}^{(k)} + \alpha \cdot \mathrm{sign}\big(\nabla_{\boldsymbol{x}} \mathcal{L}(f(\boldsymbol{x}^{(k)}), y)\big)\Big), \quad \boldsymbol{x}^{(0)} = \boldsymbol{x} + \mathcal{U}(-\varepsilon, \varepsilon), \tag{4}$$

where $k$ indexes iterations, $\alpha$ is the step size, and $\Pi_{\mathcal{B}_\infty(\boldsymbol{x}, \varepsilon)}$ projects back to the feasible region. For SNNs, $\nabla_{\boldsymbol{x}}$ is obtained through the spiking dynamics such as the *PDSG* estimator (Lun et al., 2025).

**Event-data attacks (grid).** We focus on dense event grids: *integer* and *binary*. Integer grids essentially reuse the frame-based attacks under an $\mathcal{B}_\infty$ radius. For binary grids, the perturbation is an $\mathcal{B}_0$ flip budget where methods typically *score* candidate bins (via gradients/saliency), *select* up to the budget using straight-through or probabilistic discrete relaxations, and *project* with a keep/flip step. There is no single unified paradigm, but "score → select → project" is the common pattern.

## 4 METHODOLOGY

We formalize a timing-only threat model (Sec. 4.1), develop a differentiable retiming surrogate with a strict projection (Sec. 4.2), and couple them via projected-in-the-loop optimization (Sec. 4.3).

### 4.1 PROBLEM SETUP

**Threat model.** We formalize a **timing-only** threat model. The adversary *retimes existing spikes* while preserving amplitudes and spike count. Feasibility requires staying on the timeline and no overlap per event line and time bin. Budgets are $\mathcal{B}_p$ *constraints* on the retimings. Unless otherwise noted, following Yao et al. (2024); Lun et al. (2025), we assume a white-box attacker with access to model parameters and consider untargeted attacks.

**Definition 1** (Spike Retiming Attack). *A timing-only adversary produces an adversarial event stream by moving existing spikes along the time axis while preserving amplitudes; no spike is created, deleted, or split, and at most one spike may occupy any event-line/time-bin (capacity-1). Let $\boldsymbol{x} \in \mathbb{Z}_{\ge 0}^{T \times C \times H \times W}$ be the input event tensor with $y \in \mathcal{Y}$ as the ground-truth label. Flatten spatial–channel to $j \in \{1, \ldots, N\}$ with $N{=}CHW$, and define the active index set $\mathcal{A}(\boldsymbol{x}) = \{(s, j) : \boldsymbol{x}[s, j] > 0\}$. For each $(s, j) \in \mathcal{A}(\boldsymbol{x})$, choose an integer shift $\boldsymbol{\delta}_{s,j}$ and set $t = s + \boldsymbol{\delta}_{s,j}$ with $0 \le t < T$. The feasible assignments $\mathcal{F}(\boldsymbol{x})$ and the capacity-preserving placement $P$ are*

$$\mathcal{F}(\boldsymbol{x}) = \Big\{ \boldsymbol{\delta} : \underbrace{0 \le s + \boldsymbol{\delta}_{s,j} < T}_{\text{stay on timeline}} \; \wedge \; \underbrace{\forall j, t : \big| \{ s : (s, j) \in \mathcal{A}(\boldsymbol{x}), \; s + \boldsymbol{\delta}_{s,j} = t \} \big| \le 1}_{\text{non-overlap (capacity-1)}} \Big\},$$

$$P(\boldsymbol{x}; \boldsymbol{\delta})[t, j] = \begin{cases} \boldsymbol{x}[s, j], & \exists s \text{ s.t. } (s, j) \in \mathcal{A}(\boldsymbol{x}), \; t = s + \boldsymbol{\delta}_{s,j}, \\ 0, & \text{otherwise.} \end{cases} \tag{5}$$

*In words, $\boldsymbol{\delta}$ chooses an integer shift for each existing spike; $P(\boldsymbol{x}; \boldsymbol{\delta})$ replays the same spikes at new times. Given an SNN classifier $f : \mathbb{Z}_{\ge 0}^{T \times C \times H \times W} \to \mathbb{P}(\mathcal{Y})$ with logit vector $f(\boldsymbol{x})$ and loss $\mathcal{L}\big(f(\boldsymbol{x}), y\big)$ (e.g., cross-entropy loss), the timing-only attack solves*

$$\max_{\boldsymbol{\delta} \in \mathcal{F}(\boldsymbol{x}) \cap \mathcal{B}_p} \mathcal{L}\big(f(P(\boldsymbol{x}; \boldsymbol{\delta})), y\big). \tag{6}$$

*where $\mathcal{B}_p(\varepsilon) := \big\{ \boldsymbol{\delta} : \|\boldsymbol{\delta}\|_{p, \mathcal{A}(\boldsymbol{x})} = \big( \sum_{(s,j) \in \mathcal{A}(\boldsymbol{x})} |\boldsymbol{\delta}_{s,j}|^p \big)^{1/p} \le \varepsilon \big\}$ is a p-norm budget over $\mathcal{A}(\boldsymbol{x})$.*

**Packetization for integer grids.** We represent binned events as $x[t, j]$, where $t$ indexes discrete time bins and $j$ indexes an event line (a fixed pixel–polarity location); $x[t, j]$ is the spike count on line $j$ at time bin $t$. For integer event grids, we conceptually decompose each count into unit packets so that *capacity-1* acts on packets per event line and time bin. Detailed discussion of global optimality guarantees for Eq. 6 and the computational complexity of our solver is provided in Appendix G.

**Budgets.** Budgets connect hardware plausibility, attacker power, and fair reporting. With $\mathcal{B}_p(\varepsilon)$ in Eq. 6, the formulation is norm-agnostic, and we focus on three canonical cases:

$$\mathcal{B}_\infty(\varepsilon) = \underbrace{\left\{ \boldsymbol{\delta} : |\boldsymbol{\delta}_{s,j}| \leq \varepsilon \right\}}_{\text{local jitter}}, \ \mathcal{B}_1(\varepsilon) = \underbrace{\left\{ \boldsymbol{\delta} : \sum_{(s,j)\in\mathcal{A}(\boldsymbol{x})} |\boldsymbol{\delta}_{s,j}| \leq \varepsilon \right\}}_{\text{total timing shift}}, \ \mathcal{B}_0(\varepsilon) = \underbrace{\left\{ \boldsymbol{\delta} : \sum_{(s,j)\in\mathcal{A}(\boldsymbol{x})} \mathbb{I}\{\boldsymbol{\delta}_{s,j} \neq 0\} \leq \varepsilon \right\}}_{\text{tamper count}}. \tag{7}$$

$\mathcal{B}_\infty$ caps per-spike jitter, aligning with timestamp uncertainty and favoring local retimings; $\mathcal{B}_1$ limits aggregate timing shift, providing a single global knob that scales with event density; $\mathcal{B}_0$ bounds how many spikes are touched, capturing stealthy, minimal-footprint attacks.

## 4.2 RELAXATION AND OPTIMIZATION VIA SHIFT PROBABILITIES

The definition above specifies admissible retimings as integer assignments under *capacity-1* and a budget on $\boldsymbol{\delta}$. To optimize within this **discrete space** at scale, we build a differentiable surrogate that keeps the forward sample feasible, provides useful gradients, and works for any choice of $p$ in Eq. 6.

**Shift logits as distributions over admissible targets.** For each active source $(s, j) \in \mathcal{A}(\boldsymbol{x})$, we introduce shift logits $\phi[s, j, u]$ on an index $u \in \mathcal{U}_p$ and define tempered probabilities

$$\boldsymbol{\pi}[s, j, u] = \frac{\exp\big(\phi[s, j, u]/\kappa\big)}{\sum\limits_{v\in\mathcal{U}_p} \exp\big(\phi[s, j, v]/\kappa\big)}, \qquad \mathcal{U}_p = \begin{cases} \{-\varepsilon, \ldots, \varepsilon\}, & p = \infty, \\ \{0, \ldots, T-1\}, & \text{otherwise,} \end{cases} \tag{8}$$

where $\varepsilon$ is the radius of $\mathcal{B}_\infty(\varepsilon)$. We do not enforce the timeline constraint $0 \leq s + u < T$ at this stage for $p = \infty$; boundary handling is deferred to the soft operator and the strict projection.

**Soft shift operator.** Given the distribution $\boldsymbol{\pi}$ over admissible targets, we map each index $u \in \mathcal{U}_p$ to a target time by $T_p(s, u) = s + u$ for $p = \infty$ and $T_p(s, u) = u$ otherwise. Let $\boldsymbol{x}[t, j] = 0$ for $t \notin \{0, \ldots, T-1\}$. The expected (soft) retiming on event line $j$ is

$$S_{\boldsymbol{\pi}}(\boldsymbol{x})[t, j] = \sum_{s=0}^{T-1} \sum_{u\in\mathcal{U}_p} \boldsymbol{\pi}[s, j, u] \, \boldsymbol{x}[s, j] \, \mathbb{I}\{T_p(s, u) = t\}. \tag{9}$$

Here $x[s, j]$ denotes the packet at original time $s$ on event line $j$, and $\pi[s, j, u]$ in Eq. 8 is the probability (from the shift logits) of moving this packet by an integer offset $u$ so that it lands at $t = s + u$. Then $S_\pi(x)[t, j]$ in Eq. 9 is the expected post-attack packet at $(t, j)$, obtained by summing the contributions $x[s, j]$ from all sources and shifts with $t = s + u$ weighted by $\pi[s, j, u]$. We define $\tilde{\boldsymbol{x}} = S_{\boldsymbol{\pi}}(\boldsymbol{x})$. For $p \in \{1, 0\}$, normalization of $\boldsymbol{\pi}$ yields value conservation on every line: $\sum_{t=0}^{T-1} S_{\boldsymbol{\pi}}(\boldsymbol{x})[t, j] = \sum_{s=0}^{T-1} \boldsymbol{x}[s, j]$. For $p = \infty$, the same identity holds when the window $\{-\varepsilon, \ldots, \varepsilon\}$ stays within the timeline. The operator is linear and fully differentiable, providing gradients aligned with temporal retimings and leading naturally to the strict projection next.

**Expected occupancy and capacity regularization.** To align the soft distribution with *capacity–1* before strict placement, we track the *expected* packet count at bin $(t, j)$: $\text{occ}_{\boldsymbol{\pi}}[t, j] = \sum_{(s,j)\in\mathcal{A}(\boldsymbol{x})} \sum_{u\in\mathcal{U}_p} \boldsymbol{\pi}[s, j, u] \, \mathbb{I}\{T_p(s, u) = t\}$. We penalize only the excess beyond unit capacity,

$$\text{Cap}(\boldsymbol{\pi}; \boldsymbol{x}) = \frac{1}{|\mathcal{A}(\boldsymbol{x})|} \sum_{j=1}^{N} \sum_{t=0}^{T-1} \big[\text{occ}_{\boldsymbol{\pi}}[t, j] - 1\big]_+^2, \tag{10}$$

where $[z]_+ := \max(z, 0)$ denotes the positive-part (hinge) operator.

**Budget-aware soft surrogates.** The budget in Eq. 6 is enforced exactly by the final projection. During optimization, we guide $\phi$ with smooth surrogates matched to the chosen budget so gradients favor feasible retimings. For $p = \infty$, no surrogate is required because the support $\mathcal{U}_\infty$ already

---

**Algorithm 1** PIL-PGD: Projected-in-the-loop PGD over shift logits

---

1: **Input:** SNN $f$, iterations $T_{\mathrm{pgd}}$, temperature $\kappa$, step size $\alpha$, radius $\varepsilon$, clipping $\phi_{\max}$, weights $\lambda_{\mathrm{cap}}, \lambda_{\mathrm{budget}}$
2: Initialize shift logits $\phi$ with zeros, and mask them *outside $A(x)$*
3: **for** $t = 1$ to $T_{\mathrm{pgd}}$ **do**
4:     $\pi \leftarrow \mathrm{softmax}(\phi/\kappa)$
5:     $\tilde{x} \leftarrow S_{\pi}(x); \ \hat{x} \leftarrow P^*(x; \pi, \mathcal{B}_p(\varepsilon)); \ x_{\mathrm{PIL}} \leftarrow \hat{x} + \big(\tilde{x} - \mathrm{stopgrad}(\tilde{x})\big)$      *# strict projection in-loop*
6:     Compute $\mathcal{J}$ via Eq. 14; Update $\phi \leftarrow \mathrm{clip}_{[-\phi_{\max}, \phi_{\max}]}\big(\phi + \alpha \cdot \mathrm{sign}(\nabla_{\phi}\mathcal{J})\big)$
7: **end for**
8: **Return** $x^{\mathrm{adv}} \leftarrow P^*(x; \pi, \mathcal{B}_p(\varepsilon))$

---

encodes the constraint. For $p = 1$, we use the step cost $C_{s,t} = |t - s|$ to define a soft total timing shift. And for $p = 0$, we use the diagonal mass $\pi_{=}[s, j] = \pi[s, j, s]$ to define a soft move count:

$$S^1_{\mathrm{soft}}(\pi; x) = \sum_{(s,j)\in\mathcal{A}(x)} \sum_{t=0}^{T-1} \pi[s, j, t]\, C_{s,t}, \ \text{and} \ S^0_{\mathrm{soft}}(\pi; x) = \sum_{(s,j)\in\mathcal{A}(x)} \big(1 - \pi_{=}[s, j]\big). \quad (11)$$

We push these quantities toward the target radius $\varepsilon$ with *normalized* hinge penalties

$$\mathcal{R}_p(\pi; \varepsilon) = \big[S^p_{\mathrm{soft}}(\pi; x)/\varepsilon - 1\big]_+, \ \ p \in \{0, 1\}, \ \text{and} \ \mathcal{R}_\infty(\pi; \varepsilon) = 0, \quad (12)$$

which keeps gradients comparable across $\varepsilon$, and the strict projection later enforces the exact budgets.

**Feasible strict projection under budgets.** Given $\pi$, we compute a *strict* discrete retiming $\hat{x} = P^*(x; \pi, \mathcal{B}_p(\varepsilon))$ that operates on $\mathcal{A}(x)$ and enforces *capacity-1*, value conservation, and the budget $\mathcal{B}_p(\varepsilon)$. The projection obeys the following rules: **a) Candidate generation:** For $p = \infty$, the procedure enumerates shifts $u \in \{-\varepsilon, \ldots, \varepsilon\}$ and maps each source $(s, j)$ to the target time $t = T_p(s, u) = s + u$. For $p \in \{0, 1\}$, it enumerates target times $t \in \{0, \ldots, T - 1\}$ with $t \neq s$. **b) Ordering:** The algorithm sorts all candidates once by the descending score $\pi[s, j, u]$ (or $\pi[s, j, t]$). **c) One-pass placement:** The scan traverses the ordered list and places $(s, j) \to (t, j)$ only when the bin $(t, j)$ is free. A placed source releases its origin; an unplaced source remains at $s$. Targets outside the timeline are discarded implicitly by the time-axis clipping used in $S_{\pi}$ and $P^*$. **d) Budget enforcement:** The implementation initializes global counters to the radius and consumes them during the score-ordered scan. For $p = \infty$, it accepts only shifts with $|u| \leq \varepsilon$. For $p = 1$, it sets an integer step budget $B_1 \leftarrow \varepsilon$, traverses candidates in descending $\pi$ (ties by smaller $|t - s|$), places each feasible move, and decrements $B_1$ by $|t - s|$ until $B_1 = 0$. For $p = 0$, it sets a move budget $B_0 \leftarrow \varepsilon$ and decrements $B_0$ by 1 per placement under the same ordering until $B_0 = 0$. The full procedure is in Algorithm 2, and Appendix E further explains how this strict projection achieves the budget constraint $\mathcal{B}_p(\varepsilon)$ exactly while maintaining *capacity–1* and conserving event values.

**Projected-in-the-loop (PIL) straight through.** Optimizing timing moves faces a basic tension: the model must be evaluated on *feasible* inputs that respect capacity and the chosen budget, yet gradients must reflect how small retimings change the loss. Purely strict placement destroys gradients, while purely soft surrogates break the threat model seen by the network. We therefore adopt a backward-pass differentiable approximation tailored to timing: the forward pass uses the strict projection $\hat{x} = P^*(x; \pi, \mathcal{B}_p(\varepsilon))$, and the backward pass follows the soft retiming $\tilde{x} = S_{\pi}(x)$:

$$x_{\mathrm{PIL}} = \hat{x} + \big(\tilde{x} - \mathrm{stopgrad}(\tilde{x})\big). \quad (13)$$

This "projected-in-the-loop" coupling instantiates a straight-through/backward-pass approximation for discrete operations, providing stable gradients without sacrificing exact feasibility at evaluation.

### 4.3 PIL-PGD: PROJECTED-IN-THE-LOOP PGD OVER SHIFT LOGITS

Building on Eq. 13 and the additional loss Eq. 10, 11, we optimize the shift logits by maximizing

$$\mathcal{J} = \mathcal{L}\big(f(x_{\mathrm{PIL}}), y\big) - \lambda_{\mathrm{cap}} \cdot \mathrm{Cap}(\pi; x) - \lambda_{\mathrm{budget}} \cdot \mathcal{R}_p(\pi; \varepsilon). \quad (14)$$

We update logits with a clipped sign-PGD step

$$\phi \leftarrow \mathrm{clip}_{[-\phi_{\max}, \phi_{\max}]}\Big(\phi + \alpha \cdot \mathrm{sign}(\nabla_{\phi}\mathcal{J})\Big), \quad (15)$$

where $\phi_{\max}$ is a hyperparameter that prevents logit saturation and stabilizes the softmax temperature $\kappa$. After the final iteration, we recompute $\pi$ and output the strictly feasible $x^{\mathrm{adv}} = P^*(x; \pi, \mathcal{B}_p(\varepsilon))$.

Table 1: Results on Binary-grid DVS. We report clean Accuracy (Acc., %) and **ASR (%)** under $\mathcal{B}_p$ with $p \in \{\infty, 1, 0\}$. For each budget, we evaluate three radii, and dataset-specific radii are indicated.

| Dataset | Model | Acc. (%) | | ASR (%) | | | | | | | | |
|---|---|---|---|---|---|---|---|---|---|---|---|---|
| | *Budget* | | | $\mathcal{B}_\infty$ | | | $\mathcal{B}_1$ | | | $\mathcal{B}_0$ | | |
| | $\varepsilon \quad \rightarrow$ | | 1 | 2 | 3 | 500 | 750 | 1k | 200 | 300 | 400 |
| **N-MNIST** | ConvNet | 99.06 | 100 | 100 | 100 | 58.9 | 99.9 | 100 | 13.0 | 53.1 | 98.5 |
| | ResNet18 | 99.62 | 100 | 100 | 100 | 69.2 | 97.4 | 100 | 78.9 | 100 | 100 |
| | VGGSNN | 99.64 | 98.9 | 100 | 100 | 26.4 | 65.5 | 94.7 | 18.3 | 81.8 | 99.8 |
| | *Budget* | | | $\mathcal{B}_\infty$ | | | $\mathcal{B}_1$ | | | $\mathcal{B}_0$ | | |
| | $\varepsilon \quad \rightarrow$ | | 1 | 2 | 3 | 2k | 4k | 8k | 1k | 2k | 4k |
| **DVS-Gesture** | ResNet18 | 95.14 | 98.9 | 100 | 100 | 52.6 | 84.7 | 99.6 | 27.7 | 67.9 | 98.5 |
| | VGGSNN | 95.14 | 96.4 | 100 | 100 | 53.3 | 89.8 | 98.5 | 55.8 | 87.2 | 98.9 |
| | SpResF | 91.67 | 92.1 | 96.1 | 100 | 64.4 | 87.3 | 98.4 | 43.8 | 90.9 | 99.2 |
| **CIFAR10-DVS** | ResNet18 | 78.30 | 100 | 100 | 100 | 51.0 | 77.0 | 97.0 | 26.0 | 42.0 | 80.0 |
| | VGGSNN | 78.30 | 100 | 100 | 100 | 47.0 | 72.0 | 98.0 | 25.0 | 46.0 | 73.0 |
| | SpResF | 81.30 | 100 | 100 | 100 | 84.0 | 100 | 100 | 52.0 | 92.0 | 100 |

**Why this objective and update.** Alg. 1 formulates Eq. 14 within the loop: the forward path adopts strictly projected $\hat{x} = P^*(x; \pi, B_p(\varepsilon))$, while the backward path differentiates through the soft shift $S_\pi$. The objective steers logits toward untargeted attacks and adds two terms, the over-occupancy $\mathrm{Cap}$ and the budget penalty $\mathcal{R}_p$. Logits are updated by a clipped sign-PGD step, and the bound $\phi_{\max}$ maintains scale and prevents early near-argmax collapse. With feasibility enforced in the forward and differentiability preserved in the backward, PIL yields stable gradients without violating the threat model. The same routine applies for $p \in \{\infty, 1, 0\}$, changing only $\mathcal{R}_p$ and the projection $P^*$.

## 5 EXPERIMENT

**Datasets and Models.** We choose event datasets: CIFAR10-DVS (Li et al., 2017) (10,000 CIFAR-10 images converted to event streams over 10 classes; $2 \times 128 \times 128$ grid), DVS-Gesture (Amir et al., 2017) (1,063/288 train/test streams, 11 gestures; $2 \times 128 \times 128$), and N-MNIST (Orchard et al., 2015) (60,000/10,000 saccade-rendered MNIST samples; $2 \times 34 \times 34$). Attacks are run only on correctly-classified test samples: all on DVS-Gesture, 1,000 on N-MNIST, and 100 on CIFAR10-DVS. We set time-bin $T = 10$. For N-MNIST, we use ConvNet (Fang et al., 2021a), Spiking ResNet18 (Fang et al., 2021b), and VGGSNN (Deng et al., 2022)). For the other two datasets, we replace ConvNet with SpikingResformer (SpResF) (Shi et al., 2024). All models are directly trained SNNs.

**Evaluation.** We adopt untargeted attacks under $\mathcal{B}_p(\varepsilon)$ with $p \in \{\infty, 1, 0\}$, and evaluate both *binary/integer-grid* events. Our metric is the *Attack Success Rate (ASR)* on set $S$: $\mathrm{ASR}\big(\mathcal{B}_p(\varepsilon)\big) = \frac{1}{|S|} \sum_{(x,y) \in S} \mathbb{I}\big\{ f\big( \mathrm{Attack}\,(x;\, B_p(\varepsilon))\,\big) \neq y \big\}$. We report ASR across the budget $\varepsilon$ for each $p$.

**Implementations.** For $\mathcal{B}_\infty$, we use $\varepsilon \in \{1, 2, 3\}$. For $\mathcal{B}_0$, we set $\{200, 300, 400\}$ on N-MNIST, and $\{1\mathrm{k}, 2\mathrm{k}, 4\mathrm{k}\}$ on DVS-Gesture and CIFAR10-DVS. To keep perturbations comparable, the $\mathcal{B}_1$ radii are twice the $\mathcal{B}_0$ settings. On DVS-Gesture, $\mathcal{B}_0(4\mathrm{k})$ touches 2.45% of spikes, on CIFAR10-DVS the same 4k is 3.84%, and on N-MNIST $\mathcal{B}_0(400)$ is 14.2%, indicating the stealthiness. Alg. 1 uses $\kappa = 1$, $\alpha = 1$, $\phi_{\max} = 10$, $\lambda_{\mathrm{cap}} = 20$, $\lambda_{\mathrm{budget}} = 10$, $T_{\mathrm{pgd}} = 20$ for $\mathcal{B}_\infty$ and $T_{\mathrm{pgd}} = 40$ for $\mathcal{B}_1$ and $\mathcal{B}_0$.

### 5.1 EXPERIMENTAL RESULTS

**Results on Binary DVS Data.** We first evaluate our attacks on binary grids as shown in Tab. 1. Under $\mathcal{B}_\infty$, **small jitter** already drives ASR near saturation; for example, most results reach over 96% at $\epsilon = 1$. Under $\mathcal{B}_1$, ASR increases smoothly with budget with **moderate architecture dependence**; *e.g.*, on DVS-Gesture the SpikingResformer attains 98.5% at $\mathcal{B}_1(4\mathrm{k})$. Under $\mathcal{B}_0$, retiming a small **subset of spikes** is effective; for instance, on CIFAR10-DVS the SpikingResformer achieves 96.0% at 4k touched spikes. Overall, timing-only perturbations are highly effective on binary grids, with susceptibility shaped by the budget and the model family.

**Results on Integer DVS Data.** As shown in Tab. 2, integer DVS grids mirror the binary trend under $\mathcal{B}_\infty$ with **fast saturation**, but are **consistently more robust** under $\mathcal{B}_1$ and $\mathcal{B}_0$ at the same nominal budgets. For example, on DVS-Gesture with ResNet18, the binary grid exceeds 99% ASR at $\mathcal{B}_1 = 8\mathrm{k}$, whereas the integer grid is around the mid-60% at 8k and mid-80% at 16k; a similar gap appears on CIFAR10-DVS. Under $\mathcal{B}_0$, the binary grid reaches high ASR by 4k touched spikes, while the integer grid typically requires up to 8k to approach comparable levels.

Table 2: Results on Integer-grid DVS. We report clean Accuracy (Acc., %) and **ASR (%)** under $\mathcal{B}_p$.

| Dataset | Model | Acc. (%) | | $\mathcal{B}_\infty$ | | | $\mathcal{B}_1$ | | | | $\mathcal{B}_0$ | | |
|---------|-------|----------|---|---|---|---|---|---|---|---|---|---|---|
| | *Budget* $\varepsilon \rightarrow$ | | 1 | 2 | 3 | 500 | 750 | 1k | 1.5k | 200 | 300 | 400 | 600 |
| **N-MNIST** | ConvNet | 99.19 | 100 | 100 | 100 | 61.8 | 99.2 | 100 | 100 | 13.0 | 56.1 | 99.1 | 100 |
| | ResNet18 | 99.62 | 100 | 100 | 100 | 53.9 | 93.6 | 99.8 | 100 | 86.1 | 99.8 | 100 | 100 |
| | VGGSNN | 99.71 | 46.3 | 100 | 100 | 8.3 | 18.5 | 39.9 | 76.7 | 5.8 | 11.2 | 16.1 | 49.8 |
| | *Budget* $\varepsilon \rightarrow$ | | 1 | 2 | 3 | 2k | 4k | 8k | 16k | 1k | 2k | 4k | 8k |
| **DVS-Gesture** | ResNet18 | 94.40 | 71.0 | 83.3 | 93.3 | 17.1 | 40.9 | 65.4 | 85.1 | 8.9 | 35.3 | 68.0 | 98.1 |
| | VGGSNN | 94.79 | 65.9 | 79.9 | 85.0 | 14.7 | 31.9 | 55.0 | 79.1 | 21.6 | 40.7 | 67.4 | 95.9 |
| | SpResF | 92.71 | 70.7 | 79.8 | 84.0 | 36.8 | 52.8 | 68.8 | 80.6 | 26.2 | 46.3 | 70.7 | 80.6 |
| **CIFAR10-DVS** | ResNet18 | 79.20 | 99.0 | 100 | 100 | 21.0 | 37.0 | 66.0 | 85.0 | 7.0 | 18.0 | 33.0 | 73.0 |
| | VGGSNN | 78.80 | 98.0 | 100 | 100 | 26.0 | 43.0 | 67.0 | 87.0 | 16.0 | 25.0 | 46.0 | 78.0 |
| | SpResF | 82.90 | 100 | 100 | 100 | 63.0 | 97.0 | 100 | 100 | 33.0 | 72.0 | 100 | 100 |

**Clean accuracy and robustness.** Accuracy is **comparable** across binary/integer grids for the same model and dataset (*e.g.,* ResNet18 + DVS-Gesture: 95.14 vs. 94.40), while robustness diverges.

**Why integer grids appear more robust under bin-level retiming?** *(i) Presence bias and narrow temporal margin on binary.* Binary inputs are thresholded from integers, so training emphasizes bin-level *presence* features. Under bin-level retiming, shifting isolated spikes across a temporal receptive field can toggle on/off patterns and induce large discrete changes in downstream activations, implying a narrow temporal margin. *(ii) Multiplicity and smoother temporal response on integer.* Integer inputs preserve per-bin multiplicity. A bin-level shift moves count packets that temporal convolutions, pooling, and normalization integrate more smoothly, so the same retiming manifests as a gentler phase change in feature space. To obtain comparable feature variation, retiming typically needs a larger total timing shift. *(iii) Surrogate-gradient and normalization stability.* Although spikes are binary, gradients and normalization act on pre-spike *continuous* variables. With binary inputs, pre-activations concentrate near threshold, and small bin shifts move many units into and out of the surrogate-gradient support, producing spiky gradients and volatile normalization statistics. With integer inputs, pre-activations vary more smoothly and statistics are more stable, yielding broader tolerance to small temporal phase shifts. **Theoretical explanations** are in Appendix A.

**Ablation.** We ablate 3 modules with VGGSNN and DVS-Gesture in Tab. 3. **PIL** (forward with the strict projection while backward with the soft shifter) is *critical*: replacing it with only the soft $\tilde{x}$ consistently lowers ASR (binary $\mathcal{B}_1$: 98.5 → 84.3; integer $\mathcal{B}_\infty$: 85.0 → 63.0), showing that matching the *evaluated* threat while retaining gradients matters. The **budget penalty** $\mathcal{R}_p$ is also *crucial*, with the largest impact

Table 3: Ablation. We report ASR (%). "w/o PIL" denotes replacing $x_{\text{PIL}}$ with $S_\pi(x)$.

| Variant | **Binary Grid** | | | **Integer Grid** | | |
|---------|---|---|---|---|---|---|
| | $\mathcal{B}_\infty(1)$ | $\mathcal{B}_1(8k)$ | $\mathcal{B}_0(4k)$ | $\mathcal{B}_\infty(3)$ | $\mathcal{B}_1(16k)$ | $\mathcal{B}_0(8k)$ |
| Full (ours) | **96.4** | **98.5** | **98.9** | **85.0** | **79.1** | **95.9** |
| w/o PIL | 92.7 | 84.3 | 88.6 | 63.0 | 61.5 | 83.1 |
| w/o Cap | 95.6 | 98.5 | 98.5 | 77.6 | 77.2 | 89.6 |
| w/o $\mathcal{R}_p$ | – | 76.6 | 93.0 | – | 42.4 | 84.9 |

under $\mathcal{B}_1$ (binary: 98.5 → 76.6, integer: 79.1 → 42.4) and noticeable drops for $\mathcal{B}_0$(binary: 98.9→ 93.0, integer: 95.9→84.9). **Capacity regularizer** yields modest gains on binary but larger gains on integer grids; removing it reduces ASR (integer $\mathcal{B}_\infty$: 85.0→77.6). Overall, *PIL + budget awareness* account for most improvements, and capacity control helps when temporal bins become congested.

**Comparisons against raw-event baselines.** In addition, we also benchmark our attack against two strong raw-event baselines, SpikeFool (Büchel et al., 2022) and PDSG-SDA (Lun et al., 2025), under matched tamper-count budgets $B_0$ on N-MNIST, DVS-Gesture, and CIFAR10-DVS. Detailed accuracy–vs.–budget curves and full tables are in Appendix K. Overall, our timing-only, rate-preserving attack remains competitive with these raw-event methods and is often stronger on DVS-Gesture and N-MNIST, despite operating under stricter capacity and timing constraints.

## 5.2 Robustness against adversarially trained models

**Adversarial Training with Retiming Attacks.** To assess robustness when the model is *exposed during training* to our attacks, we adopt the adversarial training (AT) (Madry et al., 2018) paradigm: use attacks under budget $\mathcal{B}_p(\varepsilon)$ as the inner maximization and minimize robust risk in the outer loop:

$$\min_{\theta} \mathbb{E}_{(x,y)\sim\mathcal{D}}\big[\max_{\phi} \mathcal{L}\big(f_\theta\big(P^*(x; \pi, \mathcal{B}_p(\varepsilon))\big), y\big)\big], \text{ s.t. } \pi = \text{softmax}(\phi/\kappa). \quad (16)$$

Since the strict projection $P^*$ is relatively slow, we optimize over $\phi$ using the soft shifter $S_\pi$ and apply $P^*$ at the final step. This keeps the inner loop efficient while still meeting budget constraints.

Table 4: Results of AT with timing-only attacks on VGGSNN and DVS-Gesture. Following commonly used metrics in AT, we report clean accuracy (Acc., %) and **robust accuracy** (%).

| Grid | Adversarially trained model | Acc. | Robust Acc. (%) | | | | | | | | |
|---|---|---|---|---|---|---|---|---|---|---|---|
| | | | $\mathcal{B}_\infty(1)$ | $\mathcal{B}_\infty(2)$ | $\mathcal{B}_\infty(3)$ | $\mathcal{B}_1(2k)$ | $\mathcal{B}_1(4k)$ | $\mathcal{B}_1(8k)$ | $\mathcal{B}_0(1k)$ | $\mathcal{B}_0(2k)$ | $\mathcal{B}_0(4k)$ |
| **Binary** | $\mathcal{B}_\infty(1)$ | 22.92 | 9.72 | 6.60 | 7.29 | 15.63 | 12.50 | 9.03 | 19.45 | 16.67 | 10.42 |
| | $\mathcal{B}_1(8k)$ | 48.26 | 9.37 | 3.82 | 2.43 | 18.05 | 9.37 | 6.25 | 27.77 | 17.01 | 6.25 |
| | $\mathcal{B}_0(4k)$ | 22.57 | 5.90 | 3.47 | 2.43 | 11.11 | 9.38 | 4.17 | 14.58 | 12.15 | 5.56 |
| **Integer** | $\mathcal{B}_\infty(1)$ | 52.08 | 27.08 | 27.43 | 29.86 | 44.79 | 43.40 | 34.72 | 46.87 | 43.75 | 39.23 |
| | $\mathcal{B}_1(8k)$ | 68.75 | 40.62 | 40.98 | 43.40 | 64.58 | 61.81 | 54.51 | 68.40 | 64.58 | 55.56 |
| | $\mathcal{B}_0(4k)$ | 72.22 | 40.97 | 38.19 | 37.50 | 67.01 | 60.41 | 51.38 | 70.49 | 64.58 | 51.38 |

**Results.** We do experiments with VGGSNN on DVS-Gesture. Tab. 4 shows AT on the integer grid yields a better **clean–robust trade-off**. With $\mathcal{B}_1(8k)$, the integer model keeps clean accuracy at $\approx 69\%$ and reaches robust accuracy $> 60\%$ at small $\mathcal{B}_1$ radii, settling in the mid-50% at 8k. The gains transfer to $\mathcal{B}_0$ ($\approx 55\text{–}68\%$) and to $\mathcal{B}_\infty$ ($\approx 41\text{–}43\%$). Training with $\mathcal{B}_0(4k)$ on the integer grid gives the highest clean accuracy ($\approx 72\%$) and robust accuracy $> 50\%$. On the binary grid, the clean accuracy reaches around 23–48%, and robustness remains weak even under the matched budget.

**Insights.** *(i)* **Robustness tracks clean accuracy**: if clean collapses, robustness is limited. *(ii)* **Integer temporal inertia**: integer grids aggregate multi-bit counts per bin, adding inertia making the inner maximization less destructive; this stabilizes training and improves cross-budget transfer. *(iii)* **Destructive inner maximization**: our attack is highly destructive. Even with AT it substantially reduces clean accuracy, while robustness gains remain modest, highlighting the need for more practical defenses and training schemes in future work. See Appendix F for a detailed comparison with standard non-timing AT baselines. Additional AT experiments, including multi-norm timing AT and TRADES-based timing AT, are reported in Appendix H.

## 5.3 DISCUSSION

**Time-shift distribution.** Fig. 2 illustrates the time-shift on the binary DVS-Gesture, and shows a clear polarity pattern: the positive channel tends to delay (*red–shift*) and the negative channel tends to advance (*blue–shift*). Under $\mathcal{B}_1$, most shifts are 1 bin with a smaller mass at 2. Under $\mathcal{B}_0$, shifts reach farther.

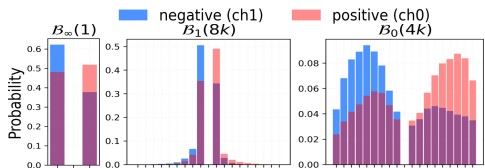

Figure 2: Time-shift distribution (w/o 0).

**Time-bins** $T$**.** We evaluate under different time bins in Tab. 5. On the *binary* grid, finer bins create more *active time bins*, so under a fixed $\mathcal{B}_1$ or $\mathcal{B}_0$ budget the attacker must retime many more positions to achieve the same effect, which drives a sharp ASR drop (*e.g.,* $\mathcal{B}_1$: $90.6 \to 29.1$ from $T = 20$ to $T = 40$). On the *integer* grid, finer binning mostly *redistributes* the same spike counts. Temporal convolutions and normalization integrate summed counts within each receptive field, so the integrated mass remains stable. The attacker can still *rephase*

Table 5: ASR (%) on VGGSNN (DVS-Gesture). We vary time bins $T$, neuron models (PLIF, PSN), and targeted attacks. "Default" is the main config.

| Variant | | Binary Grid | | | Integer Grid | | |
|---|---|---|---|---|---|---|---|
| | | $\mathcal{B}_\infty(1)$ | $\mathcal{B}_1(8k)$ | $\mathcal{B}_0(4k)$ | $\mathcal{B}_\infty(3)$ | $\mathcal{B}_1(16k)$ | $\mathcal{B}_0(8k)$ |
| **Default setup** | | 96.4 | 98.5 | 98.9 | 85.0 | 79.1 | 95.9 |
| **Time-bins** | $T = 20$ | 97.1 | 90.6 | 48.0 | 97.4 | 79.8 | 92.4 |
| | $T = 30$ | 93.8 | 49.2 | 23.0 | 96.0 | 70.5 | 87.0 |
| | $T = 40$ | 89.9 | 29.1 | 13.3 | 95.0 | 57.8 | 81.7 |
| **Neuron** | PLIF | 96.7 | 99.6 | 96.3 | 95.1 | 92.9 | 98.9 |
| | PSN | 100 | 100 | 100 | 97.3 | 79.3 | 88.6 |
| **Targeted Attack** | | 26.8 | 57.3 | 62.8 | 24.6 | 29.7 | 47.2 |

high-count packets under $\mathcal{B}_1$, and under $\mathcal{B}_0$ moving a few high-mass bins remains impactful. This mass-preserving yet phase-shifting effect explains why ASR stays high at $T = 20$ across budgets.

**Neuron.** We evaluate neuron models in Tab. 5. PLIF (Fang et al., 2021a) surpasses LIF in most cases (*e.g.,* integer $\mathcal{B}_1$: 92.9). PSN (Fang et al., 2023) saturates on the binary grid and remains high on the listed integer entry, indicating that richer membrane dynamics permit easier timing manipulations.

**Targeted Attack.** We evaluate targeted attacks with 0 as the target label in Tab. 5. Targeted ASR is lower than untargeted across both grids, with the largest drop under $\mathcal{B}_\infty$. Increasing the budget may mitigate this gap. Appendix M further reports random-target experiments on integer DVS-Gesture / VGGSNN, comparing our timing attack with the raw-event baseline PDSG-SDA (Lun et al., 2025). Under matched $\mathcal{B}_0$ budgets, our attack attains targeted ASR comparable to PDSG-SDA, indicating that the relatively low targeted success is an intrinsic difficulty of attacks against event grids.

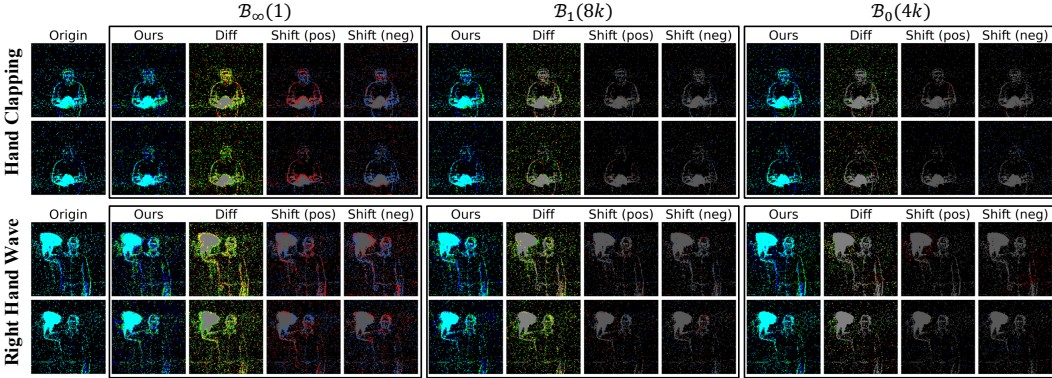

Figure 3: Visualization of DVS-Gesture across selected frames (Frame 3/6 in odd/even rows). **Origin** shows the clean frame (positive in green, negative in blue), **Ours** shows the retimed frame under a fixed retiming budget, **Diff** highlights changes (new in green, removed in red, unchanged in gray, polarity swap in yellow), and **Shift (pos) / Shift (neg)** map per-polarity time shifts (delay in red, advance in blue, zero in gray, no spike in black). Shift color intensity scales with the absolute shift.

**Transferability.** Our transferability and multi-model ensemble experiments follow the standard surrogate-to-victim and ensemble-based protocols from transferable attacks on image models (Liu et al., 2017; Mahmood et al., 2021). In Appendix B, we experiments on VGGSNN → ResNet18 and SpikingResformer. Our attack transfers across architectures, with higher success on the CNN-like ResNet18 due to the VGGSNN surrogate. Further work may explore stronger targeted objectives, and surrogate ensembles to improve transferability. Additional results on ensemble-based multi-model and multi-norm multi-model timing attacks, following Xu et al. (2025), are in Appendix I.

**Visualization analysis.** Fig. 3 compares three timing budgets on binary events. With $\mathcal{B}_\infty(1)$, the attack induces local jitter: *Shift* shows thin red and blue halos near edges, and *Diff* concentrates on contours. With $\mathcal{B}_1(8k)$, shifts are redistributed across many pixels, so *Shift* and *Diff* spread more and intensities rise. With $\mathcal{B}_0(4k)$, only a few spikes move, giving sparse bright red and blue points and a few *Diff* hits, often away from the main motion. Across budgets, the adversarial frame stays close to the clean one because counts are preserved and only timing changes. A recurring pattern is that **key action strokes in hands and arms remain largely intact**, while many retimings land on background or incidental spikes. This exposes sensitivity to **nonsalient timing** that still shapes **membrane integration** and decision timing. It motivates defenses that reduce such reliance, for example, saliency-aware timing regularization on foreground regions, AT with timing-only budgets that emphasize **foreground stability**, and objectives that penalize **background timing**.

**Robustness to filtering defenses.** We evaluate three simple label-free filtering defenses (refractory, temporal, and spatial smoothing) on binary DVS-Gesture / VGGSNN. See Appendix L for details. Across a wide range of defense strengths, moderate filtering already costs tens of percentage points in clean accuracy while barely reducing the ASR of our timing-only attack, and our method is at least as robust to these defenses as the value-based PDSG-SDA (Lun et al., 2025). Only extremely aggressive filtering can substantially suppress our attack, but this simultaneously collapses clean accuracy, underscoring that simple intensity-based pre-processing is insufficient against our attack.

# 6 CONCLUSION

We establish spike retiming as a timing-only threat to event driven SNNs under budgets $\mathcal{B}_\infty, \mathcal{B}_1, \mathcal{B}_0$. We introduce projected in the loop (PIL): the forward pass uses a strictly projected input and the backward differentiates through a soft shifter, with a capacity regularizer and a budget penalty. Across multiple event datasets and binary and integer grids, the attack achieves high success with a small footprint. Our analysis shows integer grids are more robust because multiplicity smooths pre-activations and stabilizes normalization. Adversarial training yields partial gains yet reduces clean accuracy, motivating practical timing aware defenses. These results set a reference for temporal robustness and elevate timing to a primary axis for evaluation and defense in event driven SNNs.

## ACKNOWLEDGEMENT

This work was carried out at the Rapid-Rich Object Search (ROSE) Lab, Nanyang Technological University (NTU), Singapore. This research is supported by the National Research Foundation, Singapore and Infocomm Media Development Authority under its Trust Tech Funding Initiative, and the National Research Foundation, Singapore, under its NRF Professorship Award No. NRF-P2024-001. Any opinions, findings and conclusions or recommendations expressed in this material are those of the author(s) and do not reflect the views of National Research Foundation, Singapore and Infocomm Media Development Authority.

## ETHICS STATEMENT

This work analyzes timing-only adversarial attacks on event-driven spiking neural networks to expose failure modes and inform defenses. Aware of dual-use risks, we restrict experiments to public benchmarks and open-source models (never deployed or proprietary systems) and disclose only what is necessary for scientific reproducibility without enabling turnkey misuse. Any released artifacts will use a research-only license with default "evaluation-only" configurations; potentially abusable components (e.g., automated black-box attack pipelines) are down-scoped. We also outline practical safeguards for defenders (timing-jitter augmentation, count-invariant timing checks, and certification baselines). No new data were collected; no human subjects or personally identifiable information are involved; usage complies with dataset licenses. We adhere to the ICLR Code of Ethics and research-integrity norms, report limitations transparently, and keep computational budgets modest to limit environmental impact.

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

## A   ANALYSIS OF GRADIENTS AND NORMALIZATION UNDER BINARY VS. INTEGER INPUTS

**Setup.**   Let $X_t \in \mathbb{R}^{C \times H \times W}$ denote the per-bin input (either binary $\{0, 1\}$ or nonnegative integer counts), and let

$$A_t \ = \ (W * X)_t + b \tag{17}$$

be the first pre-activation (spatio-temporal convolution or linear filtering absorbed into $W$). Spikes are $S_t = \Theta(A_t - V_{\text{th}})$ in the forward pass, and a surrogate $\sigma$ (with derivative $\sigma'$) is used in backprop. A unit bin shift is written $T_1 X$, and the retiming-induced perturbations are

$$\Delta X \ := \ T_1 X - X, \qquad \Delta A_t \ := \ A_t^{(1)} - A_t \ = \ (W * \Delta X)_t. \tag{18}$$

We analyze the *single-bin* shift; multi-bin shifts and multi-spike patterns follow by additivity/triangle inequality.

**1) Retiming changes pre-activations in a Lipschitz way.**   By Young's inequality for convolutions, for $p \in \{1, 2\}$,

$$\|\Delta A\|_p \ \leq \ \|W\|_1 \|\Delta X\|_p. \tag{19}$$

Hence any bound we derive in terms of $\|\Delta A\|_p$ immediately translates to $\|\Delta X\|_p$ via $\|W\|_1$.

**2) Surrogate-gradient support varies Lipschitzly under small shifts.**   Backprop gradients w.r.t. pre-activations take the form

$$g_t \ = \ \frac{\partial \mathcal{L}}{\partial A_t} \ \approx \ \frac{\partial \mathcal{L}}{\partial S_t} \cdot \sigma'(A_t - V_{\text{th}}). \tag{20}$$

To quantify how many units contribute nontrivially to surrogate gradients, replace the hard band indicator by a smooth $C^1$ bump $\rho : \mathbb{R} \rightarrow [0, 1]$ with $\text{Lip}(\rho) < \infty$ and define the $\delta$-*smoothed gradient-support mass*

$$\mathcal{M}_\delta(\tau) \ = \ \mathbb{E}\left[\rho\left(\frac{A_t^{(\tau)} - V_{\text{th}}}{\delta}\right)\right]. \tag{21}$$

By the mean-value theorem and Eq. 18,

$$\left|\mathcal{M}_\delta(\tau{+}1) - \mathcal{M}_\delta(\tau)\right| \ \leq \ \frac{\text{Lip}(\rho)}{\delta} \mathbb{E}\left[|A_t^{(\tau+1)} - A_t^{(\tau)}|\right] \ \leq \ \frac{\text{Lip}(\rho)}{\delta} \|W\|_1 \mathbb{E}\left[\|\Delta X\|_1\right]. \tag{22}$$

Thus, the *shift-sensitivity* of the surrogate-gradient support is proportional to the temporal variation $\|\Delta X\|_1$ and scaled only by model/surrogate constants.

**3) Normalization drifts (mean/variance) are controlled by $\Delta X$.**   Let per-channel running statistics over a window of $N$ bins be

$$\mu(\tau) = \frac{1}{N} \sum_t A_t^{(\tau)}, \qquad \sigma^2(\tau) = \frac{1}{N} \sum_t \left(A_t^{(\tau)} - \mu(\tau)\right)^2. \tag{23}$$

A unit shift changes the mean by

$$|\mu(\tau{+}1) - \mu(\tau)| \ = \ \left|\frac{1}{N} \sum_t \left(A_t^{(\tau+1)} - A_t^{(\tau)}\right)\right| \ \leq \ \frac{1}{N} \|\Delta A\|_1 \ \leq \ \frac{\|W\|_1}{N} \|\Delta X\|_1. \tag{24}$$

For the variance, expanding $\sigma^2(\tau{+}1) - \sigma^2(\tau)$ and collecting first/second-order terms gives

$$\left|\sigma^2(\tau{+}1) - \sigma^2(\tau)\right| \ \leq \ \frac{2\,\sigma(\tau)}{\sqrt{N}} \|\Delta A\|_2 \ + \ \frac{1}{N} \|\Delta A\|_2^2 \ + \ |\mu(\tau{+}1) - \mu(\tau)|^2, \tag{25}$$

and by Eq. 19–24 this is bounded by $\|\Delta X\|_p$ as well. Hence both mean and variance drifts across small shifts are *linearly controlled* by $\|\Delta X\|$.

**4) Normalized pre-activation perturbation shrinks with the baseline scale.** Consider the (affine-free) channel-wise normalization

$$Z_t \;=\; \frac{A_t - \mu}{\sigma}, \qquad Z_t' \;=\; \frac{A_t + \Delta A_t - (\mu + \Delta\mu)}{\sigma + \Delta\sigma}. \tag{26}$$

A first-order expansion in $(\Delta A, \Delta\mu, \Delta\sigma)$ yields

$$\Delta Z_t \;\approx\; \frac{\Delta A_t - \Delta\mu}{\sigma} \;-\; \frac{A_t - \mu}{\sigma^2}\,\Delta\sigma. \tag{27}$$

Taking $\ell_2$ norms and using Cauchy–Schwarz together with Eq. 19–25, there exists $c_N = 1 + \frac{2}{\sqrt{N}}$ such that

$$\|\Delta Z\|_2 \;\leq\; \frac{c_N}{\sigma}\,\|\Delta A\|_2 \;\leq\; \frac{c_N\,\|W\|_1}{\sigma}\,\|\Delta X\|_2. \tag{28}$$

Interpretation: for the *same* retimed mass $\|\Delta X\|_2$ (i.e., same number of touched spikes and same bin shift), the normalized change is inversely proportional to the baseline standard deviation $\sigma$. Integer inputs aggregate multiplicities within a bin; under mild independence/sparsity of contributing atoms, this increases the pre-activation variance (hence $\sigma$), so the normalized perturbation $\|\Delta Z\|_2$ becomes smaller.

**5) Temporal margin and its dispersion.** Define a smoothed temporal margin

$$\mathrm{Mar}_\delta(\tau) \;=\; \mathbb{E}\Big[\psi_\delta\big(A_t^{(\tau)} - V_{\mathrm{th}}\big)\Big], \tag{29}$$

where $\psi_\delta$ is an even $C^1$ function that equals $|z|$ convolved with a width-$\delta$ mollifier (matching the surrogate bandwidth). By the mean-value theorem and Eq. 18,

$$\big|\mathrm{Mar}_\delta(\tau+1) - \mathrm{Mar}_\delta(\tau)\big| \;\leq\; L_{\psi,\delta}\,\mathbb{E}\big[\,|A_t^{(\tau+1)} - A_t^{(\tau)}|\,\big] \;\leq\; L_{\psi,\delta}\,\|W\|_1\,\mathbb{E}\big[\|\Delta X\|_1\big]. \tag{30}$$

Consequently the *dispersion across small shifts*, $\mathrm{Var}_\tau\big(\mathrm{Mar}_\delta(\tau)\big)$, is larger when $\|\Delta X\|$ is larger and/or more irregular; the converse yields steadier margins.

**6) Why integer > binary for timing robustness.** Two mechanisms combine:

*(A) Larger baseline scale $\sigma$ for integer inputs.* Because integer grids preserve multiplicity within each bin, $A_t$ aggregates more contributing atoms than binary grids (which only encode presence/absence). Under mild independence and comparable per-atom weights, $\mathrm{Var}(A_t)$ grows with the expected count per bin, so $\sigma$ is larger for integer than for binary. By Eq. 28, the *normalized* effect of a fixed retimed mass $\|\Delta X\|_2$ is therefore smaller for integer inputs, implying:

- **(i) Larger total shift needed.** To achieve the *same* normalized change (hence comparable feature-phase displacement and decision impact), an attacker must increase $\|\Delta X\|_2$ proportionally to $\sigma$, i.e., needs a larger *total timing shift* on integer inputs.
- **(ii) More stable gradients/normalization.** The same retiming budget produces smaller $\|\Delta Z\|_2$ (Eq. 28), smaller changes of gradient-support mass (Eq. 22 with $A$ replaced by $Z$), and smaller mean/variance drifts (Eqs. 24–25) on integer inputs.

*(B) Smaller temporal variation $\|\Delta X\|$ for integer under rate-smoothness.* Let the underlying per-location event rate vary smoothly across adjacent bins. Binarization introduces on/off boundaries: even small rate fluctuations create frequent $\{0, 1\}$ flips, inflating the temporal difference $\|T_1 X - X\|$ relative to the aggregate count change. In contrast, integer counts change more *gradually* across bins when rates are smooth, yielding a smaller typical $\|\Delta X\|$ for the *same* underlying dynamics. Plugging this into Eq. 22, Eq. 24, and Eq. 30 shows that gradient-support variation, normalization drift, and margin dispersion are all smaller for integer inputs.

**Takeaway.** Eq. 19–28 and 30 together establish that, under the same retiming budget, integer inputs (i) *attenuate* normalized perturbations via a larger baseline $\sigma$, and (ii) *reduce* shift-sensitivity by exhibiting smaller temporal differences $\|\Delta X\|$ when rates are smooth. These two effects precisely explain the main-text statements: (ii) retiming on integer inputs typically requires a larger total timing shift to match the effect; and (iii) surrogate gradients and normalization statistics are more stable under small shifts.

Table 6: Transfer ASR (%) on **DVS-Gesture**: attacks crafted on **VGGSNN** (surrogate) and evaluated on targets.

| Target Model | Binary Grid | | | | | | | Integer Grid | | | | | | |
|---|---|---|---|---|---|---|---|---|---|---|---|---|---|---|
| | $\mathcal{B}_\infty$ | | | $\mathcal{B}_1$ | | $\mathcal{B}_0$ | | $\mathcal{B}_\infty$ | | | $\mathcal{B}_1$ | | $\mathcal{B}_0$ | |
| | 1 | 2 | 3 | 8k | 16k | 4k | 8k | 1 | 2 | 3 | 8k | 16k | 4k | 8k |
| ResNet18 | 75.9 | 83.5 | 86.5 | 66.0 | 82.1 | 81.0 | 90.8 | 32.2 | 42.1 | 45.0 | 25.6 | 39.1 | 43.2 | 65.9 |
| SpikingResformer | 47.0 | 56.2 | 61.6 | 45.9 | 51.8 | 55.4 | 62.7 | 30.7 | 30.4 | 34.4 | 19.0 | 26.3 | 16.4 | 31.5 |

## B TRANSFERABILITY

We evaluate transfer from **VGGSNN** to **ResNet18** and **SpikingResformer** on DVS-Gesture in Tab. 6, and compare against the white-box ASR in Tabs. 1 and 2. On the *binary* grid, transfer is strong and grows with budget, with *ResNet18* consistently higher than *SpikingResformer*. For example, increasing $\mathcal{B}_1$ or $\mathcal{B}_0$ raises ASR on the targets toward the corresponding white-box levels. This aligns with shared inductive biases in early temporal filters, so presence features and phase edges crafted on VGGSNN generalize to CNN-like targets.

On the *integer* grid, transfer is clearly weaker at small $\mathcal{B}_\infty$ and $\mathcal{B}_1$ radii, while larger $\mathcal{B}_0$ still yields meaningful transfer. Integer inputs preserve multiplicity and induce smoother preactivation and normalization, so the inner solution becomes more model-aligned and less cross-model, which limits generalization from the surrogate. Overall, *binary* attacks transfer more readily, *ResNet18* is an easier target than *SpikingResformer*, and increasing the budget narrows the gap to white-box performance, especially under $\mathcal{B}_0$ where a few high-impact retimings remain effective across models.

## C USE OF LARGE LANGUAGE MODELS (LLMS)

We used a large language model (LLM) solely as a writing assistant for *language editing*—grammar correction, wording/fluency polishing, and minor rephrasing for clarity. The LLM was *not* involved in research ideation, problem formulation, methodology or experiment design, coding, data analysis, result generation, or citation selection. All technical content and conclusions were authored and verified by the human authors, who take full responsibility for the paper. The LLM is not eligible for authorship.

## D ETHICS STATEMENT

This work analyzes timing-only adversarial attacks on event-driven spiking neural networks to expose failure modes and inform defenses. Aware of dual-use risks, we restrict experiments to public benchmarks and open-source models (never deployed or proprietary systems) and disclose only what is necessary for scientific reproducibility without enabling turnkey misuse. Any released artifacts will use a research-only license with default "evaluation-only" configurations; potentially abusable components (e.g., automated black-box attack pipelines) are down-scoped. We also outline practical safeguards for defenders (timing-jitter augmentation, count-invariant timing checks, and certification baselines). No new data were collected; no human subjects or personally identifiable information are involved; usage complies with dataset licenses. We adhere to the ICLR Code of Ethics and research-integrity norms, report limitations transparently, and keep computational budgets modest to limit environmental impact.

## E STRICT PROJECTION $P^*$ AND RATE PRESERVATION

In the main paper, we define the feasible assignment set in Eq. (5) by requiring (i) a *capacity-1* constraint along each event line and (ii) a rate-preserving, timing-only adversary that never changes event counts or amplitudes. Here we make the strict projection $P^*(x, \pi, \mathcal{B}_p)$ explicit for the global $L_0$ budget case and clarify why it is rate-preserving by construction.

### E.1 Algorithmic definition under global $L_0$ budget

We work on flattened event lines: an input $x \in \mathbb{R}^{T \times B \times C \times H \times W}$ is reshaped to $x \in \mathbb{R}^{T \times N}$, where $N = BCHW$ indexes event lines $j \in \{1, \ldots, N\}$. A non-zero entry $x[s, j]$ denotes one *event packet* (a single spike on binary grids or an integer-valued packet on integer grids). The learned shift logits are turned into probabilities $\pi[s, j, t] \in [0, 1]$ over target times $t \in \{1, \ldots, T\}$ for each source packet $(s, j)$.

The global $L_0$ budget $B_0$ upper-bounds the *number of packets* that may be moved (each moved packet consumes one unit of budget). The strict projection $P^*$ then greedily selects packet moves $(s \to t, j)$ by prioritizing high shift probabilities (and, as a tie-breaker, shorter temporal distances), while enforcing capacity-1 and the global budget. The complete procedure is given in Algorithm 2.

Intuitively, packets whose candidates are never accepted remain at their original time indices $s$, while accepted candidates correspond to pure *moves* of the entire packet from $(s, j)$ to $(t, j)$ on the same event line. The additional checks with `occupied` and `reserved` guarantee that no time bin $(t, j)$ ever hosts more than one packet after projection, and that we never overwrite an original packet that is still planned to stay.

### E.2 Rate preservation by construction

The above algorithm is rate-preserving in a strong sense:

- **Each original packet appears exactly once after projection.** Every packet is either moved once or left at its original location. No operation creates new packets or deletes existing ones; the procedure only changes their time coordinates.
- **Packets never change event lines.** Moves are always of the form $(s, j) \to (t, j)$, so packets only move along the time axis on a fixed line $j$.

Consequently, for every event line $j$, we have exact preservation of the event multiset along time:

$$\big\{\mathrm{adv}[t, j]\big\}_{t=1}^{T} \equiv \big\{x[t, j]\big\}_{t=1}^{T}, \tag{31}$$

and hence the per-line and global event counts are preserved:

$$\sum_{t=1}^{T} \mathrm{adv}[t, j] = \sum_{t=1}^{T} x[t, j], \qquad \sum_{t,j} \mathrm{adv}[t, j] = \sum_{t,j} x[t, j]. \tag{32}$$

Equivalently, $P^*$ implements a per-line permutation of non-zero packets along the time axis (with some packets possibly staying fixed), which is *strictly* rate-preserving for both binary and integer event grids.

In our projected-in-the-loop (PIL) optimization, the only approximation appears in the backward pass: gradients are computed through the soft operator $S_\pi$ on an expected retiming, while the forward evaluation of the SNN always uses the strictly projected $P^*(x, \pi, \mathcal{B}_p)$. All adversarial examples seen by the model therefore satisfy the rate-preservation equalities above.

### E.3 Empirical sanity check of rate preservation

To verify that our implementation matches the intended behavior, we performed a systematic sanity check on all experiments in the main paper (all datasets, models, budgets, and random seeds):

- For each adversarial example, we compared the per-line event configurations before and after projection.
- For every event line $j$, we confirmed that the sorted multiset $\{\mathrm{adv}[t, j]\}_{t=1}^{T}$ coincides with $\{x[t, j]\}_{t=1}^{T}$, i.e., packets are only permuted in time and never added, removed, or rescaled.

These checks were satisfied for all runs, confirming that $P^*$ is rate-preserving in practice for both binary and integer grids. We will summarize aggregate statistics of this check in a small table in the supplementary material.

---

**Algorithm 2** Strict projection $P^*(x, \pi, \mathcal{B}_p)$ with global $L_0$ budget

---

1: **Input:** flattened events $x \in \mathbb{R}^{T \times N}$ (time $\times$ event lines), shift probabilities $\pi \in [0,1]^{T \times N \times T}$, global $L_0$ budget $B_0$
2: **Output:** projected events $\text{adv} = P^*(x, \pi, \mathcal{B}_p)$ reshaped back to $[T, B, C, H, W]$

3: *# identify active packets and candidate moves*
4: $\text{has\_src}[s, j] \leftarrow \mathbf{1}\{x[s, j] > 0\}$ for all $s \in \{1, \dots, T\}, j \in \{1, \dots, N\}$
5: $\mathcal{C} \leftarrow \emptyset$
6: **for all** $(s, j)$ s.t. $\text{has\_src}[s, j] = 1$ **do**
7:     **for** $t = 1$ **to** $T$ **do**
8:         **if** $t \neq s$ **then**
9:             add $(s, j, t)$ to $\mathcal{C}$
10:         **end if**
11:     **end for**
12: **end for**

13: *# priority score: probability first, distance as tie-breaker*
14: choose tiny $\varepsilon > 0$
15: **for all** $(s, j, t) \in \mathcal{C}$ **do**
16:     $\text{key}(s, j, t) \leftarrow \pi[s, j, t] + \varepsilon\big((T - 1) - |t - s|\big)$
17: **end for**
18: sort $\mathcal{C}$ in descending order of $\text{key}(s, j, t)$

19: *# initialize states*
20: $\text{occupied}[j, t] \leftarrow \text{False}$, for all $j, t$              *# targets already taken by moved packets*
21: $\text{reserved}[j, t] \leftarrow \text{has\_src}[t, j]$, for all $j, t$        *# original packets planned to stay*
22: $\text{moved}[j, s] \leftarrow \text{False}$, for all $j, s$
23: $\text{adv}[s, j] \leftarrow 0$, for all $s, j$
24: $B_{\text{rem}} \leftarrow B_0$

25: *# greedy retiming under capacity and $L_0$ budget*
26: **for all** $(s, j, t) \in \mathcal{C}$ in sorted order **do**
27:     **if** $B_{\text{rem}} \leq 0$ **then**
28:         **break**                           *# budget exhausted*
29:     **end if**
30:     **if** $\text{moved}[j, s]$ **then**
31:         **continue**
32:     **end if**                        *# source already moved*
33:     **if** $\text{occupied}[j, t]$ **then**
34:         **continue**
35:     **end if**                        *# target already taken*
36:     **if** $\text{reserved}[j, t]$ **then**
37:         **continue**
38:     **end if**            *# collides with stay-at-source packet*
39:     *# accept candidate: move whole packet from $(s, j)$ to $(t, j)$*
40:     $\text{adv}[t, j] \leftarrow x[s, j]$
41:     $\text{occupied}[j, t] \leftarrow \text{True}$
42:     $\text{moved}[j, s] \leftarrow \text{True}$
43:     $\text{reserved}[j, s] \leftarrow \text{False}$
44:     $B_{\text{rem}} \leftarrow B_{\text{rem}} - 1$
45: **end for**

46: *# packets not moved stay at their original time*
47: **for all** $(s, j)$ s.t. $\text{has\_src}[s, j] = 1$ and $\text{moved}[j, s] = \text{False}$ **do**
48:     $\text{adv}[s, j] \leftarrow x[s, j]$
49: **end for**
50: **Return** adv reshaped back to $[T, B, C, H, W]$

---

### E.4 COMPARISON TO NON-TIMING ATTACKS

Standard image/event-domain attacks, such as PGD on integer event grids or attacks that insert and delete spikes, inevitably change event counts or intensities. They cannot enforce strict equalities such as $\sum_t \text{adv}[t, j] = \sum_t x[t, j]$ and $\{\text{adv}[t, j]\}_{t=1}^{T} = \{x[t, j]\}_{t=1}^{T}$ for all $j$ as hard constraints. Even if some non-timing attacks happen to induce small average rate drift in certain settings, this is incidental rather than guaranteed by the threat model.

Table 7: Adversarial training on DVS-Gesture (VGGSNN) evaluated in terms of attack success rate (ASR, %; lower is better). The first six rows are standard non-timing AT baselines (including $\ell_1$-APGD-based AT (Croce & Hein, 2021) and binary $\ell_0$ flip AT in the spirit of (Balkanski et al., 2020)); the last three rows are our timing-only AT.

| Training scheme | clean | $B_\infty$ | | | $B_1$ | | | $B_0$ | | | Avg ASR |
|---|---|---|---|---|---|---|---|---|---|---|---|
| | | 1 | 2 | 3 | 2000 | 4000 | 8000 | 1000 | 2000 | 4000 | |
| $\ell_\infty$ PGD AT ($\epsilon$=0.5) | 64.24 | 45.41 | 50.27 | 55.14 | 10.81 | 21.08 | 36.76 | 8.65 | 14.05 | 34.05 | 30.69 |
| $\ell_\infty$ PGD AT ($\epsilon$=0.4) | 71.88 | 48.79 | 54.59 | 57.97 | 14.98 | 25.60 | 40.58 | 12.56 | 23.19 | 37.20 | 35.05 |
| binary $\ell_0$ flip AT ($r$=0.32) | 77.78 | 54.91 | 67.86 | 71.88 | 10.71 | 22.32 | 43.30 | 8.48 | 18.30 | 39.29 | 37.45 |
| binary $\ell_0$ flip AT ($r$=0.45) | 74.31 | 53.74 | 63.55 | 66.82 | 10.75 | 21.96 | 38.32 | 8.41 | 20.56 | 35.05 | 35.46 |
| $\ell_1$-PGD AT ($\tau$=10000) | 69.10 | 54.17 | 57.69 | 52.66 | 15.53 | 21.56 | 42.11 | 16.53 | 24.57 | 45.63 | 36.72 |
| $\ell_1$-PGD AT ($\tau$=14000) | 67.01 | 51.81 | 58.03 | 55.44 | 10.88 | 22.80 | 43.01 | 13.99 | 23.32 | 40.41 | 35.52 |
| **Timing AT ($B_\infty$=1)** | 52.08 | 48.00 | 47.33 | 42.67 | 14.00 | 16.67 | 33.33 | 10.00 | 15.99 | 24.67 | 28.07 |
| **Timing AT ($B_1$=8000)** | 68.75 | 40.92 | 40.39 | 36.87 | 6.07 | 10.09 | 20.71 | 0.51 | 6.07 | 19.19 | **20.09** |
| **Timing AT ($B_0$=4000)** | 72.22 | 43.27 | 47.12 | 48.08 | 7.21 | 16.35 | 28.86 | 2.40 | 10.58 | 28.86 | **25.86** |

By contrast, our spike-retiming adversary explicitly characterizes and optimizes the worst-case perturbations under a *strict* rate-preservation constraint. This makes it qualitatively different from norm-bounded perturbations in the value or count space and highlights spike retiming as a distinct, timing-only attack surface.

## F   COMPARISON WITH STANDARD NON-TIMING ADVERSARIAL TRAINING

Following previous work (Mukhoty et al., 2025), here we compare our timing-only adversarial training (AT) with standard non-timing AT schemes on **DVS-Gesture** (integer event grid) and **VGGSNN**. In the main paper, Table 4 reports *robust accuracy*; for the same trained models, we additionally report *attack success rate (ASR)* here for easier head-to-head comparison.

We follow standard practice and instantiate the $\ell_1$ AT baseline with the $\ell_1$-APGD attack of Croce and Hein (Croce & Hein, 2021). For the binary $\ell_0$ flip baselines, we adopt a pixel-flip style sparse attack on binary images in the spirit of Balkanski et al. (Balkanski et al., 2020), adapted to our event-grid setting. For $\ell_\infty$, we follow the PGD AT (Madry et al., 2018).

**Setup (dataset, metric, and evaluation).**   All results in this section are on DVS-Gesture with VGGSNN. ASR is computed *only over samples that are correctly classified under the clean model*, exactly as in our main attack experiments. Robust accuracy and ASR are related by

$$\text{ASR} = 100\% - \text{robust accuracy}.$$

At test time, we evaluate nine timing-only attacks with budgets $B_\infty \in \{1, 2, 3\}$, $B_1 \in \{2000, 4000, 8000\}$, and $B_0 \in \{1000, 2000, 4000\}$. For each training scheme, we report: (i) `clean` accuracy (%); (ii) nine ASR values (lower is better); (iii) the mean of the nine ASR values (`Avg ASR`).

The first six rows in Table 7 are standard non-timing AT baselines; the last three rows are our timing-only AT ("Spike Retiming AT") with different inner-loop budgets.

**Clean–robust trade-off vs. standard AT.**   From Table 7, we observe that the best non-timing AT baselines (across $\ell_\infty$, binary $\ell_0$, and $\ell_1$) achieve clean accuracies in the range 67%–78% with average ASR around 30%–37% (e.g., $\ell_\infty$ PGD AT with $\epsilon$=0.5 has clean 64.24% and Avg ASR 30.69%). In contrast, our timing-only AT attains: *(i)* for $B_1$=8000, clean 68.75% and **Avg ASR** 20.09%; *(ii)* for $B_0$=4000, clean 72.22% and **Avg ASR** 25.86%. Thus, for similar clean accuracy (around 69–72%), timing AT reduces the average ASR by roughly 9–15 percentage points compared to all non-timing AT baselines. Equivalently, for a fixed robustness level, timing AT maintains noticeably higher clean accuracy. We will reference this table in the main paper to make the advantage of timing-only AT over standard AT explicit.

## G ON THE OPTIMALITY AND COMPLEXITY OF THE ATTACK SOLVER

This section clarifies what can and cannot be claimed about the optimality of our attack-finding formulation in Eq. (6), and discusses its computational complexity in comparison to non-timing attacks on event data such as SpikeFool (Büchel et al., 2022) and PDSG-SDA (Lun et al., 2025).

### G.1 OPTIMALITY OF THE INNER MAXIMIZATION

Eq. (6) defines an inner maximization over *discrete*, budget-constrained retimings under capacity-1 along each event line. Each non-zero packet $x[s, j]$ can be shifted by an integer offset $u$ so that it lands at $t = s + u$ on the *same* event line $j$. The feasible set is defined by (i) a capacity-1 constraint per $(t, j)$ and (ii) global timing budgets (e.g., $B_\infty, B_1, B_0$). Together with the non-convex SNN loss under BPTT, this yields a combinatorial, non-convex optimization problem.

Under these constraints, global optimality of Eq. (6) is *not* tractable: even for standard ANNs, widely used attacks such as PGD and DeepFool do not provide global optimality guarantees; similarly, event-based attacks such as SpikeFool and PDSG-SDA focus on strong, principled approximations rather than exact solutions to a discrete global optimum. Our method follows the same philosophy.

What we provide is:

- a *structured threat model* (timing-only, rate-preserving, capacity-1) and an explicit discrete feasible set of retimings; and

- a *projected-in-the-loop* (PIL) optimization scheme that is designed to reduce the gap between a soft relaxation and the discrete, budget-constrained problem.

Within the PIL framework, three components are particularly important:

- **Capacity regularizer (Eq. (10)).** This term penalizes "over-booking" in the expected occupancy of the soft retiming $S_\pi(x)$ when multiple packets try to land in the same bin $(t, j)$. It encourages the shift distribution $\pi$ to concentrate on patterns that are close to capacity-1, so that the strict projection $P^*(x, \pi, \mathcal{B}_p)$ resolves fewer conflicts and the final discrete assignment remains close to what the soft surrogate already optimized.

- **Budget-aware penalties (Eq. (12)).** These regularizers penalize soft jitter, total delay, and tamper count in expectation, so that the probabilities $\pi$ already respect the same budgets $B_\infty, B_1, B_0$ that $P^*$ enforces exactly. This aligns the *soft* search space with the *hard* budget constraints, reducing the mismatch between the relaxed problem and the true constrained objective.

- **PIL loss coupling.** In each update, the task loss is evaluated on the strictly projected input $P^*(x, \pi, \mathcal{B}_p)$, while gradients flow through the soft surrogate $S_\pi(x)$ (together with the capacity and budget penalties). This is analogous in spirit to straight-through optimization for discrete variables, but tailored to our structured retiming and budgets. It ensures that we are always *optimizing what we evaluate*: the gradient signal is shaped to favor retimings that survive projection and remain effective under the exact constraints.

In summary, we do not claim global optimality for Eq. (6), which would be unrealistic given the combinatorial, non-convex nature of the problem. Instead, our claim is that the capacity regularizer, budget-aware penalties, and PIL coupling are explicitly designed to *tighten* the relaxation–projection gap and to yield strong local optima for the *true* constrained problem. This is supported empirically in the ablation study (Sec. 5.1), where removing either the capacity regularizer or the budget-aware terms leads to noticeably weaker attacks and less stable behavior (e.g., more failed attacks at the same budgets and larger mismatches between nominal and realized budgets).

### G.2 COMPUTATIONAL COMPLEXITY AND COMPARISON TO NON-TIMING ATTACKS

Our spike-retiming attack is a gradient-based iterative method under a structured threat model. The per-iteration cost consists of three parts:

1. **SNN forward and backward (BPTT).** We run one forward and one backward pass of the SNN on the current retimed input. This is the dominant cost shared with other white-box SNN attacks.

2. **Soft retiming and regularizers.** Computing $S_{\boldsymbol{\pi}}(x)$ and the capacity / budget penalties scales with the number of candidate shifts. Let $N_{\mathrm{pkt}}$ be the number of non-zero packets (events) in $x$, and let $U_{\max}$ be the maximum number of allowed shifts per packet (the size of the local shift set $\mathcal{U}_{s,j}$). Then the soft operator and regularizers require

$$O\big(N_{\mathrm{pkt}} \cdot U_{\max}\big)$$

operations per iteration. On event-driven benchmarks, $N_{\mathrm{pkt}} \ll T \cdot H \cdot W$, so this cost scales linearly with the *sparse* event count.

3. **Strict projection $P^*(x, \boldsymbol{\pi}, \mathcal{B}_p)$.** The greedy assignment described in Appendix E operates over the same candidate set and also scales as $O(N_{\mathrm{pkt}} \cdot U_{\max})$ per iteration, with small constant factors compared to BPTT.

Overall, a single iteration consists of one SNN forward–backward pass plus an additional term that is linear in the number of active packets and local shifts. In practice, the SNN forward–backward dominates wall-clock time; the overhead of $S_{\boldsymbol{\pi}}$ and $P^*$ is modest because it exploits event sparsity.

**Comparison to non-timing event attacks.** Non-timing attacks on event data, such as Spike-Fool (Büchel et al., 2022) and PDSG-SDA (Lun et al., 2025), operate under different perturbation models (adding, deleting, or changing events in dynamic images) and use different relaxation strategies, but share the same high-level pattern:

- SpikeFool relaxes dynamic images to continuous values, computes gradients, and iteratively solves a linearized perturbation problem with rounding back to spike grids and straight-through gradients. The complexity is dominated by SNN forward–backward plus operations over (a large subset of) the dynamic image voxels.

- PDSG-SDA introduces potential-dependent surrogate gradients and a sparse dynamic attack that iteratively adds and removes spikes in dynamic images. The attack propagates gradients over all binary dynamic-image voxels and maintains sparse masks whose size scales with the grid and attack radius.

In terms of *asymptotic* complexity per iteration, all these methods, including ours, are dominated by the SNN forward–backward cost. The main difference lies in how they traverse the input space:

- Non-timing attacks on event grids typically treat a large number of voxels (time–pixel positions) as potential perturbation locations, so their perturbation-update loops scale with the grid size or with a large candidate subset.

- Our timing-only attack never changes intensities or counts; it only retimes existing packets. The update loops scale with the number of non-zero packets and their local shift windows, which is often much smaller than the full grid size on event-sparse benchmarks.

We therefore view our method as being *comparable* in big-$O$ terms to other gradient-based event attacks, while leveraging event sparsity and a structured timing-only threat model to avoid manipulating dense grids or solving additional global subproblems beyond BPTT.

## H  ADDITIONAL RESULTS ON MULTI-NORM TIMING AT AND TRADES

This section provides additional experiments on (i) combining timing-based attacks with different budgets into a single *multi-norm* adversarial training objective, and (ii) replacing the Madry-style formulation (Madry et al., 2018) with TRADES (Zhang et al., 2019) in our timing-only adversarial training. For (i) we follow the multi-perturbation formulation of Maini et al. (Maini et al., 2020), and for (ii) we instantiate TRADES with our timing-only inner maximizer.

Table 8: Single-norm timing AT vs. multi-norm timing AT on DVS-Gesture (VGGSNN), reported as ASR (%, lower is better). "Binary" and "Integer" refer to the event-grid representation.

| Grid | Training | clean | $B_\infty{=}1$ | 2 | 3 | $B_1{=}2k$ | 4k | 8k | $B_0{=}1k$ | 2k | 4k | Avg ASR |
|---|---|---|---|---|---|---|---|---|---|---|---|---|
| Binary | Single-norm $B_\infty(1)$ | 22.92 | 57.59 | 71.20 | 68.19 | 31.81 | 45.46 | 60.60 | 15.14 | 27.27 | 54.54 | 47.98 |
| Binary | Single-norm $B_1(8000)$ | 48.26 | 80.58 | 92.08 | 94.96 | 62.60 | 80.58 | 87.05 | 42.46 | 64.75 | 87.05 | 76.90 |
| Binary | Single-norm $B_0(4000)$ | 22.57 | 73.86 | 84.63 | 89.23 | 50.78 | 58.44 | 81.52 | 35.40 | 46.17 | 75.37 | 66.15 |
| Binary | MultiNorm-Avg | 31.60 | 71.43 | 65.91 | 64.77 | 51.14 | 59.09 | 72.73 | 42.86 | 45.45 | 65.91 | 59.92 |
| Binary | MultiNorm-Max | 36.11 | 79.81 | 78.85 | 77.88 | 64.42 | 77.88 | 81.73 | 51.92 | 71.15 | 77.88 | 73.50 |
| Integer | Single-norm $B_\infty(1)$ | 52.08 | 48.00 | 47.33 | 42.67 | 14.00 | 16.67 | 33.33 | 10.00 | 15.99 | 24.67 | 28.07 |
| Integer | Single-norm $B_1(8000)$ | 68.75 | 40.92 | 40.39 | 36.87 | 6.07 | 10.09 | 20.71 | 0.51 | 6.07 | 19.19 | **20.09** |
| Integer | Single-norm $B_0(4000)$ | 72.22 | 43.27 | 47.12 | 48.08 | 7.21 | 16.35 | 28.86 | 2.40 | 10.58 | 28.86 | 25.86 |
| Integer | MultiNorm-Avg | 37.50 | 28.70 | 35.19 | 40.74 | 10.19 | 10.19 | 21.30 | 8.33 | 13.89 | 23.15 | 21.30 |
| Integer | MultiNorm-Max | 15.97 | 54.35 | 56.52 | 54.35 | 6.52 | 8.70 | 26.09 | 6.52 | 10.87 | 26.09 | 27.78 |

## H.1 MULTI-NORM TIMING AT IN THE STYLE OF MAINI ET AL.

Our three timing attacks live in different combinatorial budget spaces: the per-spike jitter radius $\mathcal{B}_\infty$, the total latency budget $\mathcal{B}_1$, and the tamper-count budget $\mathcal{B}_0$. Inspired by the multi-perturbation setup of Maini et al. (2020), we define two multi-norm timing objectives:

$$\mathcal{L}_{\text{avg}}(x,y) = \frac{1}{3} \sum_{p \in \{\infty, 1, 0\}} \max_{\Delta \in \mathcal{B}_p} \ell\big(f_\theta(P^*(x, \pi_p, \mathcal{B}_p)), y\big), \tag{33}$$

$$\mathcal{L}_{\text{max}}(x,y) = \max_{p \in \{\infty, 1, 0\}} \max_{\Delta \in \mathcal{B}_p} \ell\big(f_\theta(P^*(x, \pi_p, \mathcal{B}_p)), y\big), \tag{34}$$

where for each $p \in \{\infty, 1, 0\}$ the inner maximizer is our timing-only attack with the *training* budgets $B_\infty(1)$, $B_1(8000)$, and $B_0(4000)$, and $P^*$ is the strict, feasible projection from the main paper.

We train these variants on DVS-Gesture with VGGSNN for both binary and integer event grids. We report attack success rate (ASR, in %) measured only on samples correctly classified by the clean model. For the single-norm rows, we reuse the same trained models as in Table 4 of the main paper and re-express their robustness in terms of ASR for comparability.

On the integer grid, the best single-norm configuration $B_1(8000)$ attains clean accuracy 68.75% with the lowest average ASR 20.09%. The MultiNorm-Avg objective achieves a comparable Avg ASR (21.30%) but its clean accuracy collapses to 37.50%, while MultiNorm-Max further reduces clean accuracy to 15.97% and *increases* the Avg ASR to 27.78%. On the binary grid, both multi-norm variants have high Avg ASR ($\approx$ 60–74%) despite moderate clean accuracy, and are dominated by the best single-norm configurations.

In contrast to the image-space setting of Maini et al. (2020), where all norms share the same continuous pixel space, our $\mathcal{B}_\infty$, $\mathcal{B}_1$, and $\mathcal{B}_0$ act on different combinatorial timing budgets. When merged into a single inner maximization, the strongest budgeted component tends to dominate, producing overly aggressive timing perturbations that harm clean accuracy much more than they improve robustness. This supports our choice to keep single-norm timing AT as the main recipe in the paper, and to report multi-norm variants only as complementary evidence.

## H.2 TRADES VS. MADRY-STYLE TIMING AT ON BINARY GRIDS

We also investigate replacing the Madry-style adversarial training (Madry et al., 2018) with TRADES (Zhang et al., 2019) on the binary DVS-Gesture grid. We use the TRADES objective

$$\mathcal{L}_{\text{TRADES}} = \ell\big(f_\theta(x), y\big) + \beta \cdot \text{KL}\big(f_\theta(x) \,\|\, f_\theta(x_{\text{adv}})\big), \tag{35}$$

and instantiate the inner maximizer $x_{\text{adv}}$ with our timing-only attack, using the same three training budgets as Table 4: $B_\infty(1)$, $B_1(8000)$, and $B_1(4000)$. We sweep the trade-off parameter $\beta \in \{0.01, 0.1, 6.0\}$.

Table 9 reports clean accuracy and ASR (in %) for three evaluation budgets ($B_\infty(1)$, $B_1(4000)$, $B_0(2000)$), comparing TRADES runs to PGD-style timing AT.

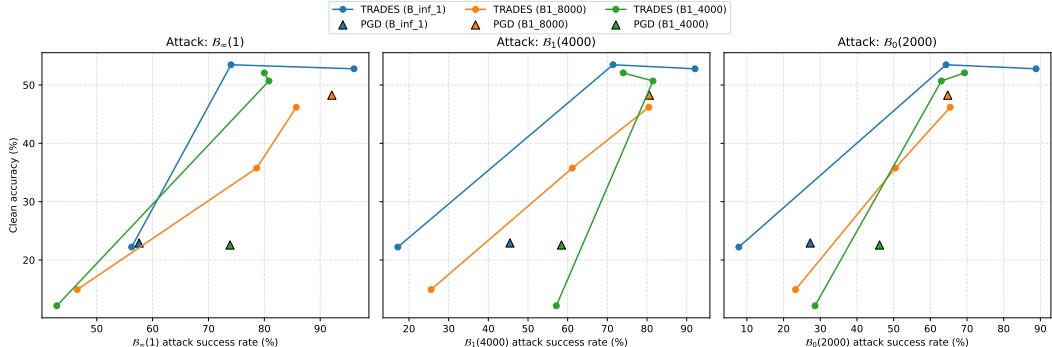

Figure 4: Results of TRADES V.s. PGD AT.

Table 9: TRADES vs. PGD-style timing AT on binary DVS-Gesture. Clean accuracy and ASR (%, lower is better) for three evaluation budgets.

| Training scheme | clean | $B_\infty=1$ | $B_1=4000$ | $B_0=2000$ |
|---|---|---|---|---|
| TRADES $\beta=0.01 + B_\infty(1)$ AT | 52.78 | 96.05 | 92.11 | 88.82 |
| TRADES $\beta=0.01 + B_1(8000)$ AT | 46.18 | 85.71 | 80.45 | 65.41 |
| TRADES $\beta=0.01 + B_1(4000)$ AT | 52.08 | 80.00 | 74.00 | 69.33 |
| TRADES $\beta=0.1 + B_\infty(1)$ AT | 53.47 | 74.03 | 71.43 | 64.29 |
| TRADES $\beta=0.1 + B_1(8000)$ AT | 35.76 | 78.64 | 61.17 | 50.49 |
| TRADES $\beta=0.1 + B_1(4000)$ AT | 50.69 | 80.82 | 81.51 | 63.01 |
| TRADES $\beta=6.0 + B_\infty(1)$ AT | 22.22 | 56.25 | 17.19 | 7.81 |
| TRADES $\beta=6.0 + B_1(8000)$ AT | 14.93 | 46.51 | 25.58 | 23.26 |
| TRADES $\beta=6.0 + B_1(4000)$ AT | 12.15 | 42.86 | 57.14 | 28.57 |
| Our timing AT + $B_\infty(1)$ | 22.92 | 57.59 | 45.46 | 27.26 |
| Our timing AT + $B_1(8000)$ | 48.26 | 92.08 | 80.58 | 64.75 |
| Our timing AT + $B_1(4000)$ | 22.57 | 73.86 | 58.44 | 46.17 |

To visualize the clean–robust trade-offs, Figure 4 in the supplementary plots clean accuracy against ASR for each evaluation budget, with TRADES configurations forming curves for fixed inner budgets and varying $\beta$, and the corresponding PGD timing-AT points marked as triangles. For several operating points, TRADES achieves lower ASR than PGD at similar clean accuracy (points above-and-to-the-left of the PGD baselines), confirming that the TRADES regularization can slightly sharpen the timing-based robustness vs. accuracy trade-off. However, large $\beta$ values also illustrate the usual TRADES behavior: further reductions in ASR come at the cost of substantial clean-accuracy degradation.

Overall, these experiments show that (i) multi-norm timing AT, when naively aggregating $B_\infty$, $B_1$, and $B_0$ in the style of Maini et al. (2020), offers no clear advantage over carefully tuned single-norm timing AT in our discrete retiming setting, and (ii) TRADES provides modest improvements over Madry-style timing AT on binary grids, but does not fundamentally change the conclusion that strong timing-only adversaries remain hard to defend against without incurring noticeable drops in clean performance.

# I   MULTI-MODEL AND MULTI-NORM MULTI-MODEL TIMING ATTACKS

For completeness, we also evaluate *multi-model* and *multi-norm multi-model* timing attacks on event-driven SNNs, following the ensemble idea of timing attacks over multiple victim models similar in spirit to ensemble attacks for SNNs (Xu et al., 2025). In all experiments below, the perturbation is still a capacity-1 spike-retiming with strict rate preservation enforced by $P^*$.

## I.1   MULTI-MODEL TIMING ATTACKS ON N-MNIST

**Methodology.**   On N-MNIST, we construct an ensemble of the six models used in our transfer experiments: three SNNs (ConvNet (SNN), ResNet18 (SNN), VGGSNN (SNN)) and three CNN

Table 10: Multi-model timing attacks on binary N-MNIST (ASR, %).

| Model | Clean | $B_\infty$:1 | $B_\infty$:2 | $B_\infty$:3 | $B_1$:500 | $B_1$:750 | $B_1$:1000 | $B_0$:200 | $B_0$:300 | $B_0$:400 |
|---|---|---|---|---|---|---|---|---|---|---|
| ConvNet (SNN) | 99.06 | 100.0 | 100.0 | 100.0 | 50.9 | 97.6 | 100.0 | 35.0 | 93.4 | 100.0 |
| ConvNet (CNN) | 99.34 | 100.0 | 100.0 | 100.0 | 92.6 | 99.8 | 100.0 | 42.8 | 97.6 | 99.9 |
| ResNet18 (SNN) | 99.62 | 97.7 | 100.0 | 100.0 | 28.2 | 70.6 | 96.2 | 29.7 | 89.3 | 99.9 |
| ResNet18 (CNN) | 99.70 | 100.0 | 100.0 | 100.0 | 96.5 | 99.2 | 99.9 | 34.7 | 90.8 | 98.3 |
| VGGSNN (SNN) | 99.64 | 100.0 | 100.0 | 100.0 | 32.3 | 82.1 | 99.5 | 25.6 | 86.3 | 100.0 |
| VGG (CNN) | 99.72 | 100.0 | 100.0 | 100.0 | 97.1 | 99.6 | 99.7 | 31.8 | 90.0 | 98.6 |

Table 11: Multi-model timing attacks on integer N-MNIST (ASR, %).

| Model | Clean | $B_\infty$:1 | $B_\infty$:2 | $B_\infty$:3 | $B_1$:500 | $B_1$:750 | $B_1$:1000 | $B_0$:200 | $B_0$:300 | $B_0$:400 | $B_0$:600 |
|---|---|---|---|---|---|---|---|---|---|---|---|
| ConvNet (SNN) | 99.19 | 100.0 | 100.0 | 100.0 | 46.6 | 96.4 | 100.0 | 40.2 | 99.2 | 100.0 | 100.0 |
| ConvNet (CNN) | 99.38 | 100.0 | 100.0 | 100.0 | 98.2 | 100.0 | 100.0 | 59.8 | 100.0 | 100.0 | 100.0 |
| ResNet18 (SNN) | 99.62 | 91.5 | 100.0 | 100.0 | 31.6 | 75.8 | 98.0 | 27.3 | 94.5 | 100.0 | 99.8 |
| ResNet18 (CNN) | 99.73 | 98.9 | 100.0 | 100.0 | 73.1 | 66.7 | 54.7 | 81.9 | 95.6 | 100.0 | 100.0 |
| VGGSNN (SNN) | 99.71 | 99.8 | 100.0 | 100.0 | 42.1 | 89.3 | 99.8 | 29.9 | 97.6 | 100.0 | 100.0 |
| VGG (CNN) | 99.79 | 100.0 | 100.0 | 100.0 | 85.0 | 90.7 | 90.0 | 98.4 | 97.0 | 100.0 | 100.0 |

counterparts (ConvNet (CNN), ResNet18 (CNN), VGG (CNN)). For a given timing budget $\mathcal{B}_p$ and norm $p \in \{\infty, 1, 0\}$, we optimize

$$\mathcal{L}_{\mathrm{mm}}(x, y) = \frac{1}{M} \sum_{m=1}^{M} \ell\big(f_\theta^{(m)}(P^*(x, \pi^{(m)}, \mathcal{B}_p)), y\big), \quad M = 6, \tag{36}$$

where $f_\theta^{(m)}$ are the six models and $P^*$ is our strict rate-preserving projection. Budgets on N-MNIST are $B_\infty \in \{1, 2, 3\}$ (per-spike jitter), $B_1 \in \{500, 750, 1000\}$ (total delay), $B_0 \in \{200, 300, 400\}$ (binary) plus $B_0(600)$ (integer). We report clean accuracy ("Clean") and attack success rate (ASR, %) on clean-correct examples.

Compared with the single-model attacks in Tables 1–2 of the main paper, these ensemble attacks remain very strong: under moderate $B_1$ or $B_0$ budgets, ASR on SNNs routinely exceeds 70–90%, confirming that our timing-only adversary transfers across architectures even when optimized jointly over six models. CNN counterparts are also vulnerable, but the relative gap between SNNs and CNNs is consistent with our main message: timing perturbations exploit SNN temporal dynamics more effectively than frame-based CNNs.

## I.2 MULTI-NORM MULTI-MODEL TIMING ATTACKS

**Definition of multi-norm timing budgets.** In our setting, a multi-norm timing attack means that every adversarial example simultaneously satisfies: (i) a local jitter radius $B_\infty$ (per-spike timing change), (ii) a total delay budget $B_1$ (sum of absolute delays), and (iii) a tamper-count budget $B_0$ (number of moved spikes). We keep the same $B_\infty$-style loss over shift logits $\pi$, but in the strict projection we enforce *all three* budgets:

$$x' = P^*\big(x, \pi; \mathcal{B}_\infty, \mathcal{B}_1, \mathcal{B}_0\big), \tag{37}$$

so that the final retimed events lie in the intersection $\mathcal{B}_\infty \cap \mathcal{B}_1 \cap \mathcal{B}_0$ under capacity-1 and rate-preserving constraints.

**Multi-norm, multi-model ensemble.** On N-MNIST, we instantiate a multi-norm, multi-model attack over the same $M{=}6$-model ensemble (three SNNs + three CNNs):

$$\mathcal{L}_{\mathrm{joint}}(x, y) = \frac{1}{M} \sum_{m=1}^{M} \ell\Big(f_\theta^{(m)}\big(P^*(x, \pi^{(m)}; \mathcal{B}_\infty, \mathcal{B}_1, \mathcal{B}_0)\big), y\Big). \tag{38}$$

We choose a balanced triple of budgets $\mathcal{B}_\infty = B_\infty(3)$, $\mathcal{B}_1 = B_1(750)$, $\mathcal{B}_0 = B_0(300)$, so that maximal jitter, total latency, and tamper count remain comparable to the single-norm experiments.

Table 12: ASR (%) of multi-norm multi-model timing attacks on N-MNIST.

| Model | Binary: MultiNorm $B_\infty(3), B_1(750), B_0(300)$ | Integer: MultiNorm $B_\infty(3), B_1(750), B_0(300)$ |
|---|---|---|
| ConvNet (SNN) | 93.6 | 96.0 |
| ConvNet (CNN) | 94.0 | 99.8 |
| ResNet18 (SNN) | 93.0 | 92.1 |
| ResNet18 (CNN) | 71.6 | 44.3 |
| VGGSNN (SNN) | 91.9 | 94.6 |
| VGG (CNN) | 75.6 | 71.5 |

**Results.** Table 12 reports ASR for this joint attack. The joint multi-norm, multi-model attack is extremely strong: ASR on all three SNNs exceeds $90\%$ in both binary and integer grids, even though each adversarial example respects *all three* timing budgets simultaneously. CNN counterparts are also highly vulnerable (e.g., ConvNet (CNN) above $94\%$ ASR), but the relative SNN–CNN gap is consistent with our single-norm and multi-model results. Compared to the single-norm ensemble attacks in Tables 10–11, the multi-norm version mainly serves as a more pessimistic "all-budgets-on" stress test: it does not reveal qualitatively new behavior, while the single-norm budgets $B_\infty$, $B_1$, and $B_0$ remain more interpretable for realistic threat models (e.g., "small jitter only" or "limited latency only").

## J ON $L_0$ BUDGETS, REDUNDANT MOVES, AND RELATION TO PIXEL-WISE SPARSITY

Our threat model is timing-only and rate-preserving, so all three budgets are defined at the *spike/packet* level rather than per-pixel on dynamic images.

**What the budgets measure.** After flattening $[B, C, H, W]$ into event lines $j \in \{1, \dots, N\}$, each non-zero entry $x[s, j]$ at time index $s \in \{1, \dots, T\}$ denotes an *event packet* (one spike for binary grids, or an integer-valued packet for integer grids). For a retimed example with shifts $\Delta_{s,j} = t - s$, the three budgets are

$$B_\infty: \max_{(s,j):\Delta_{s,j} \neq 0} |\Delta_{s,j}|, \tag{39}$$

$$B_1: \sum_{(s,j):\Delta_{s,j} \neq 0} |\Delta_{s,j}|, \tag{40}$$

$$B_0: \#\{(s, j): x[s, j] > 0, \ \Delta_{s,j} \neq 0\}. \tag{41}$$

Thus $B_0$ counts how many individual spikes are retimed ("tamper count"), while $B_\infty$ and $B_1$ bound how far in time these spikes are moved. This is a sensor-level notion of sparsity tailored to timing perturbations on SNNs.

**Why symmetric "swaps" are ruled out by projection.** A toy example is swapping two active indices at the same spatial position, e.g., $(1, 0, 1, 1)$ and $(2, 0, 1, 1)$ in $TCHW$, which could leave some frame-based summary unchanged while incurring $B_0 = 2$. Two points are important here:

- From the SNN's perspective, changing spike times is *not* neutral: membrane integration and firing decisions depend on the exact timing, so "swapping" spikes across time steps is in general a genuine perturbation.
- More importantly, our strict projection $P^*(x, \pi, \mathcal{B}_p)$ is explicitly designed so that such symmetric swaps are not selected. For each flattened line $j$ and time $t$, we maintain $\text{reserved}[j, t]$ (initially 1 wherever $x[t, j] > 0$) and $\text{occupied}[j, t]$ (targets already taken by moved packets). When processing a candidate move $(s \to t, j)$, we *skip* it if $\text{moved}[j, s]$ is true, or $\text{occupied}[j, t]$ is true, or $\text{reserved}[j, t]$ is still true. A packet can only move into originally empty bins, or into bins whose original packet has already moved out and released its reservation. As a result, two spikes at the same spatial position cannot simply exchange times: moving into a bin that still hosts its original spike is blocked by $\text{reserved}[j, t]$ and capacity–1.

Therefore, the $B_0$ budget is not spent on symmetric swaps that leave the timeline effectively unchanged; it is spent on injective retimings into genuinely free time slots.

**Optimization does not "waste" $B_0$.** Beyond the projection logic, the optimization itself discourages budget waste. Shift probabilities $\pi[s, j, t]$ are updated to maximize the task loss evaluated on $P^*(x, \pi, \mathcal{B}_p)$; candidate moves that do not meaningfully increase the loss receive vanishing gradients and their probabilities shrink relative to more damaging moves. The capacity regularizer and budget-aware penalties further push $\pi$ toward configurations that respect capacity–1 and budgets already in expectation. In practice, we often observe that the realized number of moved packets is *below* the nominal upper bound $B_0$, and ablations removing these regularizers yield weaker attacks and less stable budget usage.

**Relation to pixel-wise $\ell_0$ in prior DVS attacks.** Prior sparse DVS attacks typically define $\ell_0$ at the pixel/voxel level on dynamic images. Our packet-level $B_0$ can be related to this pixel-wise sparsity as follows: each moved packet at $(s, j)$ with value $v > 0$ changes at most two grid voxels $(t, j)$—one at the source time (set from $v$ to $0$) and one at the target time (set from $0$ to $v$). Hence the number of changed grid voxels satisfies

$$\|\mathbf{x}' - \mathbf{x}\|_0 \leq 2B_0. \tag{42}$$

In the supplementary experiments, we therefore also report the number of changed $(t, x, y)$ positions in our adversarial examples and compare this pixel-wise $\ell_0$ to that of prior sparse DVS attacks under matching datasets and models.

Table 13: Binary N-MNIST: ASR (%) under tamper-count budgets $B_0$.

| Method | Model | $B_0$=200 | $B_0$=300 | $B_0$=400 |
|---|---|---|---|---|
| Ours (timing) | ConvNet | 13.0 | 53.1 | 98.5 |
| | ResNet18 | 78.9 | 100.0 | 100.0 |
| | VGGSNN | 18.3 | 81.8 | 99.8 |
| SpikeFool Büchel et al. (2022) | ConvNet | 50.4 | 92.2 | 98.9 |
| | ResNet18 | 12.5 | 37.1 | 69.7 |
| | VGGSNN | 16.2 | 38.7 | 69.5 |
| PDSG-SDA Lun et al. (2025) | ConvNet | 64.7 | 93.4 | 97.4 |
| | ResNet18 | 62.1 | 92.2 | 98.8 |
| | VGGSNN | 13.9 | 38.7 | 72.4 |

Table 14: Binary DVS-Gesture: ASR (%) under tamper-count budgets $B_0$.

| Method | Model | $B_0$=1000 | $B_0$=2000 | $B_0$=4000 |
|---|---|---|---|---|
| Ours (timing) | ResNet18 | 27.7 | 67.9 | 98.5 |
| | VGGSNN | 55.8 | 87.2 | 98.9 |
| SpikeFool Büchel et al. (2022) | ResNet18 | 17.8 | 34.3 | 72.9 |
| | VGGSNN | 14.9 | 18.9 | 25.5 |
| PDSG-SDA Lun et al. (2025) | ResNet18 | 60.2 | 82.4 | 91.2 |
| | VGGSNN | 52.9 | 67.1 | 85.0 |

## K  COMPARISON WITH RAW-EVENT BASELINES

This section compares our timing-only, rate-preserving attack with two strong raw-event baselines: SpikeFool Büchel et al. (2022) and PDSG-SDA Lun et al. (2025). All methods are evaluated on the *same* clean models and datasets as in Tables 1–2 of the main paper (N-MNIST, DVS-Gesture, CIFAR10-DVS). We always report attack success rate (ASR, %) on clean-correct examples.

**Unified tamper-count budget.**  For a fair comparison, we express sparsity using a common tamper-count budget $B_0$:

- For our timing-only attack, $B_0$ is the number of event packets $(s, j)$ whose time index changes (tamper count).
- For SpikeFool and PDSG-SDA, $B_0$ is the number of event bins $(t, j)$ whose value changes (a spike is added, removed, or its integer value is modified).

SpikeFool and PDSG-SDA operate on binary event grids by design. On integer grids, we adapt PDSG-SDA by allowing inserted spikes to take the mean event value of the dataset, which preserves the method's behavior.

### K.1  BINARY N-MNIST: OURS VS. SPIKEFOOL AND PDSG-SDA

Table 13 reports ASR at $B_0 \in \{200, 300, 400\}$ for the three models in Table 1.

On ResNet18 our timing-only attack is very strong (ASR 78.9% at $B_0$=200 and 100% at $B_0 \geq 300$), closely matching or exceeding PDSG-SDA. On ConvNet and VGGSNN, SpikeFool and PDSG-SDA are stronger at moderate $B_0$, which is expected because they can freely add and delete spikes without preserving rate or per-spike jitter. Importantly, our attack achieves these ASR values while remaining rate-preserving, constrained by $B_\infty$ and $B_1$, and enforcing capacity 1.

### K.2  BINARY DVS-GESTURE: OURS VS. SPIKEFOOL AND PDSG-SDA

We next compare ResNet18 and VGGSNN on DVS-Gesture for $B_0 \in \{1000, 2000, 4000\}$; see Table 14.

Table 15: Binary CIFAR10-DVS (ResNet18): ASR (%) under tamper-count budgets $B_0$.

| Method | $B_0{=}1000$ | $B_0{=}2000$ | $B_0{=}4000$ |
|---|---|---|---|
| Ours (timing) | 26.0 | 42.0 | 80.0 |
| SpikeFool Büchel et al. (2022) | 55.0 | 83.0 | 93.0 |
| PDSG-SDA Lun et al. (2025) | 62.0 | 91.0 | 100.0 |

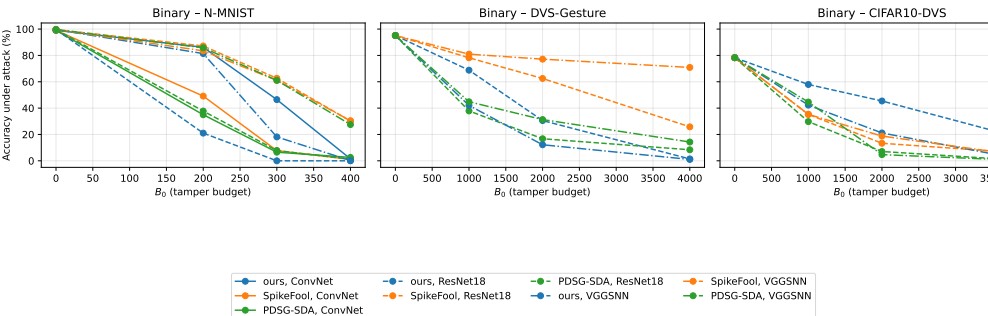

Figure 5: Accuracy V.s. budget curve on binary grid.

On VGGSNN our timing-only attack is the strongest across all $B_0$ budgets, despite its stricter constraints. On ResNet18, PDSG-SDA is stronger at small and medium $B_0$, but our attack still reaches high ASR (67.9% at $B_0{=}2000$ and 98.5% at $B_0{=}4000$), clearly outperforming SpikeFool. Again, PDSG-SDA enjoys a larger perturbation space (free insertions/deletions and value changes), while we only retime existing spikes under $B_\infty, B_1, B_0$ and capacity 1.

### K.3 BINARY CIFAR10-DVS: OURS VS. BASELINES

For CIFAR10-DVS, we show ResNet18; VGGSNN exhibits similar trends and is included in the extended tables.

On CIFAR10-DVS the raw-event baselines have a clear advantage at small budgets, as they can exploit many more degrees of freedom by adding and removing spikes. Even so, our timing-only attack attains 80% ASR at $B_0{=}4000$, showing that constrained timing perturbations alone can be very harmful.

### K.4 INTEGER GRIDS

On integer N-MNIST and CIFAR10-DVS, we compare our timing-only attack to the adapted PDSG-SDA. For example, on integer N-MNIST / ResNet18: PDSG-SDA achieves ASR $48.3\%, 86.1\%, 97.3\%, 99.9\%$ at $B_0 \in \{200, 300, 400, 600\}$, while our timing-only attack reaches $86.1\%, 99.8\%, 100\%, 100\%$ at the same budgets. Thus, on integer grids our method is often *stronger* than PDSG-SDA even though we maintain rate preservation and timing constraints. Full integer tables for all models are included in the extended supplement.

### K.5 ACCURACY–VS.–BUDGET CURVES

Following the accuracy–vs.–constraint suggestion, we also provide accuracy–vs.–$B_0$ curves for all datasets and models, as shown in Figure 5 and Figure 6.

For each grid type (binary / integer), we plot three subplots (N-MNIST, DVS-Gesture, CIFAR10-DVS), with tamper-count budget $B_0$ on the $x$-axis (including $B_0{=}0$) and accuracy under attack on the $y$-axis. The point at $B_0{=}0$ corresponds to clean accuracy; growing $B_0$ moves along the constraint axis. Our timing-only attack, SpikeFool, and PDSG-SDA appear as separate curves, and different architectures (ConvNet, ResNet18, VGGSNN) are distinguished by line style.

On binary grids, our curves often lie on par with or below those of SpikeFool and PDSG-SDA on N-MNIST and DVS-Gesture, indicating equal or higher destructive power under the same $B_0$, despite

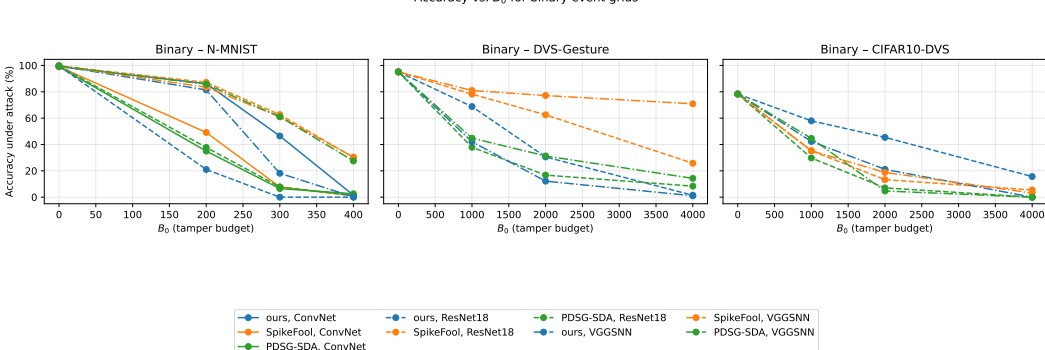

Figure 6: Accuracy V.s. budget curve on integer grid.

our stricter constraints. On CIFAR10-DVS, value-modifying baselines become slightly stronger at large $B_0$, but the gap remains moderate. On integer grids, our accuracy generally drops as fast as, or faster than, PDSG-SDA, showing that a rate-preserving timing attack can be as destructive as a value-modifying baseline.

Overall, these comparisons provide the requested anchor: our timing-only attack is competitive with, and often stronger than, leading raw-event attacks across most datasets and budgets, while operating under a much stricter and physically motivated constraint set (rate preservation, per-spike jitter, total delay, and capacity 1).

## L  EFFECT OF SIMPLE EVENT FILTERING DEFENSES

Real event–camera pipelines often include low–level filtering to suppress jitter and noisy events. To test whether such simple pre–processing can remove our *timing–only, rate–preserving* perturbations, we evaluate three concrete, label–free defenses on the binary DVS-Gesture / VGGSNN configuration (one of our strongest attack settings), and compare against the non-timing baseline PDSG-SDA Lun et al. (2025).

**Defenses.**  All defenses operate directly on the event stream, without labels or model gradients:

- **Refractory filtering** (`refractory_first`). For each pixel, we look back over a temporal window of length `rp_bins`; if an event has occurred in that window, a new event at the same pixel is dropped with probability $p$. This mimics a sensor–level refractory mechanism that suppresses very high–frequency bursts.
- **Temporal mean smoothing** (`temporal_mean_smooth`). We convolve each sequence along the time axis with a length–3 box filter (radius 1). With probability $p$ (per sample), we replace the input by this temporally smoothed version; otherwise we keep the original. This targets short–lived temporal fluctuations and jitter.
- **Spatial mean smoothing** (`spatial_mean_smooth`). We apply a $3 \times 3$ spatial mean filter (stride 1) to each frame and, with probability $p$ per sample, replace the input by the spatially smoothed version. This targets isolated "salt–and–pepper" spikes and enforces local spatial consistency.

To expose the clean–robustness trade–off, we treat $p \in [0, 1]$ as a *defense strength knob*: $p=0$ corresponds to no defense, small $p$ to light–to–moderate denoising, and large $p$ to very strong filtering that heavily distorts the input.

**Evaluation setup.**  On binary DVS-Gesture / VGGSNN, we report: (i) clean accuracy (`clean`), and (ii) attack success rate (ASR, %) under our timing attack with budgets $B_\infty(1)$, $B_1(8000)$, $B_0(4000)$ and the non-timing PDSG-SDA Lun et al. (2025) with $B_0(4000)$. ASR is always computed on clean–correct examples.

**Discussion.**  Three trends emerge from Table 16:

Table 16: Effect of simple event filtering defenses on binary DVS-Gesture / VGGSNN. We report clean accuracy (%) and ASR (%) for our timing–only attack and PDSG-SDA Lun et al. (2025) under $B_0(4000)$, as a function of defense probability $p$.

| Defense | $p$ | clean | Ours $B_\infty(1)$ | Ours $B_1(8000)$ | Ours $B_0(4000)$ | PDSG-SDA $B_0(4000)$ |
|---|---|---|---|---|---|---|
| refractory_first | 0.0 | 95.14 | 96.7 | 98.5 | 98.5 | 100.0 |
| | 0.1 | 71.53 | 92.7 | 96.6 | 97.0 | 65.5 |
| | 0.2 | 48.96 | 84.4 | 85.8 | 87.2 | 60.2 |
| | 0.4 | 27.08 | 70.5 | 70.5 | 71.7 | 57.6 |
| | 0.6 | 15.28 | 47.7 | 47.7 | 47.7 | 43.1 |
| | 0.8 | 14.93 | 46.5 | 46.5 | 46.5 | 41.8 |
| | 1.0 | 16.04 | 40.6 | 44.0 | 48.0 | 33.3 |
| temporal_mean_smooth | 0.0 | 95.14 | 96.7 | 98.5 | 98.5 | 100.0 |
| | 0.1 | 84.03 | 95.8 | 95.0 | 97.1 | 35.5 |
| | 0.2 | 72.57 | 95.6 | 92.3 | 94.7 | 33.4 |
| | 0.4 | 56.60 | 92.6 | 79.1 | 84.6 | 34.3 |
| | 0.6 | 49.65 | 86.0 | 67.8 | 79.0 | 34.2 |
| | 0.8 | 52.43 | 81.4 | 55.6 | 78.8 | 33.7 |
| | 1.0 | 56.25 | 61.7 | 38.2 | 68.5 | 30.2 |
| spatial_mean_smooth | 0.0 | 95.14 | 96.7 | 98.5 | 98.5 | 100.0 |
| | 0.1 | 71.53 | 95.6 | 93.6 | 96.1 | 38.3 |
| | 0.2 | 54.86 | 92.4 | 82.9 | 91.1 | 42.4 |
| | 0.4 | 32.64 | 59.5 | 38.3 | 59.5 | 39.3 |
| | 0.6 | 23.61 | 4.4 | 1.4 | 13.2 | 17.6 |
| | 0.8 | 21.18 | 4.9 | 3.2 | 14.7 | 14.7 |
| | 1.0 | 22.92 | 5.6 | 4.5 | 3.0 | 2.4 |

- **Moderate filtering barely dents our timing attack but already hurts clean accuracy.** For example, with `refractory_first` at $p=0.2$, clean accuracy drops from $95.14\%$ to $48.96\%$ (a $\sim 46$ point loss), yet our ASR at $B_0(4000)$ remains high at $87.2\%$. Similar patterns hold for temporal and spatial smoothing at $p=0.1$–$0.2$.

- **Very strong filtering can suppress the attack only at the cost of destroying the task.** Spatial smoothing with $p=1.0$ reduces our ASR at $B_0(4000)$ to $3.0\%$, but clean accuracy also collapses to $22.92\%$. Strong refractory or temporal smoothing show the same trade–off.

- **Our timing–only attack is at least as robust to filtering as a strong non–timing baseline.** The value–modifying PDSG-SDA is more easily attenuated by these filters: under temporal smoothing with $p \in [0.1, 0.4]$, its ASR drops from $100\%$ to roughly $30$–$35\%$, whereas our ASR at $B_0(4000)$ remains in the $80$–$97\%$ range. Under spatial smoothing with $p=0.4$, our ASR at $B_0(4000)$ is $59.5\%$ versus $39.3\%$ for PDSG-SDA.

Overall, these results support our main claim: simple intensity– or value–based event filtering is *not* sufficient to neutralize capacity–1, rate–preserving spike retiming attacks without incurring severe clean accuracy loss. This highlights spike retiming as a practically important and difficult–to–defend attack surface that calls for temporally aware defenses beyond naive denoising.

# M    EXTENDED RESULTS ON TARGETED TIMING ATTACKS

In this section, we expand on the targeted experiments and address two issues: (i) the fixed target label in the main text, and (ii) the lack of comparison to a strong non-timing baseline.

**Random-target protocol.**    To avoid bias from a fixed target (class "0"), we adopt a standard *random-target* protocol: for each clean-correct sample with ground truth label $y$, we draw a target label $\tilde{y} \neq y$ uniformly at random and optimize the attack to force the prediction to $\tilde{y}$. We repeat this procedure for 5 random seeds and report mean $\pm$ standard deviation of the *targeted* attack success rate (ASR), where success means the final prediction equals the chosen target $\tilde{y}$. As in all other

experiments, ASR is measured only on samples that are correctly classified in the clean setting, so the metric is directly comparable to our untargeted results.

**Comparison with a non-timing baseline.** To separate limitations of our method from the intrinsic difficulty of targeted attacks, we compare against the state-of-the-art non-timing raw-event attack PDSG-SDA Lun et al. (2025) on the *integer* DVS-Gesture / VGGSNN configuration, which is one of the hardest settings in our paper. We evaluate both methods under increasing tamper-count budgets $\mathcal{B}_0 \in \{4000, 8000, 12000\}$ and average over 5 random seeds.

Table 17: Targeted ASR (%, mean $\pm$ std over 5 seeds) on integer DVS-Gesture / VGGSNN under increasing tamper-count budgets $\mathcal{B}_0$.

| Attack | $\mathcal{B}_0(4000)$ | $\mathcal{B}_0(8000)$ | $\mathcal{B}_0(12000)$ |
|---|---|---|---|
| Ours (timing-only) | **15.7 $\pm$ 1.3** | **26.8 $\pm$ 1.4** | **30.8 $\pm$ 1.3** |
| PDSG-SDA Lun et al. (2025) | 24.4 $\pm$ 1.3 | 28.7 $\pm$ 1.3 | 31.4 $\pm$ 1.3 |

We further observe that when the $\mathcal{B}_0$ constraint is removed for PDSG-SDA, its targeted ASR saturates around $\sim 34\%$, even though it is allowed to freely add and delete spikes without any rate or jitter constraints. In the supplementary (Fig. 7), we plot the mean targeted ASR as a function of $\mathcal{B}_0$ for both methods, showing that targeted ASR monotonically increases with budget and that our *more constrained* timing-only attack approaches the performance of PDSG-SDA at higher budgets.

**Why targeted timing-only attacks are challenging.** The targeted ASR in Table 17 is notably lower than our untargeted ASR, especially on integer grids. This reflects an *intrinsic* challenge of the threat model rather than a flaw in the optimization: our attack is capacity-1, rate-preserving, and timing-only, so it cannot create or delete spikes and must keep at most one packet per $(t, j)$ bin. Under these constraints, steering the model toward a *specific* target class is substantially harder than simply causing misclassification. Integer event grids add further difficulty because many bins already store multi-spike packets, and retiming must respect both the global budget $\mathcal{B}_0$ and the capacity constraint.

The comparison with PDSG-SDA Lun et al. (2025) supports this view: even a much less constrained, value-modifying attack struggles to exceed $\approx 34\%$ targeted ASR on this setting (without a tamper bound). Our timing-only attack achieves comparable performance at high budgets while respecting strict timing and rate constraints, indicating that the gap between untargeted and targeted spike-retiming attacks is a fundamental phenomenon. We highlight closing this gap—for example via target-aware objectives or curriculum schedules on timing budgets—as an interesting direction for future work.

# N  DISCUSSION: ADVANTAGES OF TIMING-ONLY, RATE-PRESERVING ATTACKS

**Threat model and what the extra constraint enforces.** Our timing attack operates at the level of *event packets* on each event line. After flattening $(B, C, H, W)$ into a line index $j$, each non-zero entry $x[s, j]$ denotes a packet at time index $s$ (one spike on binary grids, or an integer-valued packet on integer grids). Under our attack, every packet can only be *retimed along its own line* by an integer offset $u$, landing at $t = s + u$ within a bounded jitter window, while the strict projection $P^*(x, \pi, \mathcal{B}_p)$ enforces: (i) *capacity-1*: at most one packet per $(t, j)$, (ii) *rate preservation*: for each line $j$, the multiset $\{x'[t, j]\}_{t=1}^T$ is identical to $\{x[t, j]\}_{t=1}^T$, and (iii) an $\ell_0$ "tamper-count" budget $\mathcal{B}_0$ on the number of packets that actually move (non-zero displacement). In other words, every packet that is "removed" from some $(s, j)$ must be added back at some $(t, j)$ on the *same* line within the budgeted jitter window: we only *reorder* events in time, never create or delete them.

**Beyond generic raw-event $\ell_0$ attacks.** Even if a generic raw-event attack is forced to preserve the global (or per-line) event count, our formulation imposes a strictly stronger and more structured constraint: (i) packets never move across pixels or polarities, (ii) capacity-1 prohibits stacking multiple packets into the same $(t, j)$ bin, and (iii) $B_\infty$ and $B_1$ bound the local jitter and total delay. This

leads to three advantages compared to value-based raw-event attacks such as SpikeFool Büchel et al. (2022) and PDSG-SDA Lun et al. (2025) under a matched $\mathcal{B}_0$:

- **Closer to physical jitter/latency.** Real DVS sensors primarily exhibit timestamp jitter and latency rather than large, structured changes in event amplitudes or polarities. Our attack keeps every packet on its original pixel and polarity, preserves per-line counts exactly, and constrains temporal motion via $B_\infty$ and $B_1$, making the perturbations resemble realistic timing noise rather than synthetic intensity artifacts.

- **Stealth against simple monitors and preprocessing.** Because the multiset of packets on each line is preserved, any monitoring that relies on per-pixel event counts, line-wise firing rates, or simple intensity statistics sees no anomaly: only the *timing* changes. In Appendix L, we explicitly tested three simple, label-free pre-processors (refractory filtering, temporal mean smoothing, spatial mean smoothing) on binary DVS-Gesture / VGGSNN and observed that moderate filtering already causes 20–40 point drops in clean accuracy, while our timing-only attack still attains $> 90\%$ ASR under $\mathcal{B}_0(4000)$. Under the same filters, the value-based baseline PDSG-SDA is noticeably more suppressed, indicating that our additional temporal constraint makes the perturbations *harder to wash out* without destroying task performance.

- **Directly aligned with SNN computation.** SNNs encode much of their information in spike *timing* rather than aggregate counts. Retiming spikes while preserving counts manipulates the temporal decision boundary in a way that leaves fewer footprints in simple rate statistics. Empirically, our main results and the raw-event baseline study in Appendix K show that, under matched tamper-count budgets $\mathcal{B}_0$, our timing-only, rate-preserving attack achieves attack success rates comparable to or higher than SpikeFool Büchel et al. (2022) and PDSG-SDA Lun et al. (2025), despite operating in a much smaller feasible perturbation set (capacity-1 + rate preservation + jitter bounds).

Taken together, these points clarify the advantage of the "remove–then–add-back" constraint: it does not merely make the attack optimization harder; it defines a threat model that is (i) closer to how real event cameras behave, (ii) more stealthy with respect to standard count- or value-based checks, and (iii) still highly effective in practice, as evidenced by our comparisons to strong raw-event baselines and our robustness-to-filtering experiments.

