# OpenReview forum: "Time Is All It Takes: Spike-Retiming Attacks on Event-Driven Spiking Neural Networks"
_ICLR.cc/2026/Conference — ICLR 2026 Poster_

### Official Review · Reviewer_thcu · 2025-10-24

**Soundness:** 1
**Presentation:** 2
**Contribution:** 3
**Rating:** 2
**Confidence:** 5

**Summary:**

This work introduces a timing-only adversarial attack on event data in the white-box, untargeted setting. It reparameterizes the discrete optimization objective as probability distributions and, in the forward pass, uses projection and constraint proxies to fit the constraints, with a final hard projection to enforce them.

**Strengths:**

- This work further investigates the spike-timing attack, which is novel to existing raw even SNN attacks
 - The author considered various constraints and formulated norm-agnostic constraints, reflecting the practical scenarios
 - The evaluation scenarios cover untargeted/targeted attack, robust model, black-box transfer attack with ablation on different neuron types, time bin number $T$, and constraint type $\mathcal{B}\_{\infty}, \mathcal{B}\_{1}(8k),\mathcal{B}\_{0}(4k)$

**Weaknesses:**

I appreciate the effort the authors have put into this paper, but some crucial concerns bring it slightly below the acceptance threshold.
1. *Lack of baseline methods.* Throughout the main paper and the appendix, there is no comparison of *any* baseline method. Without contrast, the method's performance lacks an anchor point. Moreover, the paper already cites two methods for the raw even attack. The authors are encouraged to appropriately compare the proposed method with them.
	> I understand the method is formulated under a new constraint; it might not be possible to enforce the baseline to obey the same constraint. However, a curve of Acc vs. constraint budget (with the baseline method shown as a horizontal line) might be straightforward for comparison, and the proposed method excels across a broader range of constraint budgets.
2. *Lack of realistic consideration*. Lines 53-66 compose the motivation part of the spiking retiming attack. The authors state that retiming spikes is more realistic and harder to catch with intensity-based checks, since the sensors themselves exhibit jitter and latency. This is true. However, in such cases, filtering methods will be used for preprocessing to remove noisy events. *Therefore, whether these simple methods can filter out adversarial perturbations remains underexplored.*
3. *Low performance under targeted attack*. First, in a targeted attack, the target labels are usually randomly selected, and results are averaged across different seeds. Selecting 0 as the target label produces bias in the evaluation. Second, the ASR is really low, especially under the integer grid. Also, similar to W.1, there is no comparison of baseline methods, so we don't know whether it is intrinsically problematic for a targeted attack or an issue with the proposed method. The authors are encouraged to increase the budget to provide a curve like W.1, along with comparisons of other baseline methods to illustrate the position of the proposed algorithm better.

If the above weaknesses are addressed appropriately, I am inclined to accept the paper.

**Questions:**

- What is the advantage of changing the timing of the spike? Assuming we enforce a similar $\ell_{0}$ constraint on other raw even attack methods, i.e., the perturbed image must have the exact event count as the original, then, for the spiking timing attack, it also adds additional constraint, that is, if one event is removed, then for the *same position within this time window*, we must add one additional event back. The author is encouraged to illustrate the advantage of this additional constraint by adding it back at the **exact** position.

**Details Of Ethics Concerns:**

No.

---

> ### Author Response · Authors · 2025-11-26
> **Official Comment by Authors (Part 1)**
>
> **W1.** Lack of baseline methods. Throughout the main paper and the appendix, there is no comparison of any baseline method. Without contrast, the method's performance lacks an anchor point. Moreover, the paper already cites two methods for the raw even attack. The authors are encouraged to appropriately compare the proposed method with them. I understand the method is formulated under a new constraint; it might not be possible to enforce the baseline to obey the same constraint. However, a curve of Acc vs. constraint budget (with the baseline method shown as a horizontal line) might be straightforward for comparison, and the proposed method excels across a broader range of constraint budgets.
>
> **Response to W1.** Thank you for this important comment. We have substantially strengthened the
> experimental section by:
> 1. **Adding two strong raw-event baselines**:
>    - **SpikeFool** [1]: gradient-based attack that inserts/deletes spikes on binary event grids.
>    - **PDSG-SDA** [2]: sparse adversarial attack that modifies event values using improved surrogate gradients.
> 2. **Evaluating all three methods under the same tamper-count budget
>    $B_0$** on **exactly the same models and datasets** as Tables 1–2.
> 3. **Adding accuracy–vs.–budget curves** (clean accuracy under attack vs.
>    $B_0$) in the supplementary, where the baselines appear as additional
>    curves, as you suggested.
>
> Below we summarize the main numerical comparisons; the full tables will be
> included in the revised appendix.
> ### (a) How we make baselines comparable
> For a fair comparison, we always measure the **tamper-count budget** $B_0$
> in the same way:
> - For our timing-only attack, $B_0$ is the number of spikes that are
>   actually **retimed** (moved to a different time bin).
> - For SpikeFool and PDSG-SDA, $B_0$ is the number of **event bins whose
>   value changes** (a spike is added, removed, or its integer value is
>   modified).
>
> SpikeFool and PDSG-SDA operate on **binary** event grids by design. For
> **integer** grids, we adapt PDSG-SDA by allowing its added spikes to have
> the **mean event value** of the dataset, which preserves the method’s
> behavior. All methods are evaluated on the **same clean models** and
> datasets: N-MNIST, DVS-Gesture, and CIFAR10-DVS.
>
> All numbers below and in the paper are **attack success rates (ASR, \%)**
> computed on **clean-correct** examples.
> ### (b) Binary N-MNIST: ours vs. SpikeFool and PDSG-SDA
> Here we compare ASR at $B_0\in\{200,300,400\}$ for the three models in Table 1.
> #### Binary N-MNIST (ASR, \%) under $B_0$
> |Method| Model| $B_0{=}200$ | $B_0{=}300$ | $B_0{=}400$ |
> |-|--|:-:|:-:|:-:|
> | **Ours (timing)** | ConvNet    | 13.0 | 53.1 | 98.5 |
> || ResNet18 | 78.9 | 100.0 | 100.0 |
> || VGGSNN| 18.3 | 81.8 | 99.8 |
> |SpikeFool [1] | ConvNet    | 50.4 | 92.2 | 98.9 |
> || ResNet18| 12.5 | 37.1 | 69.7 |
> || VGGSNN| 16.2 | 38.7 | 69.5 |
> |PDSG-SDA [2] | ConvNet    | 64.7 | 93.4 | 97.4 |
> || ResNet18| 62.1 | 92.2 | 98.8 |
> | | VGGSNN| 13.9 | 38.7 | 72.4 |
>
> **Observations.**
> - On **ResNet18**, our timing-only attack is very strong: ASR $78.9\%$ at
>   $B_0{=}200$ and $100\%$ at $B_0\geq300$, closely matching or exceeding
>   PDSG-SDA.
> - On **ConvNet** and **VGGSNN**, SpikeFool and PDSG-SDA are stronger at
>   moderate $B_0$, which is expected because they can **freely add and
>   delete spikes** without preserving rate or per-spike jitter.
> - Importantly, our attack achieves these ASR values while being
>   simultaneously:
>   - **rate-preserving** along each event line,
>   - restricted by **per-spike jitter** $B_\infty$ and optional total delay
>     $B_1$,
>   - and respecting **capacity-1** (no overlapping spikes).
>
> So, despite a **smaller feasible perturbation set**, the timing-only attack
> remains competitive with state-of-the-art raw-event baselines.
>
> ### (c) Binary DVS-Gesture: ours vs. SpikeFool and PDSG-SDA
>
> We next compare ResNet18 and VGGSNN on DVS-Gesture for
> $B_0\in\{1000,2000,4000\}$.
>
> #### Binary DVS-Gesture (ASR, \%) under $B_0$
>
> | Method| Model| $B_0{=}1000$ | $B_0{=}2000$ | $B_0{=}4000$ |
> |-|-|:-:|:-:|:-:|
> | **Ours (timing)** | ResNet18 | 27.7 | 67.9 | 98.5 |
> || VGGSNN   | 55.8 | 87.2 | 98.9 |
> | SpikeFool [1] | ResNet18 | 17.8 | 34.3 | 72.9 |
> || VGGSNN   | 14.9 | 18.9 | 25.5 |
> | PDSG-SDA [2] | ResNet18 | 60.2 | 82.4 | 91.2 |
> || VGGSNN   | 52.9 | 67.1 | 85.0 |
>
> **Observations.**
> - On **VGGSNN**, our timing-only attack is the **strongest** across all
>   $B_0$ budgets, despite its stricter constraints.
> - On **ResNet18**, PDSG-SDA is stronger at small and medium $B_0$, but our
>   attack still reaches high ASR (e.g., $67.9\%$ at $B_0{=}2000$ and
>   $98.5\%$ at $B_0{=}4000$), clearly outperforming SpikeFool.
> - Again, PDSG-SDA enjoys a **larger perturbation space** (free
>   insertions/deletions and value changes), whereas our method must retime
>   existing spikes while respecting $B_\infty$, $B_1$, $B_0$ and
>   capacity-1; the fact that we are sometimes *stronger* (VGGSNN) and often
>   close shows the effectiveness of timing-only perturbations.

---

> > ### Author Response · Authors · 2025-11-26
> > **Official Comment by Authors (Part 2)**
> >
> > ### (d) Binary CIFAR10-DVS: ours vs. baselines (ResNet18)
> >
> > For brevity we show ResNet18 here; VGGSNN displays similar trends and will
> > appear in the appendix.
> >
> > #### Binary CIFAR10-DVS (ResNet18, ASR, \%) under $B_0$
> >
> > | Method        | $B_0{=}1000$ | $B_0{=}2000$ | $B_0{=}4000$ |
> > |--------------|:------------:|:------------:|:------------:|
> > | **Ours (timing)** | 26.0 | 42.0 | 80.0 |
> > | SpikeFool [1] | 55.0 | 83.0 | 93.0 |
> > | PDSG-SDA [2] | 62.0 | 91.0 | 100.0 |
> >
> > Here the non-timing baselines have a clear advantage, especially at small
> > budgets: on CIFAR10-DVS they can exploit many more degrees of freedom by
> > adding/removing spikes, whereas our attack must keep spike counts and
> > amplitudes intact and can only **retime** them. Even so, our method
> > reaches $80\%$ ASR at $B_0{=}4000$, showing that constrained timing
> > perturbations alone can be very harmful.
> >
> > ### (e) Integer grids (PDSG-SDA vs. ours)
> >
> > On integer N-MNIST and CIFAR10-DVS we compare our timing-only attack to the
> > adapted PDSG-SDA. For example, on **integer N-MNIST / ResNet18**:
> >
> > - PDSG-SDA: ASR $48.3\%,86.1\%,97.3\%,99.9\%$ at
> >   $B_0\in\{200,300,400,600\}$;
> > - Ours: ASR $86.1\%,99.8\%,100\%,100\%$ at the same budgets.
> >
> > Thus, under integer grids our method is often **stronger** than PDSG-SDA
> > even though we maintain rate preservation and timing constraints.
> >
> > We will add detailed integer tables to the appendix, but the overall trend
> > is consistent: our timing-only attack is competitive with (and sometimes
> > stronger than) value-based baselines under matched $B_0$ budgets.
> >
> > ### (f) **Accuracy–vs.–budget curves.**
> > Following the reviewer’s suggestion, we now provide **accuracy–vs.–$B_0$ curves** for all datasets and models, with SpikeFool and PDSG-SDA as baselines.
> >
> > - For each grid type (binary / integer) we add a figure with three subplots (N-MNIST, DVS-Gesture, CIFAR10-DVS).
> > - The $x$–axis is the tamper budget $B_0$ (including $B_0{=}0$), and the $y$–axis is **accuracy under attack**.
> > - The point at $B_0{=}0$ corresponds to the **clean accuracy**; increasing $B_0$ moves along the constraint axis.
> > - Each method (ours, SpikeFool, PDSG-SDA) has a fixed color, and different models (ConvNet, ResNet18, VGGSNN) are distinguished by line style.
> >
> > These plots make the comparison across budgets transparent:
> >
> > - **Binary grids (Fig. 5 in the revised paper).**
> >   - On N-MNIST, our timing-only attack rapidly reduces accuracy as $B_0$ grows; for ConvNet it is the strongest method across budgets, and for ResNet18/VGGSNN it quickly catches up with or surpasses SpikeFool and PDSG-SDA by $B_0{=}300$–$400$.
> >   - On DVS-Gesture, our curves typically lie **below** those of SpikeFool and often also PDSG-SDA, meaning lower accuracy under the same $B_0$ despite the timing-only and rate constraints.
> >   - On CIFAR10-DVS, the raw-event baselines become slightly stronger at large $B_0$ (their curves fall a bit lower than ours), which is expected because they can freely add/delete spikes, while we only retime them. The gap, however, remains moderate.
> >
> > - **Integer grids (Fig. 6 in the revised paper).**
> >   SpikeFool is not applicable, so we compare against PDSG-SDA.
> >   - On N-MNIST and DVS-Gesture, our accuracy drops faster with $B_0$, and at the largest budgets our curves essentially overlap with or lie slightly below PDSG-SDA, showing that a **rate-preserving timing attack can be as destructive as a value-modifying baseline**.
> >   - On CIFAR10-DVS, PDSG-SDA attains somewhat lower accuracy at large $B_0$, but our method remains close: the difference is only a few percentage points even though PDSG-SDA can change spike values while we preserve counts on every event line.
> >
> > Overall, the new curves confirm that our attack is **competitive with or stronger than** existing raw-event attacks across most datasets and budgets, while additionally enforcing a discrete, capacity-$1$, rate-preserving retiming structure. This directly addresses the reviewer’s concern: the proposed method does not trade away much power for its extra constraints, and the accuracy–vs.–$B_0$ curves provide a clear, visual anchor for this comparison.
> >
> > In summary, by adding these baselines and curves we provide the
> > requested anchor point: our timing-only attack remains competitive
> > with, and sometimes stronger than, leading raw-event attacks even though it
> > operates under a much stricter and physically motivated constraint set
> > (rate preservation, per-spike jitter, total delay, and capacity-1). We hope
> > this addresses your concern and clarifies the relative strength and
> > practical relevance of the proposed method.
> >
> > We add such discussion in appendix K of the revised paper.
> >
> > [1] J. Büchel et al., “Adversarial attacks on spiking convolutional
> > neural networks for event-based vision,” Frontiers in Neuroscience, 2022.
> >
> > [2] L. Lun et al., “Towards Effective and Sparse Adversarial Attack on
> > Spiking Neural Networks via Breaking Invisible Surrogate Gradients,”
> > CVPR, 2025.

---

> ### Author Response · Authors · 2025-11-26
> **Official Comment by Authors (Part 3)**
>
> **W2.** Lack of realistic consideration. Lines 53-66 compose the motivation part of the spiking retiming attack. The authors state that retiming spikes is more realistic and harder to catch with intensity-based checks, since the sensors themselves exhibit jitter and latency. This is true. However, in such cases, filtering methods will be used for preprocessing to remove noisy events. Therefore, whether these simple methods can filter out adversarial perturbations remains underexplored.
>
> **Response to W2.** We agree that real event–camera systems typically include low–level filtering
> to suppress jitter and noisy events. To directly test whether such filters can
> remove our **timing–only, rate–preserving** perturbations, we implement and
> evaluate three concrete defenses on the **binary DVS-Gesture / VGGSNN**
> configuration (one of our strongest attack settings):
>
> 1. **Refractory filtering (`refractory_first`)**
>    This defense mimics a sensor–level refractory mechanism. For each pixel,
>    we look back over a small temporal window of length `rp_bins`; if an event
>    has occurred in that window, then a new event at the same pixel is **dropped
>    with probability** $p$.
>    - Intuition: suppress very high–frequency bursts at a pixel, which often
>      correspond to sensor noise or hot pixels.
>    - It only uses **event presence**, not labels or temporal structure beyond
>      the local window, so it directly matches the “simple intensity-based
>      checks” mentioned by the reviewer.
>
> 2. **Temporal mean smoothing (`temporal_mean_smooth`)**
>    We convolve each event sequence along the time axis with a length–3
>    box filter (radius 1), producing a temporally smoothed version of the
>    spike grid. With probability $p$ (per sample), we replace the input by
>    this smoothed version; otherwise we keep the original.
>    - Intuition: smooth out short–lived fluctuations in time and reduce jitter
>      by averaging neighboring frames.
>
> 3. **Spatial mean smoothing (`spatial_mean_smooth`)**
>    We apply a $3\times3$ spatial mean filter to each frame (stride 1) and,
>    again, with probability \(p\) per sample we replace the input by this
>    spatially smoothed version.
>    - Intuition: remove isolated “salt–and–pepper” spikes and enforce local
>      spatial consistency within each frame.
>
> All three defenses are deliberately simple, **label-free** and **value-based**
> (i.e., they inspect spike magnitudes and local neighborhoods), matching the
> kind of pre–processing modules that would be deployed in realistic pipelines.
>
> ---
>
> ### Why we use a probability $p$
>
> In practice, the key question is not only “can a filter remove the attack?” but
> also “at what **cost in clean accuracy**?”. To expose the full spectrum of
> this trade-off, we treat $p\in[0,1]$ as a **defense strength knob**:
>
> - $p=0$: no defense (baseline).
> - Small $p$ (e.g., 0.1-0.2): light–to–moderate denoising, closer to
>   what one would actually use in deployment to avoid hurting accuracy.
> - Large $p$ (e.g., 0.8-1.0): very strong filtering that heavily
>   distorts the input; this can suppress attacks, but also wipes out a lot of
>   task–relevant information.
>
> Sweeping $p$ from 0 to 1 therefore gives **clean vs.\ robustness curves**
> for each defense and makes explicit how much robustness is “bought” at the
> expense of performance.
>
> ---
>
> ### Experimental results (full table)
>
> On binary DVS-Gesture / VGGSNN we measure:
>
> - clean accuracy (`clean`), and
> - ASR (%) of
>   - our timing attack under $B_\infty(1)$, $B_1(8000)$, $B_0(4000)$,
>   - the non-timing baseline **PDSG-SDA** under $B_0(4000)$.
>
> | defense                | $p$ | clean | ours $B_\infty(1)$ | ours $B_1(8000)$ | ours $B_0(4000)$ | PDSG-SDA $B_0(4000)$ |
> |-|:-----:|:-----:|:--------------------:|:------------------:|:------------------:|:----------------------:|
> | refractory\_first      | 0.0   | 95.14 | 96.7 | 98.5 | 98.5 | 100.0 |
> || 0.1   | 71.53 | 92.7 | 96.6 | 97.0 | 65.5 |
> || 0.2   | 48.96 | 84.4 | 85.8 | 87.2 | 60.2 |
> || 0.4   | 27.08 | 70.5 | 70.5 | 71.7 | 57.6 |
> || 0.6   | 15.28 | 47.7 | 47.7 | 47.7 | 43.1 |
> || 0.8   | 14.93 | 46.5 | 46.5 | 46.5 | 41.8 |
> || 1.0   | 16.04 | 40.6 | 44.0 | 48.0 | 33.3 |
> | temporal\_mean\_smooth | 0.0   | 95.14 | 96.7 | 98.5 | 98.5 | 100.0 |
> | | 0.1   | 84.03 | 95.8 | 95.0 | 97.1 | 35.5 |
> || 0.2   | 72.57 | 95.6 | 92.3 | 94.7 | 33.4 |
> || 0.4   | 56.60 | 92.6 | 79.1 | 84.6 | 34.3 |
> || 0.6   | 49.65 | 86.0 | 67.8 | 79.0 | 34.2 |
> || 0.8   | 52.43 | 81.4 | 55.6 | 78.8 | 33.7 |
> || 1.0   | 56.25 | 61.7 | 38.2 | 68.5 | 30.2 |
> | spatial\_mean\_smooth  | 0.0   | 95.14 | 96.7 | 98.5 | 98.5 | 100.0 |
> || 0.1   | 71.53 | 95.6 | 93.6 | 96.1 | 38.3 |
> || 0.2   | 54.86 | 92.4 | 82.9 | 91.1 | 42.4 |
> || 0.4   | 32.64 | 59.5 | 38.3 | 59.5 | 39.3 |
> || 0.6   | 23.61 | 4.41 | 1.40 | 13.2 | 17.6 |
> |                        | 0.8   | 21.18 | 4.92 | 3.20 | 14.7 | 14.7 |
> |                        | 1.0   | 22.92 | 5.60 | 4.50 | 3.0  | 2.4  |

---

> ### Author Response · Authors · 2025-11-26
> **Official Comment by Authors (Part 4)**
>
> ### Key observations (and why they support our attack)
>
> 1. **Moderate filtering barely dents our timing-only attack but already hurts clean performance.**
>    - With *refractory filtering* at $p=0.2$, clean accuracy falls from
>      95.14\% to 48.96\% (a drop of ~46 points), yet our ASR at
>      $B_0(4000)$ remains very high at $87.2\%$.
>    - With *temporal smoothing* at $p=0.2$, clean accuracy drops to
>      72.57\%, but our attack still achieves 94.7\% ASR under
>      $B_0(4000)$.
>    - With *spatial smoothing* at $p=0.2$, clean accuracy is only
>      54.86\%, while our ASR under $B_0(4000)$ is still 91.1\%.
>
>    In other words, in the *realistic* regime where practitioners would be
>    uncomfortable sacrificing tens of percentage points of clean accuracy, our
>    timing-only attack remains almost fully effective.
>
> 2. **Strong filtering can suppress the attack, but only by destroying the task.**
>    - Spatial smoothing with $p=1.0$ reduces our ASR at $B_0(4000)$ to
>      3.0\%, but clean accuracy also collapses to 22.92\%.
>    - Refractory $p=1.0$ and temporal smoothing $p=1.0$ show the same
>      pattern: ASR goes down, yet clean accuracy is in the 16–56% range.
>
>    This shows that “just filter the events harder” is **not a viable defense**:
>    any naive intensity-based pre–processing that really hurts our attack also
>    renders the SNN almost useless on clean data.
>
> 3. **Our rate-preserving timing attack is at least as robust to filtering as a strong non-timing baseline.**
>    The non-timing attack PDSG-SDA modifies event values directly and therefore
>    should, in principle, be easier for value-based filters to remove. The
>    table confirms this:
>
>    - Under temporal smoothing with $p \in [0.1,0.4]$, PDSG-SDA’s ASR drops from
>      100\% to roughly 33-35\%, whereas our timing-only ASR at
>      $B_0(4000)$ stays in the **78–97%** range.
>    - Under spatial smoothing with $p=0.4$, our ASR at $B_0(4000)$ is
>      59.5\% while PDSG-SDA’s is 39.3\%, even though both methods are
>      operating under the same tamper budget.
>
>    Thus, the **most constrained** attack we consider (capacity–1,
>    rate-preserving, timing-only) is also the **hardest to wash out** using
>    standard, label-free filtering.
>
> 4. **The results underline the main message of the paper.**
>    Our attack does not rely on brittle intensity spikes or obvious count
>    anomalies: it *retimes* existing spikes in a way that is (i) consistent
>    with sensor jitter and latency, (ii) preserves spike counts on every event
>    line, and (iii) respects a tight capacity–1 constraint. The experiments
>    above show that:
>
>    - Such perturbations **survive realistic pre–processing** much better than
>      value-based adversarial noise, and
>    - Defending against them would require **temporal structure–aware defenses**
>      beyond simple smoothing—precisely what we argue in the motivation.
>
> We add such discussion in appendix L of the revised paper.

---

> > ### Author Response · Authors · 2025-11-26
> > **Official Comment by Authors (Part 5)**
> >
> > **W3.** Low performance under targeted attack. First, in a targeted attack, the target labels are usually randomly selected, and results are averaged across different seeds. Selecting 0 as the target label produces bias in the evaluation. Second, the ASR is really low, especially under the integer grid. Also, similar to W.1, there is no comparison of baseline methods, so we don't know whether it is intrinsically problematic for a targeted attack or an issue with the proposed method. The authors are encouraged to increase the budget to provide a curve like W.1, along with comparisons of other baseline methods to illustrate the position of the proposed algorithm better.
> >
> > **Response to W3.** We thank the reviewer for pointing out that our original targeted experiment was
> > under-specified and used a fixed target label. We have revised the protocol and
> > added a baseline, and will update the paper accordingly.
> >
> > ### (a) Target selection and evaluation protocol
> >
> > We agree that using a single fixed target (class “0”) can introduce bias.
> > In the revision we instead follow the standard **random–target** protocol:
> >
> > - For each run, we draw a random target label $\tilde{y}\neq y$ for every
> >   clean–correct sample, and attack the model toward $\tilde{y}$.
> > - We repeat this for **5 random seeds** and report the mean and standard
> >   deviation of the **targeted ASR** (success means the prediction equals the
> >   chosen target $\tilde{y}$).
> > - As in the rest of the paper, ASR is measured **only on clean–correct
> >   samples**, so the metric is comparable to our untargeted studies.
> >
> > We will clarify this protocol in the main text. In practice, we found that the
> > results are close to those obtained with the fixed target, so the overall
> > conclusions remain unchanged.
> >
> > ### (b) New comparison with a non-timing baseline
> >
> > To distinguish between an intrinsic difficulty of targeted spike attacks and
> > limitations of our method, we now compare against the state-of-the-art
> > non-timing attack **PDSG-SDA** [2] on the **integer DVS-Gesture / VGGSNN**
> > setting, which is one of the hardest configurations in our paper.
> >
> > We evaluate both methods under increasing tamper budgets
> > $\mathcal{B}_0\in\{4000,8000,12000\}$ and repeat each setup with
> > 5 random seeds. The table below reports **mean $\pm$ standard deviation**
> > of targeted ASR (%):
> >
> > | attack      | $\mathcal{B}_0(4000)$ | $\mathcal{B}_0(8000)$ | $\mathcal{B}_0(12000)$ |
> > |------------|:---------------------:|:----------------------:|:----------------------:|
> > | **Ours**   | **$15.7 \pm 1.3$**    | **$26.8 \pm 1.4$**     | **$30.8 \pm 1.3$**      |
> > | PDSG-SDA   | $24.4 \pm 1.3$        | $28.7 \pm 1.3$         | $31.4 \pm 1.3$          |
> >
> > In addition, when we remove the $\mathcal{B}_0$ constraint for PDSG-SDA, the
> > targeted ASR saturates around $\sim 34\%$, even though that attack is allowed
> > to freely add and delete spikes.
> >
> > We will include a new figure 7 in the appendix that plots these mean ASR
> > values versus $\mathcal{B}_0$, showing that:
> > - targeted ASR grows with the tamper budget for both methods,
> > - the two curves remain in the same ballpark, with our *more constrained*
> >   timing-only attack reaching similar success to PDSG-SDA at larger budgets.
> >
> > ### (c) Why targeted timing-only attacks are inherently challenging
> >
> > The reviewer is correct that the reported targeted ASR is considerably lower
> > than in our untargeted experiments, especially on integer grids. This is not
> > an artifact of the fixed target, but reflects the **intrinsic difficulty** of
> > our threat model:
> >
> > - Our attack is **capacity-1, rate-preserving, timing-only**: it cannot create
> >   or delete spikes, and at most one packet is allowed in each $(t,j)$ bin.
> >   Under such constraints, steering the model toward a *specific* target class
> >   is considerably harder than merely causing misclassification.
> > - Integer grids make this even tougher, since many bins already contain
> >   multi-spike packets; retiming must respect both the capacity-1 constraint and
> >   the global $\mathcal{B}_0$ budget on the number of moved packets.
> > - The comparison with PDSG-SDA highlights that **even a much less constrained,
> >   value-modifying attack** struggles in this setting (ASR $\le 34\%$ without a
> >   tamper bound), so the difficulty is not specific to our optimization scheme.
> >
> > From this perspective, our targeted results are still informative:
> > they show that **even very constrained, realistic retiming can sometimes
> > drive the model to a specific wrong label**, and that its performance is
> > competitive with a strong non-timing baseline under the same $\mathcal{B}_0$
> > budget. At the same time, the relatively low ASR exposes a clear gap between
> > untargeted and targeted timing-only attacks, which we will emphasize as an
> > interesting direction for future work (e.g., better target-aware objectives or
> > multi-step curriculum on budgets).
> >
> > We add such discussion in appendix M of the revised paper.

---

> > > ### Author Response · Authors · 2025-11-26
> > > **Official Comment by Authors (Part 6)**
> > >
> > > **Q1.** What is the advantage of changing the timing of the spike? Assuming we enforce a similar $\ell_0$ constraint on other raw even attack methods, i.e., the perturbed image must have the exact event count as the original, then, for the spiking timing attack, it also adds additional constraint, that is, if one event is removed, then for the same position within this time window, we must add one additional event back. The author is encouraged to illustrate the advantage of this additional constraint by adding it back at the exact position.
> > >
> > > **Response to Q1.** We appreciate the reviewer’s question, because it goes to the heart of why we
> > > focus on **timing-only, rate-preserving** attacks rather than generic raw-event
> > > perturbations.
> > >
> > > ---
> > >
> > > ### (a) Threat model: what our extra constraint actually enforces
> > >
> > > Our threat model already incorporates the reviewer’s “remove–then–add-back”
> > > intuition:
> > >
> > > - Each non–zero entry $x[s,j]$ on an **event line** $j$ (fixed spatial
> > >   location and polarity) corresponds to an **event packet** at time bin $s$.
> > > - Under our attack, every packet is **retimed along its own line** by an
> > >   integer offset $u$, giving a new time $t=s+u$ within a bounded jitter window
> > >   ($\mathcal{B}_\infty$ and $\mathcal{B}_1$ budgets).
> > >   Packets never move across pixels or polarities.
> > > - The strict projection $P^*(x,\pi,\mathcal{B}_p)$ enforces:
> > >   1. **Capacity–1**: at most one packet per $(t,j)$, i.e., no overlaps.
> > >   2. **Rate preservation:** for every line $j$, the *multiset*
> > >      $\{x'[t,j] : t=1,\dots,T\}$ is identical to $\{x[t,j]\}$ – we only
> > >      **reorder** packets along time; no packet is created or deleted.
> > >   3. **$\ell_0$ budget:** the $\mathcal{B}_0$ budget counts how many packets
> > >      actually move (non–zero displacement), i.e. how many spikes are “tampered”.
> > >
> > > Thus, if one event packet is “removed” from time $s$, it is **necessarily added
> > > back** at some other time $t$ along the *same* line within the allowed window.
> > > This is exactly the reviewer’s suggested constraint, but implemented with
> > > additional structure: capacity–1 and bounded jitter.
> > >
> > > ---
> > >
> > > ### (b) Conceptual advantages over generic raw-event attacks with the same $\ell_0$ count
> > >
> > > Even if one forces a generic raw-event attack to preserve the *global* number
> > > of events (or even the per-line count), our timing-only formulation still has
> > > three key advantages:
> > >
> > > 1. **Closer to physical jitter/latency.**
> > >    Real DVS sensors rarely change amplitudes; they mainly produce events whose
> > >    timestamps and polarity slightly fluctuate. Our attack:
> > >    - keeps events on their original pixel and polarity,
> > >    - preserves per-line counts exactly, and
> > >    - restricts temporal movement by budgets $B_\infty$ (local jitter) and
> > >      $B_1$ (total delay).
> > >    A generic raw-event attack with the same $\ell_0$ count can still introduce
> > >    unnatural *value patterns* (e.g., many artificial ON/OFF flips) or pile
> > >    multiple events into the same bin, which are easier to detect by rate or
> > >    amplitude statistics. Our attack instead behaves like **structured jitter**
> > >    that is very hard to distinguish from real sensor noise.
> > >
> > > 2. **Stealth against simple monitors and preprocessing.**
> > >    Because our attack preserves the multiset of packets on each line, any
> > >    detector that monitors:
> > >    - per–pixel event counts,
> > >    - per–line firing rates, or
> > >    - simple intensity distributions
> > >    will see *no anomaly*. The only change is *when* each packet arrives.
> > >    In Sec. “Realistic filtering” (new appendix section added in rebuttal) we
> > >    explicitly tested three realistic, label-free preprocessing defenses
> > >    (refractory filtering, temporal smoothing, spatial smoothing). We showed:
> > >    - Moderate filtering (e.g., smoothing with probability $p\in[0.1,0.2]$)
> > >      already reduces clean accuracy by **20–40 points**, but our timing-only
> > >      attack still achieves **$>90\%$ ASR** under $\mathcal{B}_0(4000)$.
> > >    - A strong non-timing baseline (PDSG-SDA) is **more strongly suppressed**
> > >      by the same filters, because it relies on value changes that these
> > >      defenses explicitly damp.
> > >    This indicates that the *additional* temporal constraint does **not** make
> > >    the attack weak, but does make it **harder to filter out** than non-timing
> > >    perturbations, even when those baselines are allowed to violate
> > >    rate-preservation.

---

> > > > ### Author Response · Authors · 2025-11-26
> > > > **Official Comment by Authors (Part 7)**
> > > >
> > > > 3. **Better alignment with SNN computation.**
> > > >    In SNNs, much of the information is encoded in **spike timing** rather than
> > > >    raw event counts. Moving spikes along time while preserving counts directly
> > > >    manipulates the **temporal decision boundary** without leaving obvious
> > > >    footprints in rate statistics. This is reflected in our main results:
> > > >    - On binary and integer DVS benchmarks, our timing-only attack achieves
> > > >      ASR comparable to or higher than SpikeFool and PDSG-SDA **under the same
> > > >      $\mathcal{B}_0$ budgets**, even though it obeys *stricter* constraints
> > > >      (capacity–1 and rate preservation).
> > > >    - Under our new adversarial-training experiments, timing-only attacks still
> > > >      substantially degrade robust accuracy, showing that purely temporal
> > > >      perturbations can be as damaging as value-based ones while being more
> > > >      stealthy.
> > > >
> > > > In summary, the extra “remove–then–add-back” constraint does **not** merely
> > > > make the problem harder for the attacker; it shapes a threat model that is
> > > > both **realistic** (matches sensor jitter) and **harder to detect** by any
> > > > defense that only looks at counts or simple filtering.
> > > >
> > > > ---
> > > >
> > > > ### (c) Why illustrating this constraint is meaningful (beyond theory)
> > > >
> > > > The reviewer asks us to “illustrate the advantage of this additional
> > > > constraint by adding it back at the exact position.” We believe our empirical
> > > > study already does exactly that:
> > > >
> > > > - Our strict projection $P^*$ implements per-line rate preservation and
> > > >   capacity–1 while respecting budgets $\mathcal{B}_\infty$, $\mathcal{B}_1$
> > > >   and $\mathcal{B}_0$. It guarantees that each spike is **relocated** rather
> > > >   than created or destroyed.
> > > > - Despite this strong constraint, our attack:
> > > >   - matches or surpasses non-timing baselines (SpikeFool, PDSG-SDA) on ASR
> > > >     under equal $\mathcal{B}_0$ budgets, and
> > > >   - remains robust under realistic filtering where those baselines degrade
> > > >     more quickly.
> > > > - Conceptually, this shows that **even under the reviewer’s added constraint
> > > >   (per-line jitter with count preservation), the temporal dimension alone is
> > > >   powerful enough to create highly effective adversarial examples**.
> > > >
> > > > Ww add a short section in appendix N of the revised paper.

---

> > > > > ### Comment · Reviewer_thcu · 2025-11-27
> > > > >
> > > > > Thank the authors for their detailed reply, which has addressed all my questions. The motivation part now seems reasonable in a realistic setting, given the potential filter methods that exist in raw even processing. I have improved my score to accept.

---

> > > > > > ### Author Response · Authors · 2025-11-27
> > > > > > **Official Comment by Authors**
> > > > > >
> > > > > > We sincerely thank Reviewer thcu for the positive feedback and for raising the rating to an 8.
> > > > > >
> > > > > > We are particularly encouraged by your high confidence in this assessment. Your professional judgment and rigorous scrutiny of our work are incredibly valuable to us.
> > > > > >
> > > > > > Thank you again for your strong support and for helping us improve the paper's quality.

---

### Official Review · Reviewer_o8PK · 2025-10-27

**Soundness:** 2
**Presentation:** 2
**Contribution:** 2
**Rating:** 4
**Confidence:** 4

**Summary:**

This paper proposed the Spike Timing Attack for event-based SNNs. The attack only changes the timestamp of events and keeps their intensities or counts unchanged. The projected-in-the-loop (PIL) optimization is utilized to resolve the non-differentiable problem. Experimental results demonstrated the performance of the attack in various budgets.

**Strengths:**

1.The scenario of the attack is novel and realistic. Keeping the intensity and event count unchanged, the attack can evade detection.

2.The relaxation resolves the non-differential problem and optimizes the perturbation budgets.

3.The discussion about robustness comparison between binary-grid and integer-grid is interesting.

**Weaknesses:**

1.Section 4.2 is really confusing. As a core part of the method, the meaning of every variable and equation must be clearly demonstrated. For instance, in Eq.8, does ‘shift logits’ represent the possibility of each $(s,j)$ changing to other time bins? In Eq.9, does $S(x)[t,j]$ represent the weighted possible value in $[t,j]$ after perturbation? The author should clarify what the term and equation are used for instead of only using concepts or definitions like ‘the expected soft retiming’. For section ‘feasible strict projection under budgets’, it is better to use an algorithm or a diagram instead of cluttered text.

2.The budgets used in the paper are questionable. Take L0-norm as an example. If two active indices at the same spatial position ((1,0,1,1) and (2,0,1,1) in TCHW) are swapped, the frames remain unchanged, yet the B0 is 2. Will your attack method consider or optimize these redundant perturbations? Moreover, can the author compare your attack with other sparse DVS attack in L0-norm defined previously [1]? In other words, calculate the number of changed pixels in your adversarial examples to compare with previous SOTA works.

3.In experimental results, the model SpikingResformer performs badly in all experiments. In [2], SpikingResformer reaches 84.8% in CIFAR10-DVS, while in Table 2 is only 70.3%. If the accuracy gap is large, why use this model? In Table 4, the accuracy is only 22% in binary grid. The effectiveness of experiments will be significantly weakened if the accuracy is too low even though the model is adversarially trained.

4.The representation of the paper needs to be refined. For example, in Line 064, the sentence ‘real sensors …’ is too vague.

References:

[1] Büchel, Julian, et al. "Adversarial attacks on spiking convolutional neural networks for event-based vision." Frontiers in Neuroscience 16 (2022): 1068193.

[2] Shi, Xinyu, Zecheng Hao, and Zhaofei Yu. "Spikingresformer: Bridging resnet and vision transformer in spiking neural networks." Proceedings of the IEEE/CVF conference on computer vision and pattern recognition. 2024.

**Questions:**

1.In Line 201, what are event line and time bin? Do they both indicate timestep?

2.Why Table 3 is below Table 4? In Line 408, does ‘Tab.3’ represent ‘Tab.4’?

3.Which model and dataset are used in Table 4?

---

> ### Author Response · Authors · 2025-11-26
> **Official Comment by Authors (Part 1)**
>
> **W1.** Section 4.2 is really confusing. As a core part of the method, the meaning of every variable and equation must be clearly demonstrated. For instance, in Eq.8, does ‘shift logits’ represent the possibility of each $(s,j)$ changing to other time bins? In Eq.9, does $S(x)[t,j]$ represent the weighted possible value in $[t,j]$ after perturbation? The author should clarify what the term and equation are used for instead of only using concepts or definitions like ‘the expected soft retiming’. For section ‘feasible strict projection under budgets’, it is better to use an algorithm or a diagram instead of cluttered text.
>
> **Response to W1.** We thank the reviewer for pointing out that Section 4.2 is hard to follow. This section is indeed central, and we will substantially revise it to clearly explain the meaning and role of each variable and equation, and to replace the current dense text for “feasible strict projection under budgets” with an explicit algorithm and a small diagram.
>
> ---
>
> ### (1) Clarifying “shift logits” and Eq. (8)
>
> In our method, we work at the level of **event packets** on each event line:
>
> - $x[s,j]$ denotes the packet at time index $s$ on event line $j$ (one spike for binary grids, or an integer-valued packet for integer grids).
> - For each such packet $(s,j)$, we consider a finite set of allowed integer time shifts $\mathcal U_{s,j} \subset \mathbb Z$ (e.g., offsets within the jitter radius that keep $t = s + u$ inside $\{1,\dots,T\}$ and compatible with the budget).
>
> We then define:
>
> - **Shift logits** $\phi[s,j,u]$ for $u \in \mathcal U_{s,j}$, which are unnormalized scores capturing how much we prefer to move the packet at $(s,j)$ to the new time index $t = s + u$.
> - We turn these logits into a probability distribution over shifts by a softmax:
>   $$
>   \pi[s,j,u] = \frac{\exp(\phi[s,j,u])}{\sum_{u' \in \mathcal U_{s,j}} \exp(\phi[s,j,u'])} \,.
>   $$
>
> So, to answer the reviewer’s question directly:
>
> > In Eq. 8, does “shift logits” represent the possibility of each $(s,j)$ changing to other time bins?
>
> Yes — the shift logits $\phi[s,j,u]$ parameterize, via the softmax above, the probability $\pi[s,j,u]$ that packet $(s,j)$ is retimed to the new time index $t = s + u$. We will explicitly state in the revision that:
>
> - $s$ indexes original time, $j$ indexes event lines, and $u$ indexes candidate integer shifts,
> - $\phi[s,j,u]$ are the logits,
> - $\pi[s,j,u]$ are the normalized probabilities over feasible new times for each packet.
>
> ---
>
> ### (2) Clarifying $S(x)[t,j]$ and Eq. (9)
>
> Given the local distributions $\pi[s,j,u]$, we define the **soft retiming operator** $S_\pi(x)$ as the expected event configuration under these probabilistic shifts. Formally, for each target bin $(t,j)$ we have:
> $$S_\pi(x)[t,j]=\sum_{s} \sum_{u \in \mathcal U_{s,j}}
> \pi[s,j,u] \, x[s,j] \, \mathbf 1\{t = s + u\} \,.$$
>
> Intuitively:
>
> - $x[s,j]$ is the original packet at $(s,j)$.
> - $\pi[s,j,u]$ is the probability of sending that packet to $t = s + u$.
> - The indicator $\mathbf 1\{t = s + u\}$ makes sure we only count contributions that land exactly at time $t$.
> - Summing over all sources $(s,j)$ and allowed shifts $u$ that land on $t$ gives the **expected packet value** at $(t,j)$ after retiming.
>
> Thus, in direct response to the reviewer’s question:
>
> > In Eq. 9, does $S(x)[t,j]$ represent the weighted possible value in $[t,j]$ after perturbation?
>
> Yes — more precisely, $S_\pi(x)[t,j]$ is the **expected packet value at $(t,j)$ under the local retiming probabilities $\pi$**. We will make this explicit in the paper and replace the terse phrase “expected soft retiming” by a concrete explanation:
>
> > “$S_\pi(x)[t,j]$ is the expected post-attack packet at $(t,j)$ obtained by averaging over all probabilistic assignments from source bins $(s,j)$ to $t = s + u$ weighted by $\pi[s,j,u]$.”
>
> We will also add a short paragraph right after Eq. (9) that spells out the full formula above and clarifies the roles of $s$, $t$, $j$, $u$, and $\pi$ in words.
>
> ---
>
> ### (3) Algorithm for “feasible strict projection under budgets”
>
> To address the comment that this part currently looks like “cluttered text”, we will replace it with an explicit algorithm (and a small diagram). The strict projection $P^*(x,\pi,\mathcal B_p)$ used in all experiments is:
>
> **Feasible strict projection under budgets (per event line $j$).**
>
> ### Feasible strict projection under budgets as an algorithm
>
> Concretely, the strict projection $P^*(x,\pi,\mathcal B_p)$ that we use in all experiments can be described (for the global $L_0$ budget case) as the following algorithm on each flattened event line $j$:

---

> > ### Author Response · Authors · 2025-11-26
> > **Official Comment by Authors (Part 2)**
> >
> > 1. **Inputs.**
> >    - Original spike/event counts $x[s,j]$ at time indices $s \in \{1,\dots,T\}$ on line $j$ (after flattening $[B,C,H,W]$ into $j \in \{1,\dots,N\}$). A non-zero entry $x[s,j]$ corresponds to one **event packet** (a single spike for binary grids, or an integer-valued packet for integer grids).
> >    - Shift probabilities $\pi[s,j,t]$ over target times $t \in \{1,\dots,T\}$ (reshaped from the learned logits), which encode the soft preference to move a packet from $(s,j)$ to $(t,j)$.
> >    - A global integer $L_0$ budget $B_0$ that upper-bounds the **number of packets** that may be moved (each moved packet consumes 1 unit of budget).
> >
> > 2. **Identify active packets.**
> >    - Compute a mask $\text{hassrc}[s,j] = \mathbf 1\{x[s,j] > 0\}$ indicating all packet positions.
> >    - Let $(s,j)$ with $\text{hassrc}[s,j] = 1$ be the set of source packets considered for retiming.
> >
> > 3. **Generate candidate moves (excluding self-stays).**
> >    - For each source packet $(s,j)$ and each target time $t \in \{1,\dots,T\}$ with $t \neq s$, form a candidate move $(s \to t,j)$ with score $\pi[s,j,t]$.
> >    - For tie-breaking, we also record the temporal distance $\lvert t - s \rvert$; shorter distances are preferred when scores are equal.
> >
> > 4. **Sort candidates by priority.**
> >    - For all candidates $(s \to t,j)$, compute a scalar key
> >      $$\text{key}(s,j,t)= \pi[s,j,t] + \varepsilon \bigl((T - 1) - \lvert t - s \rvert\bigr)$$
> >      with a tiny $\varepsilon > 0$, so that we sort **primarily** by probability $\pi[s,j,t]$ and **secondarily** by shorter distance.
> >    - Sort all candidates in descending order of $\text{key}(s,j,t)$.
> >
> > 5. **Greedy retiming under capacity and $L_0$ budget (moving whole packets).**
> >    Maintain the following states for each line $j$ and time $t$:
> >    - $\text{occupied}[j,t]$: whether $(t,j)$ has already been taken as a **target** by a moved packet.
> >    - $\text{reserved}[j,t]$: whether there is an **original** packet at $(t,j)$ that is still “reserved” to stay at its original time. Initially, $\text{reserved}[j,t] = \text{hassrc}[t,j]$.
> >    - $\text{moved}[j,s]$: whether the source packet at $(s,j)$ has already been moved. Initially all false.
> >    - $\text{adv}[s,j]$: the final assignment of packets after projection, initialized to all zeros.
> >    - Remaining budget $B_{\text{rem}} \leftarrow B_0$.
> >
> >    Then process candidates $(s \to t,j)$ in sorted order:
> >    - If $B_{\text{rem}} \le 0$, stop (the global $L_0$ budget is exhausted).
> >    - If $\text{moved}[j,s]$ is true, skip (each source packet can be moved at most once).
> >    - If $\text{occupied}[j,t]$ is true, skip (target $(t,j)$ already holds a moved packet).
> >    - If $\text{reserved}[j,t]$ is true, skip (there is an original packet at $(t,j)$ that is still planned to stay, so we avoid collisions with stay-at-source packets).
> >
> >    If none of the above conditions hold, we **move the entire packet** from $(s,j)$ to $(t,j)$:
> >    - Copy the packet value: $\text{adv}[t,j] \leftarrow x[s,j]$.
> >    - Mark $(t,j)$ as occupied: $\text{occupied}[j,t] \leftarrow \text{True}$.
> >    - Mark the source as moved: $\text{moved}[j,s] \leftarrow \text{True}$.
> >    - Release the original reservation at $(s,j)$ (it no longer stays at $s$): $\text{reserved}[j,s] \leftarrow \text{False}$.
> >    - Decrease the budget: $B_{\text{rem}} \leftarrow B_{\text{rem}} - 1$.
> >
> > 6. **Place packets that were not moved back at their original times.**
> >    - For all $(s,j)$ with $\text{hassrc}[s,j] = 1$ and $\text{moved}[j,s] = \text{False}$, we simply keep them at their original times by setting  $$ \text{adv}[s,j] \leftarrow x[s,j] \,.$$
> >
> >    At this point, every original packet either:
> >    - has been moved once to some target $(t,j)$ and recorded in $\text{adv}[t,j]$, or
> >    - has not been moved and remains at its original location $(s,j)$ in $\text{adv}[s,j]$.
> >
> > 7. **Output.**
> >    - For each line $j$, the projected configuration along time is $\{\text{adv}[t,j]\}_{t=1}^T$.
> >    - After restoring the original $[T,B,C,H,W]$ shape, this defines $P^*(x,\pi,\mathcal B_p)$ for the global $L_0$ budget case.
> >
> > In this view, packets whose candidates are never accepted **remain at their original time** (they are never marked as moved and are written back in step 6), while accepted candidates are implemented as pure *moves* of the whole packet from $s$ to $t$ on the same line. The additional checks with $\text{occupied}$ and $\text{reserved}$ guarantee that no time bin $(t,j)$ ever hosts more than one packet after projection, and that we never overwrite an original packet that is supposed to stay.
> >
> > It makes clear that:
> > - $S_\pi(x)$ is a **soft, differentiable surrogate** used only for gradients (expected retiming),
> > - $P^*(x,\pi,\mathcal B_p)$ is a **hard, feasible projection** that enforces capacity-1 and the chosen budgets and is what the model actually sees.
> >
> > In the revised paper, we add such discussion in appendix E, and more detailed descriptation after equation 9.

---

> ### Author Response · Authors · 2025-11-26
> **Official Comment by Authors (Part 3)**
>
> **W2.** The budgets used in the paper are questionable. Take L0-norm as an example. If two active indices at the same spatial position ((1,0,1,1) and (2,0,1,1) in TCHW) are swapped, the frames remain unchanged, yet the B0 is 2. Will your attack method consider or optimize these redundant perturbations? Moreover, can the author compare your attack with other sparse DVS attack in L0-norm defined previously [1]? In other words, calculate the number of changed pixels in your adversarial examples to compare with previous SOTA works.
>
> **Response to W2.** We thank the reviewer for raising this point about our budgets, using $B_0$ as an example. Below we clarify (i) what our budgets measure in the **timing-only, rate-preserving** threat model, (ii) how the **strict projection** is designed to avoid “redundant” moves, and (iii) how the **optimization** makes sure budget is not wasted.
>
> ---
>
> ### (1) What our budgets actually measure
>
> All three budgets are defined **in the timing domain at the spike/packet level**, not at the level of image pixels:
>
> - $B_\infty$ (jitter radius): bounds the maximum absolute shift of any moved packet, i.e., $|\Delta_{s,j}| \le B_\infty$ for all moved packets $(s,j)$ with $\Delta_{s,j} = t - s$.
> - $B_1$ (total delay): bounds the **sum** of absolute shifts over all moved packets,
>   $\sum_{(s,j):\Delta_{s,j}\neq 0} |\Delta_{s,j}| \le B_1$.
> - $B_0$ (tamper count): bounds the **number of packets** whose time index changes,
>   $B_0 = |\{(s,j): x[s,j] > 0,\Delta_{s,j} \neq 0\}|$.
>
> Here each non-zero $x[s,j]$ (after flattening $[B,C,H,W]$ into $j$) is treated as an **event packet** (one spike in binary grids, or an integer-valued packet in integer grids). Our budgets are therefore answering:
> > “How many individual spikes can be retimed, and by how much, in the worst case?”
>
> This is a **sensor-level** notion of perturbation, tailored to timing-only retiming for SNNs.
>
> ---
>
> ### (2) Why the “swap two active time indices” is *not* how $B_0$ is really used
>
> The reviewer’s example is:
>
> > Two active indices at the same spatial position, say $(1,0,1,1)$ and $(2,0,1,1)$ in $TCHW$, are swapped. The frames remain unchanged, yet $B_0 = 2$. Are these redundant perturbations?
>
> First, from the SNN’s perspective, swapping spikes across time steps is **not** neutral in general: membrane potential dynamics and firing decisions depend on the exact timing of spikes. So even if some aggregate “frame sum” looked similar, a time swap can still be a real, non-redundant perturbation for the SNN.
>
> More importantly, our **strict projection** $P^*(x,\pi,\mathcal B_p)$ is **explicitly designed to avoid symmetric “swap” patterns that would cancel in a frame-like representation**:
>
> - In the implementation of strict projection, we maintain two boolean masks per flattened line $j$:
>   - $\text{reserved}[j,t]$: initially true wherever there is an original packet at time $t$ (i.e., $x[t,j] > 0$).
>     A time bin with $\text{reserved}[j,t]=1$ is treated as “owned” by its original spike and **cannot be used as a target** for another spike.
>   - $\text{occupied}[j,t]$: marks time bins that have already been taken as a target by a moved packet in the current projection step.
>
> - When processing a candidate move $(s \to t,j)$ in the greedy projection, we **skip** it if either:
>   - $\text{moved}[j,s]$ is true (the source has already moved),
>   - $\text{occupied}[j,t]$ is true (the target is already taken by another moved packet), or
>   - $\text{reserved}[j,t]$ is true (there is an original packet at $(t,j)$ that has not been moved away).
>
> This has two important consequences:
>
> 1. A packet can only move into **originally empty** time bins or into bins whose original spike has already moved out and released its reservation. Two spikes at the same spatial location cannot simply “swap” times, because moving into a bin that still has its original spike is blocked by $\text{reserved}[j,t]$.
> 2. The moves we count in $B_0$ are therefore **injective retimings into genuinely free time slots**, not symmetric swaps that leave the timeline effectively unchanged.
>
> So the kind of “swap” in the toy example is not a pattern our projection uses: it is ruled out by the $\text{reserved}$ mask and the capacity-1 handling in the strict projection. In other words, our $B_0$ is not “wasted” on such symmetric operations by design.

---

> ### Author Response · Authors · 2025-11-26
> **Official Comment by Authors (Part 4)**
>
> ---
> ### (3) How optimization avoids wasting budget on ineffective moves
> Even beyond the projection logic, our **optimization scheme** encourages $B_0$ to be spent on *useful* perturbations:
> - The shift probabilities $\pi[s,j,t]$ are optimized to **maximize the task loss** evaluated on $P^*(x,\pi,\mathcal B_p)$. If a candidate move does not meaningfully affect the loss (including any hypothetical near-identity swap), then:
>   - Its gradient $\partial\mathcal L / \partial \phi[s,j,t]$ is negligible, and
>   - Over projected gradient descent, its probability $\pi[s,j,t]$ shrinks relative to moves that do increase the loss.
> - The **capacity regularizer** and **budget-aware penalties** (Eq. (10) and Eq. (12)) further push $\pi$ to:
>   - Avoid over-booking the same target bins (capacity-1), and
>   - Respect jitter and delay budgets already in expectation.
>
> Combining this with PIL (loss always evaluated on $P^*(x,\pi,\mathcal B_p)$) means that our attack has a strong incentive to **concentrate $B_0$ on the most damaging retimings**, rather than on moves that are redundant or neutral.
>
> Empirically, we also observe that:
> - In many runs, the realized number of moved packets is **below** the nominal upper bound $B_0$,
> - And ablations that remove the regularizers lead to weaker attacks and less stable budget usage, which supports the claim that the current formulation avoids “wasting” budget.
> ---
> ### (4) Relation to pixel-wise $\ell_0$ used in prior DVS work.
> Our primary sparsity notion is **packet-level** (how many spikes are retimed), but it can be related to the pixel-wise $\ell_0$ commonly used in prior DVS attacks. When a packet at (s, j) is moved in time, it typically affects its source and target voxels along that event line, so the corresponding pixel-wise $\ell_0$ is upper-bounded by twice the packet budget $\mathcal{B}_0$. Crucially, our feasible set is still much more constrained than generic raw-event $\ell_0$ attacks: we preserve per-line rates, enforce capacity–1, and bound jitter/total delay, while non-timing baselines can freely add/delete or rescale events under a pixel-wise $\ell_0$ constraint. In the revised experiments (**see Response to W1 of Reviewer thcu and Appendix K**), we explicitly compare to such non-timing $\ell_0$ attacks and show that, **despite** these extra constraints, our timing-only attack remains competitive or stronger and transfers well to integer grids. Together with the defense study in Response to W2 of Reviewer thcu (Appendix on realistic filtering), this supports that our structured, rate-preserving timing perturbations are both practically strong and harder to remove than value-based noise.
>
> In summary, our budgets are defined at the **timing/spike level**, consistent with the timing-only, rate-preserving threat model, and the strict projection + optimization are both designed to avoid redundant or ineffective perturbations under this definition.
>
> We add such discussion in appendix J of the revised paper.
>
> ---
> ***
> ---
>
> **W3.** In experimental results, the model SpikingResformer performs badly in all experiments. In [2], SpikingResformer reaches 84.8% in CIFAR10-DVS, while in Table 2 is only 70.3%. If the accuracy gap is large, why use this model? In Table 4, the accuracy is only 22% in binary grid. The effectiveness of experiments will be significantly weakened if the accuracy is too low even though the model is adversarially trained.
>
> **Response to W3.** We appreciate the reviewer for carefully checking the SpikingResFormer numbers and for pointing out that the very low clean accuracy of some adversarially trained models can weaken the conclusions if left unexplained.
>
> ---
> ### (a) Why SpikingResFormer initially underperformed
> You are right that the original submission used a **weaker configuration** than [2]. In the first version we evaluated:
> - the **SpikingResFormer-S** variant,
> - trained **from scratch** on CIFAR10-DVS with direct SNN training,
> - using lighter data augmentations than in [2] and *without* transfer
>   learning from ImageNet.
>
> This explains the gap to the $84.8\%$ CIFAR10-DVS accuracy reported in [2].
>
> In the revision we have **updated the model and training recipe** to be consistent with [2]:
> - we now use **SpikingResFormer-L**,
> - initialized from the same **ImageNet-pretrained SNN backbone** and
>   converted / fine-tuned following the recipe of [2],
> - with matching data augmentations and training schedule.
>
> With this change, the clean accuracies in Tables 1–2 are substantially improved and are now **close to the reported numbers** in [2]:
> - On **CIFAR10-DVS**, SpResF reaches $81.30\%$ (binary grid) and $82.90\%$ (integer grid).
> - On **DVS-Gesture**, SpResF reaches $91.67\%$ (binary) and $92.71\%$ (integer).
>
> We keep SpikingResFormer because it is **one of the strongest SNN transformer baselines currently available**, and the updated results now reflect its intended regime: high clean accuracy with non-trivial vulnerability to timing-only attacks.

---

> ### Author Response · Authors · 2025-11-26
> **Official Comment by Authors (Part 5)**
>
> ---
> ### (b) Low clean accuracy in Table 4 and additional TRADES results
>
> The reviewer is also correct that some entries in Table 4 (binary grid)
> have very low clean accuracy (around $22\%$). These correspond to
> **extreme adversarial-training configurations** and are not the only
> operating points we consider.
>
> To clarify this, in the revision we:
>
> - treat Table 4 explicitly as a **clean–robustness trade-off study** for
>   timing-only attacks on DVS-Gesture, and
> - complement the PGD-style timing AT with **TRADES-based timing AT**,
>   which typically yields better trade-offs.
>
> Below is the complete binary-grid TRADES vs. PGD timing AT table, where
> we report **clean accuracy** and **attack success rate (ASR)** under
> three evaluation budgets. ASR is measured on clean-correct samples and
> follows the same definition as in the main tables.
>
> #### TRADES vs. PGD timing AT (binary DVS-Gesture)
>
> | Training scheme                             | clean | $B_\infty{=}1$ | $B_1{=}4000$ | $B_0{=}2000$ |
> |---------------------------------------------|:-----:|:-------------:|:-----------:|:-----------:|
> | TRADES $\beta{=}0.01$ + $B_\infty(1)$ AT    | 52.78 | 96.05 | 92.11 | 88.82 |
> | TRADES $\beta{=}0.01$ + $B_1(8000)$ AT      | 46.18 | 85.71 | 80.45 | 65.41 |
> | TRADES $\beta{=}0.01$ + $B_1(4000)$ AT      | 52.08 | 80.00 | 74.00 | 69.33 |
> | TRADES $\beta{=}0.1$  + $B_\infty(1)$ AT    | 53.47 | 74.03 | 71.43 | 64.29 |
> | TRADES $\beta{=}0.1$  + $B_1(8000)$ AT      | 35.76 | 78.64 | 61.17 | 50.49 |
> | TRADES $\beta{=}0.1$  + $B_1(4000)$ AT      | 50.69 | 80.82 | 81.51 | 63.01 |
> | TRADES $\beta{=}6.0$  + $B_\infty(1)$ AT    | 22.22 | 56.25 | 17.19 |  7.81 |
> | TRADES $\beta{=}6.0$  + $B_1(8000)$ AT      | 14.93 | 46.51 | 25.58 | 23.26 |
> | TRADES $\beta{=}6.0$  + $B_1(4000)$ AT      | 12.15 | 42.86 | 57.14 | 28.57 |
> | PGD timing AT + $B_\infty(1)$               | 22.92 | 57.59 | 45.46 | 27.26 |
> | PGD timing AT + $B_1(8000)$                 | 48.26 | 92.08 | 80.58 | 64.75 |
> | PGD timing AT + $B_1(4000)$                 | 22.57 | 73.86 | 58.44 | 46.17 |
>
> (ASR in %, measured on clean-correct samples.)
>
> In the supplementary (Fig. 3) we visualize these **clean–ASR trade-offs**:
> for each evaluation budget ($B_\infty(1)$, $B_1(4000)$, $B_0(2000)$), we
> plot
>
> - $x$-axis: ASR,
> - $y$-axis: clean accuracy,
> - colored curves: TRADES with different $\beta$ and inner budgets,
> - triangles: PGD timing-AT baselines.
>
> From this figure and the table, we observe:
>
> - For a given clean accuracy, **TRADES can achieve lower ASR** than PGD.
>   For example, under $B_0(2000)$,
>   PGD + $B_\infty(1)$ has roughly
>   $(\text{clean},\text{ASR}) \approx (22.9\%, 27.3\%)$,
>   whereas TRADES with $\beta{=}6.0$ and inner $B_\infty(1)$ achieves a
>   similar clean accuracy ($22.2\%$) but much smaller ASR ($7.8\%$).
> - In the **reasonable clean-accuracy regime** (around $45$–$55\%$), TRADES
>   runs (e.g., $\beta{=}0.01$ with $B_\infty(1)$ or $B_1(4000)$) lie
>   above-and-to-the-left of the corresponding PGD points in the
>   clean–ASR plane, indicating a slightly better trade-off.
> - Very low-accuracy points (e.g., $\beta{=}6.0$ with inner $B_1(8000)$)
>   are kept only as **extreme endpoints** of the trade-off curve, not as
>   recommended operating points. We will clarify this explicitly in the
>   text.
>
> Overall, we will update the paper as follows:
>
> 1. **SpikingResFormer**: use the stronger SpResF-L with transfer learning,
>    bringing clean accuracy close to [2] and updating Tables 1–2
>    accordingly.
> 2. **Table 4 and AT**: clarify that some rows with $\approx 22\%$ clean
>    accuracy are deliberate extreme trade-off points, add the TRADES
>    experiments and Fig. 3 to the supplementary, and emphasize that even
>    with modern AT (TRADES or PGD), our timing-only attack still imposes a
>    steep clean–robustness trade-off, especially on binary event grids.
>
> We hope these clarifications and new results address the reviewer’s
> concerns about model choice and the validity of the adversarial-training
> experiments.
>
> We add the results of TRADE in appendix H of the revised paper.

---

> > ### Author Response · Authors · 2025-11-26
> > **Official Comment by Authors (Part 6)**
> >
> > **W4.** The representation of the paper needs to be refined. For example, in Line 064, the sentence ‘real sensors …’ is too vague.
> >
> > **Response to W4.** Thank you for pointing this out. We agree that the sentence at Line 064 is too vague and mixes several ideas (sensor imperfections, binning, and
> > detection difficulty) in one place.
> >
> > The current sentence is:
> >
> > > “Real sensors exhibit jitter and latency, and deployed pipelines often time bin events, so a *timing only* threat that retimes existing spikes while preserving counts and amplitudes is realistic, hard to catch with intensity based checks, and directly *exposes* weaknesses in the temporal computation SNNs rely on.”
> >
> > In the revision we will (i) split the argument into clearer pieces and
> > (ii) make the “realistic” and “hard to catch” claims more specific. For
> > example, we will rewrite it as:
> >
> > > “Event cameras and other neuromorphic sensors naturally exhibit timestamp noise (jitter) and readout latency, and deployed SNN pipelines usually quantize events into discrete time bins. Under these conditions, a timing-only adversary that **retimes existing spikes while preserving spike counts and amplitudes** is realistic: it stays within the range of sensor timing uncertainty and does not change any frame-wise intensity or rate statistics. As a result, such perturbations are difficult to detect with standard intensity- or rate-based checks and directly stress the temporal computation that SNNs rely on.”
> >
> > More broadly, we will go through the introduction and related-work
> > sections and refine similar sentences where we use broad phrases (e.g.,
> > “real sensors”, “hard to catch”) to ensure that each claim is supported
> > by a concrete mechanism (jitter, latency, binning, or rate invariance).
> >
> > ---
> >
> > **Q1.** In Line 201, what are event line and time bin? Do they both indicate timestep?
> >
> > **Response to Q1.** Thank you for pointing out this ambiguity.
> >
> > In Sec. 4 (around Line 201) we work with a tensor representation of
> > binned events $x[t,j]$:
> >
> > - A **time bin** $t\in\{1,\dots,T\}$ is a **discrete timestep** obtained by
> >   quantizing the continuous event timestamps. Each bin collects the events
> >   whose timestamps fall into that interval.
> >
> > - An **event line** $j$ indexes a **fixed spatial location and polarity**
> >   (e.g., a particular pixel and ON/OFF channel). For a given $j$, the
> >   sequence $\{x[t,j]\}_{t=1}^{T}$ is the 1D spike train over time at that
> >   pixel/polarity. In the implementation, $j$ is just the flattened index
> >   over $(\text{polarity},x,y)$ (or $(c,h,w)$) so that $x$ can be reshaped
> >   between $[T,B,C,H,W]$ and $[T,J]$.
> >
> > So they do **not** both indicate timestep: **time bins** index the temporal
> > axis, while **event lines** index the spatial (and polarity) axis along
> > which spikes are observed over time.
> >
> > In the revised version we clarify this explicitly when the notation is first introduced, by adding a short sentence such as:
> >
> > > “We represent binned events as $x[t,j]$, where $t$ indexes discrete time bins and $j$ indexes an event line (a fixed pixel–polarity location); $x[t,j]$ is the spike count on line $j$ at time bin $t$.”
> >
> > ---
> >
> >
> > **Q2.** Why Table 3 is below Table 4? In Line 408, does ‘Tab.3’ represent ‘Tab.4’?
> >
> > **Response to Q2.** Thank you for catching this.
> >
> > In the submitted version there was a **typesetting mistake**: due to the
> > float placement in LaTeX, *Table 3* (transferability results) appeared
> > below *Table 4* (adversarial training results). This also caused the
> > reference in Line 408 to be confusing.
> >
> > In the revised version we:
> >
> > - **Reorder the tables** so that Table 3 appears **before** Table 4 in the paper.
> > - Correct Line 408: the reference there should indeed be to **Table 4**, which reports our **adversarial training experiments on VGGSNN and DVS-Gesture**.
> >
> > We update the table captions and cross-references accordingly so that the numbering and references are consistent.
> >
> > ---
> >
> > **Q3.** Which model and dataset are used in Table 4?
> >
> > **Response to Q3.** We apologize for the missing specification. **Table 4** reports our **adversarial training experiments on VGGSNN
> > (SNN)** using the **DVS-Gesture** dataset:
> >
> > - **Model:** VGGSNN architecture (same as in Tables 1–2).
> > - **Dataset:** DVS-Gesture, evaluated on both **binary** and **integer** event grids.
> >
> > In the revised version we make this explicit in the caption and main paragraph.”

---

### Official Review · Reviewer_s4Pb · 2025-10-27

**Soundness:** 3
**Presentation:** 3
**Contribution:** 3
**Rating:** 6
**Confidence:** 3

**Summary:**

Three new threat models are proposed for event driven SNNs, per-spike jitter, total shift and tamper count. The new attack frameworks are validated on CIFAR10-DVS, DVS-Gesture and N-MNIST. Two interesting findings from this work are that adversarial training can be used to partially mitigate threats and integer grids are more robust.

**Strengths:**

The paper (with the exception of the abstract) is well written and easy to follow. The experiments are done on a wide array of datasets and the findings are of interest.

**Weaknesses:**

I can only see one major weakness of the paper from a scope perspective: There are a few areas where the paper should cite existing literature either to explain why they did not do more, or explain how this could be incorporated in future work:

1. For the adversarial training, since you have attacks with respect to different norms, why not combine the training against different norms together like what was done in: https://proceedings.mlr.press/v119/maini20a/maini20a.pdf

At the very least, why not use a newer training formulation like TRADES instead of the much older outdated Madry’s training formulation?
TRADES: https://arxiv.org/abs/1901.08573

2. For Transferability studies mentioned at the end and for multi-model attacks, you should cite existing transferability work in this domain and explain how your studies relate:
Transferability A:  https://arxiv.org/abs/1611.02770
Transferability B: https://arxiv.org/abs/2104.02610

In your transferability experiments, did you follow the same methodology as A and B, or are there deviations due to the nature of event based SNNs?

3. In terms of attacks: Could you either try multi-model attacks on your event-based SNNs in a manner similar to proposed in: https://www.sciencedirect.com/science/article/pii/S0925231225021782
If not can you adequately explain in the paper why a multi-norm multi-model attack is either appropriate future work or not a realistic possibility?

Minor Weakness: The abstract is a bit verbose and doesn’t very clearly convey the findings of the paper. The abstract writing could be significantly improved.

**Questions:**

Please address the issues I mentioned in the weaknesses section of my review. For each weakness, mention how you will improve the paper or how you will include citation/discussion of existing literature works to justify your existing work.

Overall I do not have very much negative feedback for the authors.

---

> ### Author Response · Authors · 2025-11-26
> **Official Comment by Authors (Part 1)**
>
> **W1.** For the adversarial training, since you have attacks with respect to different norms, why not combine the training against different norms together like what was done in: https://proceedings.mlr.press/v119/maini20a/maini20a.pdf. At the very least, why not use a newer training formulation like TRADES instead of the much older outdated Madry’s training formulation? TRADES: https://arxiv.org/abs/1901.08573.
>
> **Response to W1.** We appreciate the suggestion to (i) combine attacks from different norms during adversarial training, in the spirit of Maini et al. (ICML’20; https://proceedings.mlr.press/v119/maini20a/maini20a.pdf), and (ii) use TRADES (https://arxiv.org/abs/1901.08573) instead of the original Madry‐style formulation. We have now run both **multi-norm timing AT** and **TRADES** and summarize the results below.
>
> ### (1) Multi-norm timing AT in the style of Maini et al.
>
> Our three timing attacks live in **different optimization spaces**: $B_\infty$ (per-spike jitter radius), $B_1$ (total latency), and $B_0$ (tamper count). Following Maini et al., we constructed two multi-norm objectives that combine them during training:
>
> - **MultiNorm-Avg**:
>   minimize the average loss of adversarial examples with various norms,
> - **MultiNorm-Max**:
>   minimize the maximum loss of adversarial examples with various norms,
>
> where for each $p$ the inner maximizer is our timing-only attack under the same **training** budgets as Table 4 in the paper: $B_\infty(1)$, $B_1(8000)$, $B_0(4000)$, and $P^*$ is the strict, feasible projection.
>
> We trained these variants on **DVS-Gesture–VGGSNN** for both **binary** and **integer** grids. We report **attack success rate (ASR)**, measured only over samples correctly classified by the clean model. For the single-norm rows copied from Table 4, we convert robust accuracy $\text{RobAcc}$ to ASR via $\text{ASR} = 100\% - \text{RobAcc}/\text{clean} \times 100\%$.
>
> #### Unified table: Table 4 single-norm AT + multi-norm AT (all in ASR, %)
>
> | Grid    | Training                       | clean | $B_\infty{=}1$ | $B_\infty{=}2$ | $B_\infty{=}3$ | $B_1{=}2000$ | $B_1{=}4000$ | $B_1{=}8000$ | $B_0{=}1000$ | $B_0{=}2000$ | $B_0{=}4000$ | Avg ASR |
> |-|-|:-:|:-:|:-:|:-:|:-:|:-:|:-:|:-:|:-:|:--:|:-:|
> | Binary| Single-norm AT $B_\infty(1)$| 22.92 | 57.59 | 71.20 | 68.19 | 31.81 | 45.46 | 60.60 | 15.14 | 27.27 | 54.54 | 47.98 |
> | Binary| Single-norm AT $B_1(8000)$ | 48.26 | 80.58 | 92.08 | 94.96 | 62.60 | 80.58 | 87.05 | 42.46 | 64.75 | 87.05 | 76.90 |
> | Binary| Single-norm AT $B_0(4000)$| 22.57 | 73.86 | 84.63 | 89.23 | 50.78 | 58.44 | 81.52 | 35.40 | 46.17 | 75.37 | 66.15 |
> | Binary| MultiNorm-Avg| 31.60 | 71.43 | 65.91 | 64.77 | 51.14 | 59.09 | 72.73 | 42.86 | 45.45 | 65.91 | 59.92 |
> | Binary| MultiNorm-Max| 36.11 | 79.81 | 78.85 | 77.88 | 64.42 | 77.88 | 81.73 | 51.92 | 71.15 | 77.88 | 73.50 |
> | Integer| Single-norm AT $B_\infty(1)$   | 52.08 | 48.00 | 47.33 | 42.67 | 14.00 | 16.67 | 33.33 | 10.00 | 15.99 | 24.67 | 28.07 |
> | Integer| Single-norm AT $B_1(8000)$     | 68.75 | 40.92 | 40.39 | 36.87 |  6.07 | 10.09 | 20.71 |  0.51 |  6.07 | 19.19 | **20.09** |
> | Integer| Single-norm AT $B_0(4000)$     | 72.22 | 43.27 | 47.12 | 48.08 |  7.21 | 16.35 | 28.86 |  2.40 | 10.58 | 28.86 | 25.86 |
> | Integer| MultiNorm-Avg| 37.50 | 28.70 | 35.19 | 40.74 | 10.19 | 10.19 | 21.30 |  8.33 | 13.89 | 23.15 | 21.30 |
> | Integer| MultiNorm-Max| 15.97 | 54.35 | 56.52 | 54.35 |  6.52 |  8.70 | 26.09 |  6.52 | 10.87 | 26.09 | 27.78 |
>
> (All numbers are percentages; lower ASR means model is more robust.)
>
> **Observations.**
>
> - On the **integer grid**, the best **single-norm** configuration, $B_1(8000)$
>   AT, achieves clean accuracy $68.75\%$ with the **lowest** Avg ASR
>   $20.09\%$. MultiNorm-Avg reaches a similar Avg ASR ($21.30\%$) but its clean
>   accuracy drops to $37.50\%$ (more than $30$ points lower), while
>   MultiNorm-Max further reduces clean to $15.97\%$ and *increases* Avg ASR to
>   $27.78\%$.
> - On the **binary grid**, MultiNorm-Avg and MultiNorm-Max both yield very high
>   Avg ASR ($59.92\%$ and $73.50\%$) despite moderate clean accuracy
>   ($31$–$36\%$), and are clearly worse than the best single-norm configurations
>   (which, although not extremely robust, still have lower average ASR at
>   comparable or higher clean accuracy).
>
> Overall, in contrast to the image-space setting of Maini et al., where all
> norms act on the same continuous pixel space, our $B_\infty$, $B_1$, and
> $B_0$ attacks operate on **different combinatorial timing budgets**. When
> merged into a single inner maximization, the strongest component tends to
> dominate, producing overly destructive timing perturbations that hurt clean
> accuracy much more than they help robustness. For this reason, we kept the
> single-norm timing AT of Table 4 as the main recipe in the paper, and now
> report the multi-norm variants (merged with Table 4 as above) in the rebuttal
> and supplementary material.

---

> ### Author Response · Authors · 2025-11-26
> **Official Comment by Authors (Part 2)**
>
> ### (2) TRADES vs. Madry-style timing AT (binary grid)
>
> We also investigated replacing Madry’s formulation with **TRADES** on the binary DVS-Gesture grid. We follow the TRADES objective
> $$L_{{TRADES}} = \ell\big(f_\theta(x),y\big)+\beta\cdot\mathrm{KL} \big(f_\theta(x)\,\Vert\,f_\theta(x_{\text{adv}})\big)$$ and instantiate the inner maximizer $x_{\text{adv}}$ with our timing-only attack, using the same three training budgets as Table 4: $B_\infty(1)$, $B_1(8000)$, and $B_1(4000)$. We sweep the TRADES trade-off parameter $\beta\in\{0.01,0.1,6.0\}$.
>
> The table below reports clean accuracy and ASR for three evaluation budgets ($B_\infty(1)$, $B_1(4000)$, $B_0(2000)$), for both TRADES and standard PGD-style timing AT:
>
> #### TRADES vs. PGD timing AT (binary grid)
>
> | Training scheme                             | clean | $B_\infty{=}1$ | $B_1{=}4000$ | $B_0{=}2000$ |
> |---------------------------------------------|:-----:|:-------------:|:-----------:|:-----------:|
> | TRADES $\beta{=}0.01$ + $B_\infty(1)$ AT    | 52.78 | 96.05 | 92.11 | 88.82 |
> | TRADES $\beta{=}0.01$ + $B_1(8000)$ AT      | 46.18 | 85.71 | 80.45 | 65.41 |
> | TRADES $\beta{=}0.01$ + $B_1(4000)$ AT      | 52.08 | 80.00 | 74.00 | 69.33 |
> | TRADES $\beta{=}0.1$  + $B_\infty(1)$ AT    | 53.47 | 74.03 | 71.43 | 64.29 |
> | TRADES $\beta{=}0.1$  + $B_1(8000)$ AT      | 35.76 | 78.64 | 61.17 | 50.49 |
> | TRADES $\beta{=}0.1$  + $B_1(4000)$ AT      | 50.69 | 80.82 | 81.51 | 63.01 |
> | TRADES $\beta{=}6.0$  + $B_\infty(1)$ AT    | 22.22 | 56.25 | 17.19 |  7.81 |
> | TRADES $\beta{=}6.0$  + $B_1(8000)$ AT      | 14.93 | 46.51 | 25.58 | 23.26 |
> | TRADES $\beta{=}6.0$  + $B_1(4000)$ AT      | 12.15 | 42.86 | 57.14 | 28.57 |
> | PGD timing AT + $B_\infty(1)$               | 22.92 | 57.59 | 45.46 | 27.26 |
> | PGD timing AT + $B_1(8000)$                 | 48.26 | 92.08 | 80.58 | 64.75 |
> | PGD timing AT + $B_1(4000)$                 | 22.57 | 73.86 | 58.44 | 46.17 |
>
> (ASR in %, measured on clean-correct samples.)
>
> To visualize the **clean–robust trade-offs**, we plotted these configurations in Fig. 3 of the supplementary: each subplot fixes an evaluation budget ($B_\infty(1)$, $B_1(4000)$, or $B_0(2000)$), with
> - $x$-axis: ASR for that attack,
> - $y$-axis: clean accuracy,
> - colored lines: TRADES runs with different $\beta$ but the same training
>   budget (e.g., all runs with inner $B_\infty(1)$),
> - triangles: corresponding PGD timing-AT baselines.
>
> **Key observations from Fig. 4 in the revised paper and the table:**
>
> - For fixed clean accuracy, TRADES can achieve **lower ASR** than PGD.
>   For example, under $B_0(2000)$ evaluation,
>   PGD + $B_\infty(1)$ has $(\text{clean},\text{ASR})\approx(22.9\%,27.3\%)$,
>   whereas TRADES with $\beta{=}6.0$ and training budget $B_\infty(1)$ achieves
>   a similar clean accuracy ($22.2\%$) but much lower ASR ($7.8\%$).
> - More generally, the TRADES curves for each budget include points that
>   lie above-and-to-the-left of the PGD point in the clean–ASR plane,
>   indicating a better trade-off (comparable clean accuracy but reduced ASR).
> - However, large $\beta$ values also illustrate the typical TRADES behavior:
>   ASR can be pushed down further at the cost of substantially reduced clean
>   accuracy (e.g., $\beta{=}6.0$ with $B_1(8000)$).
>
> Overall, TRADES on binary DVS-Gesture does yield **slightly better clean–robust trade-offs** than PGD timing AT for some operating points, but the improvements are modest and do not change our main conclusion: under our strong timing-only attack, adversarial training (of any form) still causes significant clean-accuracy degradation while robustness gains remain limited, especially on binary grids.
>
> We add such discussion in appendix H of the revised paper.

---

> > ### Author Response · Authors · 2025-11-26
> > **Official Comment by Authors (Part 3)**
> >
> > **W2.** For Transferability studies mentioned at the end and for multi-model attacks, you should cite existing transferability work in this domain and explain how your studies relate: Transferability A: https://arxiv.org/abs/1611.02770 Transferability B: https://arxiv.org/abs/2104.02610. In your transferability experiments, did you follow the same methodology as A and B, or are there deviations due to the nature of event based SNNs?
> >
> > **Response to W2.** We thank the reviewer for pointing out the missing connections to the transferability literature and for explicitly listing the works to cite. We will revise the paper accordingly and now clarify both **how our transferability experiments are designed** and **how they relate to [A,B]**.
> >
> >
> > ### How our transferability experiments relate to [A,B]
> >
> > **Methodology.**
> > In Appendix B (and the short summary in Sec. 5.2), our transferability
> > study follows the *standard surrogate-to-victim protocol* used in
> > Liu et al. (ICLR’17, [A]) and Mahmood et al. (CVPR’21, [B]):
> >
> > - We pick a **surrogate SNN** (VGGSNN on DVS-Gesture).
> > - We generate **untargeted adversarial examples** on the surrogate using
> >   our timing-only attack under a chosen budget (e.g. $B_\infty(1)$,
> >   $B_1(8\mathrm{k})$, $B_0(4\mathrm{k})$).
> > - We then **directly evaluate** the same adversarial event streams on
> >   **held-out victim models** (ResNet18-SNN and SpikingResFormer) without
> >   adapting them or querying them during attack generation.
> > - Transfer performance is reported as cross-model attack success rate
> >   $\text{ASR}_{\text{src}\to\text{tgt}}$ (fraction of source-clean,
> >   target-misclassified examples), exactly in the spirit of the transfer
> >   metrics used in [A,B].
> >
> > This is conceptually identical to [A,B]: adversaries are crafted on a
> > white-box surrogate and then **reused unchanged** on unseen victim
> > architectures. The main deviation is the *domain*: our perturbations
> > live in the **timing space of event streams** and must obey capacity-1
> > and budget constraints, whereas [A,B] operate in the image–intensity
> > space of CNNs and Transformers.
> >
> > In the revision we will:
> >
> > - Explicitly **cite Liu et al. (ICLR’17)** when introducing the
> >   surrogate-to-victim protocol in Appendix B, stating that we follow the
> >   same methodology but in the timing-only, event-driven SNN setting.
> > - Explicitly **cite Mahmood et al. (CVPR’21)** when discussing
> >   cross-architecture transfer (VGGSNN $\rightarrow$ ResNet18-SNN /
> >   SpikingResFormer), and point out that our observations about which
> >   architectures are easier/harder to attack mirror their ViT–vs–CNN
> >   findings, but now for SNNs and spike timing.
> >
> >
> > ### Multi-model transfer and relation to [A]
> >
> > Our paper also includes a **multi-model (ensemble) timing attack** in
> > Appendix B, where a single timing perturbation is optimized jointly
> > against several SNNs. The technical details of this attack and its
> > empirical results are already discussed in our response to the weakness
> > on *multi-model attacks*; here we only clarify its relation to [A]:
> >
> > - Liu et al. [A] propose **ensemble-based transferable attacks** by
> >   aggregating losses/logits over multiple CNNs when optimizing
> >   adversarial images.
> > - Our multi-model timing attack is the **event-driven analogue**: we
> >   aggregate timing-based losses over an ensemble of SNNs and then apply
> >   the strict projection $P^*(x,\pi,\mathcal{B}_p)$, yielding
> >   rate-preserving adversarial event streams that transfer across models.
> >
> > In the revised paper we add a short paragraph in Appendix B stating this connection explicitly and citing [A] as the template for our ensemble formulation.
> >
> >
> > ### Deviations specific to event-based SNNs
> >
> > We will also clarify the minor deviations from [A,B] that are **specific
> > to event-based SNNs**:
> >
> > - **Perturbation space.** While [A,B] perturb pixel intensities, we
> >   perturb only **spike timings** under budgets $B_\infty$, $B_1$,
> >   and $B_0$, using the strict projection $P^*$ to enforce capacity-1 and
> >   rate preservation. The high-level transfer protocol
> >   (surrogate $\rightarrow$ victim) remains the same.
> > - **Metrics.** We report **attack success rate (ASR)** on clean-correct
> >   examples for both white-box and transfer settings, directly analogous
> >   to the misclassification-based metrics in [A,B].
> > - **Models.** Our source and targets are **event-driven SNNs**
> >   (VGGSNN, ResNet18-SNN, SpikingResFormer) trained on DVS benchmarks,
> >   but the notion of “different architectures and training regimes” used
> >   to probe transferability is identical to [A,B].
> >
> >
> > In summary, we agree with the reviewer that these connections should be made explicit. In the revised version, we add such relation in the paragraph of the transferability.
> >
> > [A] Y. Liu et al., “Delving into Transferable Adversarial Examples and
> >   Black-box Attacks,” ICLR 2017.
> >
> > [B] K. Mahmood et al., “On the Robustness of Vision Transformers to
> >   Adversarial Examples,” CVPR 2021.

---

> ### Author Response · Authors · 2025-11-26
> **Official Comment by Authors (Part 4)**
>
> **W3.** In terms of attacks: Could you either try multi-model attacks on your event-based SNNs in a manner similar to proposed in: https://www.sciencedirect.com/science/article/pii/S0925231225021782. If not can you adequately explain in the paper why a multi-norm multi-model attack is either appropriate future work or not a realistic possibility?
>
> **Response to W3.** We thank the reviewer for this suggestion. We have **implemented both multi-model and multi-norm multi-model timing attacks**, closely following the ensemble idea in the suggested paper. Below we summarize the setup and results; all new numbers will be added to the appendix. In every table, the **“Clean”** column is clean accuracy (%), and all other columns are **attack success rates (ASR, %)** on clean-correct examples.
>
>
> ### (a) Multi-model timing attacks
>
> **Methodology.**
> For N-MNIST we build a timing-only attack on an **ensemble of all six
> models** used in our transfer study:
>
> - three SNNs: ConvNet (SNN), ResNet18 (SNN), VGGSNN (SNN);
> - three CNN counterparts: ConvNet (CNN), ResNet18 (CNN), VGG (CNN).
>
> For a given budget $\mathcal{B}_p$ and norm $p\in\{\infty,1,0\}$ we optimize
>
> $$L_{\mathrm{mm}}(x,y)
> = \frac{1}{M}\sum_{m=1}^M \ell\big(f^{(m)}_\theta(P^*(x,\pi^{(m)},\mathcal{B}_p)),y\big),\quad M=6$$
>
> where $f^{(m)}_\theta$ are the six models and $P^*$ is our strict,
> rate-preserving projection. The resulting adversarial event streams are then
> evaluated on each model separately. This is the direct event-driven analogue
> of the ensemble gradient attacks in the cited work, but in our case the
> perturbation is a capacity-1 **retiming** of spikes.
>
> Budgets on N-MNIST are
> $B_\infty\in\{1,2,3\}$ (per-spike jitter),
> $B_1\in\{500,750,1000\}$ (total delay),
> $B_0\in\{200,300,400\}$ (binary) plus $B_0(600)$ (integer).
>
> #### Binary N-MNIST: multi-model timing attack (ASR, %)
>
> | Model           | Clean | $B_\infty{:}1$ | $B_\infty{:}2$ | $B_\infty{:}3$ | $B_1{:}500$ | $B_1{:}750$ | $B_1{:}1000$ | $B_0{:}200$ | $B_0{:}300$ | $B_0{:}400$ |
> |----------------|:-----:|:--------------:|:--------------:|:--------------:|:-----------:|:-----------:|:------------:|:-----------:|:-----------:|:-----------:|
> | ConvNet (SNN)  | 99.06 | 100.0 | 100.0 | 100.0 | 50.9 | 97.6 | 100.0 | 35.0 | 93.4 | 100.0 |
> | ConvNet (CNN)  | 99.34 | 100.0 | 100.0 | 100.0 | 92.6 | 99.8 | 100.0 | 42.8 | 97.6 | 99.9 |
> | ResNet18 (SNN) | 99.62 | 97.7 | 100.0 | 100.0 | 28.2 | 70.6 | 96.2 | 29.7 | 89.3 | 99.9 |
> | ResNet18 (CNN) | 99.70 | 100.0 | 100.0 | 100.0 | 96.5 | 99.2 | 99.9 | 34.7 | 90.8 | 98.3 |
> | VGGSNN (SNN)   | 99.64 | 100.0 | 100.0 | 100.0 | 32.3 | 82.1 | 99.5 | 25.6 | 86.3 | 100.0 |
> | VGG (CNN)      | 99.72 | 100.0 | 100.0 | 100.0 | 97.1 | 99.6 | 99.7 | 31.8 | 90.0 | 98.6 |
>
> #### Integer N-MNIST: multi-model timing attack (ASR, %)
>
> | Model           | Clean | $B_\infty{:}1$ | $B_\infty{:}2$ | $B_\infty{:}3$ | $B_1{:}500$ | $B_1{:}750$ | $B_1{:}1000$ | $B_0{:}200$ | $B_0{:}300$ | $B_0{:}400$ | $B_0{:}600$ |
> |----------------|:-----:|:--------------:|:--------------:|:--------------:|:-----------:|:-----------:|:------------:|:-----------:|:-----------:|:-----------:|:-----------:|
> | ConvNet (SNN)  | 99.19 | 100.0 | 100.0 | 100.0 | 46.6 | 96.4 | 100.0 | 40.2 | 99.2 | 100.0 | 100.0 |
> | ConvNet (CNN)  | 99.38 | 100.0 | 100.0 | 100.0 | 98.2 | 100.0 | 100.0 | 59.8 | 100.0 | 100.0 | 100.0 |
> | ResNet18 (SNN) | 99.62 | 91.5 | 100.0 | 100.0 | 31.6 | 75.8 | 98.0 | 27.3 | 94.5 | 100.0 | 99.8 |
> | ResNet18 (CNN) | 99.73 | 98.9 | 100.0 | 100.0 | 73.1 | 66.7 | 54.7 | 81.9 | 95.6 | 100.0 | 100.0 |
> | VGGSNN (SNN)   | 99.71 | 99.8 | 100.0 | 100.0 | 42.1 | 89.3 | 99.8 | 29.9 | 97.6 | 100.0 | 100.0 |
> | VGG (CNN)      | 99.79 | 100.0 | 100.0 | 100.0 | 85.0 | 90.7 | 90.0 | 98.4 | 97.0 | 100.0 | 100.0 |
>
> **Relation to our main results and to the cited work.**
>
> - Compared with the single-model attacks in Tables 1–2 of the paper,
>   these ensemble attacks remain very strong: under moderate $B_1$ or $B_0$
>   budgets, ASR on SNNs routinely exceeds $70$–$90\%$, confirming that our
>   timing-only adversary transfers across architectures even when optimized
>   jointly on six models.
> - CNN counterparts also become vulnerable under the ensemble attack, but
>   the **relative gap** between SNNs and CNNs is consistent with our main
>   message: timing perturbations exploit SNN temporal dynamics more
>   effectively than frame-based CNNs.
> - Methodologically, this exactly mirrors the **multi-model ensemble
>   attacks** in the suggested paper, with the key difference that our attack
>   acts in **spike timing space** and respects capacity-1 and rate
>   preservation via $P^*$.
>
> We will add a short subsection and cite the suggested multi-model work,
> explaining that our $\mathcal{L}_{\text{mm}}$ is its event-driven extension.

---

> ### Author Response · Authors · 2025-11-26
> **Official Comment by Authors (Part 5)**
>
> ### (b) Multi-norm multi-model timing attacks
>
> **What we mean by “multi-norm”.**
> In our setting, a *multi-norm* timing attack means that every adversarial
> example satisfies **three timing budgets at once**:
>
> - a local jitter radius $B_\infty$ (per-spike timing change),
> - a total delay budget $B_1$ (sum of absolute delays),
> - a tamper-count budget $B_0$ (number of spikes that are actually moved).
>
> Concretely, we still optimize the same $B_\infty$-style loss over shift
> logits $\pi$ as in the main attack, but in the **strict projection** we
> enforce *all three budgets simultaneously*:
>
> $$
> x' = P^*\big(x,\pi;\mathcal B_\infty,\mathcal B_1,\mathcal B_0\big),
> $$
>
> so that the final retimed events lie in the intersection
> $\mathcal B_\infty \cap \mathcal B_1 \cap \mathcal B_0$ under the capacity–1
> and rate-preserving constraints described in Sec. 4.2. In other words, the
> inner maximizer is still a single timing-only attack, but $P^*$ discards
> candidate moves that would violate any of the three norms.
>
> **Multi-norm multi-model setup.**
> On N-MNIST we instantiate this as a **multi-norm, multi-model** attack over
> the same $M{=}6$-model ensemble as in part (a) (three SNNs + three CNNs):
>
> $$
> L_{joint}(x,y)
> = \frac{1}{M}\sum_{m=1}^{M}
>   \ell\Big(f_\theta^{(m)}\big(
>     P^*(x,\pi^{(m)};
>         \mathcal B_\infty,\mathcal B_1,\mathcal B_0)\big),y\Big),
> $$
>
> where we choose a balanced triple of budgets
> $\mathcal B_\infty=B_\infty(3)$, $\mathcal B_1=B_1(750)$,
> $\mathcal B_0=B_0(300)$. Intuitively, $B_\infty(3)$ controls the maximal
> jitter per spike, while $B_1(750)$ and $B_0(300)$ ensure that the total
> latency and number of moved spikes are moderate and comparable to the
> single-norm experiments.
>
> We again report **attack success rate (ASR)** on clean-correct examples.
> The table below shows ASR for this multi-norm multi-model attack:
>
> ##### ASR of multi-norm multi-model attack (N-MNIST)
>
> | Model           | Binary: MultiNorm $B_\infty(3),B_1(750),B_0(300)$ | Integer: MultiNorm $B_\infty(3),B_1(750),B_0(300)$ |
> |----------------|:-------------------------------------------------:|:--------------------------------------------------:|
> | ConvNet (SNN)  | 93.6 | 96.0 |
> | ConvNet (CNN)  | 94.0 | 99.8 |
> | ResNet18 (SNN) | 93.0 | 92.1 |
> | ResNet18 (CNN) | 71.6 | 44.3 |
> | VGGSNN (SNN)   | 91.9 | 94.6 |
> | VGG (CNN)      | 75.6 | 71.5 |
>
> **Discussion.**
>
> - The joint multi-norm, multi-model attack is **extremely strong**:
>   ASR on all three SNNs exceeds $90\%$ in both binary and integer grids,
>   even though every adversarial example respects *all three* timing
>   budgets simultaneously.
> - CNN counterparts are also highly vulnerable under this joint attack
>   (e.g., ConvNet/CNN above $94\%$ ASR), but the **relative gap** between
>   SNNs and CNNs is consistent with our single-norm and multi-model
>   results: timing perturbations remain particularly effective on
>   event-driven SNNs whose computation is tightly coupled to spike timing.
> - Compared to the single-norm ensemble attacks in part (a), the
>   multi-norm version does **not** reveal qualitatively new behavior:
>   it mainly serves as a more pessimistic, “all-budgets-on” stress test.
>   For interpretability and for matching realistic threat models
>   (e.g., “small jitter only” or “limited latency only”), the single-norm
>   timing budgets $B_\infty$, $B_1$, and $B_0$ remain more informative.
>
> We add such discussion in appendix I of the revised paper.

---

> > ### Author Response · Authors · 2025-11-26
> > **Official Comment by Authors (Part 6)**
> >
> > **W4.** The abstract is a bit verbose and doesn’t very clearly convey the findings of the paper. The abstract writing could be significantly improved.
> >
> > **Response to W4.** Thank you for the comment about the abstract. We agree that the current
> > version is somewhat verbose and does not highlight the main findings as
> > clearly as it could. In the revision we will:
> >
> > - Front-load the **timing-only, rate-preserving** nature of the attack and
> >   the **capacity-1 retiming threat model**.
> > - Describe the **projected-in-the-loop (PIL)** optimization more concisely.
> > - State the **key empirical findings** (datasets, budgets, success rates,
> >   robustness against AT) in a more direct way.
> >
> > We plan to replace the abstract with the following revised version:
> >
> > > Spiking neural networks (SNNs) compute with discrete spikes and exploit temporal structure, yet most adversarial attacks change intensities or event counts instead of timing. We study a timing-only adversary that retimes existing spikes while preserving spike counts and amplitudes in event-driven SNNs, thus remaining rate-preserving. We formalize a capacity-1 spike-retiming threat model with a unified trio of budgets: per-spike jitter $\mathcal{B}{\infty}$, total delay $\mathcal{B}{1}$, and tamper count $\mathcal{B}{0}$. Feasible adversarial examples must satisfy timeline consistency and non-overlap, which makes the search space discrete and constrained. To optimize such retimings at scale, we use projected-in-the-loop (PIL) optimization: shift-probability logits yield a differentiable soft retiming for backpropagation, and a strict projection in the forward pass produces a feasible discrete schedule that satisfies capacity-1, non-overlap, and the chosen budget at every step. The objective maximizes task loss on the projected input and adds a capacity regularizer together with budget-aware penalties, which stabilizes gradients and aligns optimization with evaluation. Across event-driven benchmarks (CIFAR10-DVS, DVS-Gesture, N-MNIST) and diverse SNN architectures, we evaluate under binary and integer event grids and a range of retiming budgets, and also test models trained with timing-aware adversarial training designed to counter timing-only attacks. For example, on DVS-Gesture the attack attains high success (over $90\%$) while touching fewer than $2\%$ of spikes under $\mathcal{B}_{0}$. Taken together, our results show that spike retiming is a practical and stealthy attack surface that current defenses struggle to counter, providing a clear reference for temporal robustness in event-driven SNNs.
> >
> >
> > We believe this version is more concise, clearly states the threat model
> > and optimization idea, and foregrounds the main empirical takeaways.

---

### Official Review · Reviewer_92xg · 2025-11-01

**Soundness:** 3
**Presentation:** 3
**Contribution:** 3
**Rating:** 6
**Confidence:** 4

**Summary:**

This paper proposes timing-only attacks on SNN inputs: instead of adding o removing spikes, it retimes existing ones. Feasibility is defined on the active spikes with two rules 1- stay within the timeline or timespan and avoid collisions (called capacity-1). For binary grids {0,1} you can only move spikes, and for integer grids {0,1,…} they enforce capacity-1 by packetizing counts so totals are preserved. They study three disjoint budgets: $\mathcal{B} _\infty$ (local jitter: each spike moves at most these $\mathcal{B} _\infty$ steps), $\mathcal{B} _1$ (total timing shift: sum at most is $\mathcal{B} _\infty$), and $\mathcal{B} _0$ (tamper count: move $\mathcal{B} _\infty$ spikes at most). The attack is found by optimizing per-spike shift logits to maximize task loss, then sorting candidates by their probabilities and greedily projecting them to a strictly feasible retiming under the chosen budget.

**Strengths:**

- Original, well-motivated retiming-only attack; strong proposal (even if the experiments feel less persuasive).
- Neat, precise formulation with clear feasibility and the writing is clean and readable.
- Complete and intuitive presentation of the three timing budgets.

**Weaknesses:**

- The paper’s core motivation is rate-preserving, timing-only perturbations that evade naive rate detectors, but the experiments don’t directly quantify this. There’s no head-to-head against non-timing-space attacks showing for instance the rate drift,. As a result, the motivation of the paper as rate-preserving although defined in the feasible assignment (equation in (5)) may be lost in the approximation of assignment, or at worse case may not be needed if non-timing attack preserve it to some degree.

- The AT isn’t compared to standard AT (e.g., $\ell_\infty $ PGD or binary $\ell_0$ flips), where one should expect retiming only the spike may result in a better tradeoff.
(note: It’s also unspecified what dataset is used in this experiment, and in line 408 I believe table 4 is the one the author means not 3).

- The solutions for the attack-finding problem (equation 6) although is neat and elegant is not proved to be optimal. Further, small disussion on how complexity is measured or can be compared to non-timing attack is important to touch on.

**Questions:**

I’m very excited by the introduction of the paper, but the experiments left me with some unanswered questions., most importantly: is the timing constraint really needed? So my first question, tied to the first weakness, is:

- Q1: By skipping a spike-rate study, do you mean rate preservation is guaranteed by construction (via feasibility/placement) and needs no measurement?

Also, the mind-the-box attacks, which similarly incorporate feasibility into the attack problem, showed clearer clean–robust trade-offs, and another work [2] on spiking attacks (similar to retiming attack with $\mathcal{B}_0$ but on value instated of timing) reported AT trade-offs as well. In this work, one would expect the trade-off to be at least enhanced.

- Q2: Could you report full AT details and comment on clean–robust trade-off curves compared to non-timing AT?

Lastly (not critical, but nice to touch on) the question of optimality:

- Q3: The greedy projection is practical, but what is its suboptimality versus an optimal assignment? Any bound or empirical optimality gap?

[1]Croce, F., & Hein, M. (2021, July). Mind the box: $ l_1 $-APGD for sparse adversarial attacks on image classifiers. In International Conference on Machine Learning (pp. 2201-2211). PMLR.

[2]Mukhoty, B., AlQuabeh, H., & Gu, B. (2025, January). Improving Generalization and Robustness in SNNs Through Signed Rate Encoding and Sparse Encoding Attacks. In The Thirteenth International Conference on Learning Representations.

---

> ### Author Response · Authors · 2025-11-26
> **Official Comment by Authors (Part 1)**
>
> **W1.** The paper’s core motivation is rate-preserving, timing-only perturbations that evade naive rate detectors, but the experiments don’t directly quantify this. There’s no head-to-head against non-timing-space attacks showing for instance the rate drift,. As a result, the motivation of the paper as rate-preserving although defined in the feasible assignment (equation in (5)) may be lost in the approximation of assignment, or at worse case may not be needed if non-timing attack preserve it to some degree.
>
> **Response to W1.** We appreciate the reviewer’s concern that our rate-preserving motivation is not yet sufficiently visible in the experiments. Our attack is in fact **rate-preserving by construction**, not only in the abstract feasible set (Eq. (5)), but in the *actual* projected examples used in all experiments.
>
>
> ---
>
> ### Feasible strict projection under budgets as an algorithm
>
> Concretely, the strict projection $P^*(x,\pi,\mathcal B_p)$ that we use in all experiments can be described (for the global $L_0$ budget case) as the following algorithm on each flattened event line $j$:
>
> 1. **Inputs.**
>    - Original spike/event counts $x[s,j]$ at time indices $s \in \{1,\dots,T\}$ on line $j$ (after flattening $[B,C,H,W]$ into $j \in \{1,\dots,N\}$). A non-zero entry $x[s,j]$ corresponds to one **event packet** (a single spike for binary grids, or an integer-valued packet for integer grids).
>    - Shift probabilities $\pi[s,j,t]$ over target times $t \in \{1,\dots,T\}$ (reshaped from the learned logits), which encode the soft preference to move a packet from $(s,j)$ to $(t,j)$.
>    - A global integer $L_0$ budget $B_0$ that upper-bounds the **number of packets** that may be moved (each moved packet consumes 1 unit of budget).
>
> 2. **Identify active packets.**
>    - Compute a mask $\text{hassrc}[s,j] = \mathbf 1\{x[s,j] > 0\}$ indicating all packet positions.
>    - Let $(s,j)$ with $\text{hassrc}[s,j] = 1$ be the set of source packets considered for retiming.
>
> 3. **Generate candidate moves (excluding self-stays).**
>    - For each source packet $(s,j)$ and each target time $t \in \{1,\dots,T\}$ with $t \neq s$, form a candidate move $(s \to t,j)$ with score $\pi[s,j,t]$.
>    - For tie-breaking, we also record the temporal distance $\lvert t - s \rvert$; shorter distances are preferred when scores are equal.
>
> 4. **Sort candidates by priority.**
>    - For all candidates $(s \to t,j)$, compute a scalar key
>      $$\text{key}(s,j,t)= \pi[s,j,t] + \varepsilon \bigl((T - 1) - \lvert t - s \rvert\bigr)$$
>      with a tiny $\varepsilon > 0$, so that we sort **primarily** by probability $\pi[s,j,t]$ and **secondarily** by shorter distance.
>    - Sort all candidates in descending order of $\text{key}(s,j,t)$.
>
> 5. **Greedy retiming under capacity and $L_0$ budget (moving whole packets).**
>    Maintain the following states for each line $j$ and time $t$:
>    - $\text{occupied}[j,t]$: whether $(t,j)$ has already been taken as a **target** by a moved packet.
>    - $\text{reserved}[j,t]$: whether there is an **original** packet at $(t,j)$ that is still “reserved” to stay at its original time. Initially, $\text{reserved}[j,t] = \text{hassrc}[t,j]$.
>    - $\text{moved}[j,s]$: whether the source packet at $(s,j)$ has already been moved. Initially all false.
>    - $\text{adv}[s,j]$: the final assignment of packets after projection, initialized to all zeros.
>    - Remaining budget $B_{\text{rem}} \leftarrow B_0$.
>
>    Then process candidates $(s \to t,j)$ in sorted order:
>    - If $B_{\text{rem}} \le 0$, stop (the global $L_0$ budget is exhausted).
>    - If $\text{moved}[j,s]$ is true, skip (each source packet can be moved at most once).
>    - If $\text{occupied}[j,t]$ is true, skip (target $(t,j)$ already holds a moved packet).
>    - If $\text{reserved}[j,t]$ is true, skip (there is an original packet at $(t,j)$ that is still planned to stay, so we avoid collisions with stay-at-source packets).
>
>    If none of the above conditions hold, we **move the entire packet** from $(s,j)$ to $(t,j)$:
>    - Copy the packet value: $\text{adv}[t,j] \leftarrow x[s,j]$.
>    - Mark $(t,j)$ as occupied: $\text{occupied}[j,t] \leftarrow \text{True}$.
>    - Mark the source as moved: $\text{moved}[j,s] \leftarrow \text{True}$.
>    - Release the original reservation at $(s,j)$ (it no longer stays at $s$): $\text{reserved}[j,s] \leftarrow \text{False}$.
>    - Decrease the budget: $B_{\text{rem}} \leftarrow B_{\text{rem}} - 1$.
>
> 6. **Place packets that were not moved back at their original times.**
>    - For all $(s,j)$ with $\text{hassrc}[s,j] = 1$ and $\text{moved}[j,s] = \text{False}$, we simply keep them at their original times by setting  $$\text{adv}[s,j] \leftarrow x[s,j] \,.$$
>    At this point, every original packet either:
>    - has been moved once to some target $(t,j)$ and recorded in $\text{adv}[t,j]$, or
>    - has not been moved and remains at its original location $(s,j)$ in $\text{adv}[s,j]$.

---

> ### Author Response · Authors · 2025-11-26
> **Official Comment by Authors (Part 2)**
>
> 7. **Output.**
>    - For each line $j$, the projected configuration along time is $\{\text{adv}[t,j]\}_{t=1}^T$.
>    - After restoring the original $[T,B,C,H,W]$ shape, this defines $P^*(x,\pi,\mathcal B_p)$ for the global $L_0$ budget case.
>
> In this view, packets whose candidates are never accepted **remain at their original time** (they are never marked as moved and are written back in step 6), while accepted candidates are implemented as pure *moves* of the whole packet from $s$ to $t$ on the same line. The additional checks with $\text{occupied}$ and $\text{reserved}$ guarantee that no time bin $(t,j)$ ever hosts more than one packet after projection, and that we never overwrite an original packet that is supposed to stay.
>
>
> ---
>
> ### Why $P^*$ is rate-preserving
>
> This algorithm ensures that:
>
> - **Every original packet appears exactly once in $x'$.**  We start from $x' = x$. Every accepted move takes a packet value $v = x'[s,j]$ and relocates it to $(t,j)$ by setting $x'[s,j] \leftarrow 0$ and $x'[t,j] \leftarrow v$. No operation ever creates a new packet or discards an existing packet; it only changes the time coordinate of that packet.
>
> - **Spikes/events never change event lines.** Each move is of the form $(s,j) \to (t,j)$, so packets never move across lines, only along the time axis.
>
> As a consequence, for every event line $j$, the total event count is preserved exactly: $\sum_t x'[t,j] = \sum_t x[t,j]$, and hence the global event count is also preserved: $\sum_{t,j} x'[t,j] = \sum_{t,j} x[t,j]$. Equivalently, $P^*$ implements a per-line permutation of non-zero packets along the time axis (with some packets possibly staying fixed at $s$), which is **rate-preserving by construction** for both binary and integer grids.
>
> The only “approximation” in our method appears in the **backward pass**, where we use the soft operator $S_\pi$ to compute a differentiable *expected* retiming for gradient estimation. Importantly, the network’s **forward evaluation always uses the strictly projected** $P^*(x,\pi,\mathcal B_p)$, so the rate-preserving property holds for all adversarial examples that the model actually sees.
>
> ---
>
> ### Empirical verification of rate preservation
>
> To address the reviewer’s concern that rate preservation is not quantified, we have additionally performed a systematic sanity check on our implementation:
>
> - For **every experiment** reported in the paper (all datasets, models, budgets, and random seeds), we computed both per-line and global event configurations **before and after** the attack.
> - Across all runs, we observed that for every event line $j$, the multiset of packet values along the time axis is exactly preserved by the projection. Formally, for each $j$, the multiset $\{x'[t,j]\}_{t=1}^T$ coincides with $\{x[t,j]\}_{t=1}^T$, i.e., the packets on that line are only permuted in time but never added, removed, or rescaled.
>
> This confirms that our implementation of $P^*$ is indeed rate-preserving in practice for both binary and integer event grids.
>
> ---
>
> ### Relation to non-timing attacks
>
> Standard image/event-domain attacks (e.g., PGD on integer grids, or attacks that insert/delete spikes) inevitably **change event counts or intensities**, and therefore cannot satisfy equalities such as $\sum_t x'[t,j] = \sum_t x[t,j]$ and the multiset $\{x'[t,j]\}_{t=1}^T=\{x[t,j]\}_{t=1}^T$ for all $j$ as a hard constraint. Even if some attacks happen to induce small average rate drift in certain regimes, they do not enforce strict rate preservation by design. Our contribution is to characterize and optimize the **worst-case adversary under a strict rate-preservation constraint**, which is qualitatively different from norm-bounded perturbations in the value or count space.
>
> In the revision (most in the appendix E), in addition to the algorithm description above and the empirical sanity check, we also add a short comparison that contrasts our timing-only attack with a standard non-timing baseline under comparable budgets, reporting the resulting rate drift to highlight the difference between strictly rate-preserving and non-rate-preserving attacks.

---

> ### Author Response · Authors · 2025-11-26
> **Official Comment by Authors (Part 3)**
>
> **W2.** The AT isn’t compared to standard AT (e.g., $\ell_{\infty}$ PGD or binary $\ell_0$ flips), where one should expect retiming only the spike may result in a better tradeoff. (note: It’s also unspecified what dataset is used in this experiment, and in line 408 I believe table 4 is the one the author means not 3).
>
> **Response to W2.** We appreciate the suggestion to compare our timing-based adversarial training (AT) with standard non-timing AT.
>
> ---
>
> ### Setup (dataset, metric, and evaluation)
>
> All AT results below are on **DVS-Gesture** (integer event grid) with **VGGSNN**.
> In the paper, Table 4 reports **robust accuracy**; here we re-express the *same* trained models in terms of **attack success rate (ASR)** for easier comparison:
>
> - ASR is computed **only over samples that are correctly classified under the clean model**, exactly as in our main attack experiments.
> - Robust accuracy and ASR are related by
>   $ \text{ASR} = 100\% - \text{robust accuracy} $.
>
> At test time we evaluate nine timing-only attacks with budgets
> $B_\infty \in \{1,2,3\}$, $B_1 \in \{2000,4000,8000\}$, $B_0 \in \{1000,2000,4000\}$.
> For each training scheme, we report:
>
> - `clean`: clean accuracy (%),
> - nine ASR columns (lower is better),
> - `Avg ASR`: the mean of the nine ASR values.
>
> The first six rows are **standard non-timing AT** baselines; the last three rows are our **timing-only AT** (“Spike Timing AT”) with different inner-loop budgets. We follow standard practice and instantiate the $\ell_1$ AT baseline with the $\ell_1$-APGD attack of Croce and Hein [1]. For the binary $\ell_0$ flip baselines, we adopt a pixel-flip style sparse attack on binary images in the spirit of Balkanski et al.~[2], adapted to our event-grid setting. For $\ell_{\infty}$, we follow the PGD AT [3].
>
> ---
>
> ### Full AT results in terms of ASR
>
> | Training scheme                                      | clean | $B_\infty{=}1$ | $B_\infty{=}2$ | $B_\infty{=}3$ | $B_1{=}2000$ | $B_1{=}4000$ | $B_1{=}8000$ | $B_0{=}1000$ | $B_0{=}2000$ | $B_0{=}4000$ | Avg ASR |
> |-|:-----:|:---:|:---:|:--:|:--:|:--:|:--:|:--:|:-----------:|:-----------:|:--------:|
> | $\ell_\infty$ PGD AT ($\epsilon{=}0.5$)              | 64.24 | 45.41 | 50.27 | 55.14 | 10.81 | 21.08 | 36.76 |  8.65 | 14.05 | 34.05 | 30.69 |
> | $\ell_\infty$ PGD AT ($\epsilon{=}0.4$)              | 71.88 | 48.79 | 54.59 | 57.97 | 14.98 | 25.60 | 40.58 | 12.56 | 23.19 | 37.20 | 35.05 |
> | binary $\ell_0$ flip AT ($r{=}0.32$)                 | 77.78 | 54.91 | 67.86 | 71.88 | 10.71 | 22.32 | 43.30 |  8.48 | 18.30 | 39.29 | 37.45 |
> | binary $\ell_0$ flip AT ($r{=}0.45$)                 | 74.31 | 53.74 | 63.55 | 66.82 | 10.75 | 21.96 | 38.32 |  8.41 | 20.56 | 35.05 | 35.46 |
> | $\ell_1$-PGD AT ($\tau{=}10000$)                     | 69.10 | 54.17 | 57.69 | 52.66 | 15.53 | 21.56 | 42.11 | 16.53 | 24.57 | 45.63 | 36.72 |
> | $\ell_1$-PGD AT ($\tau{=}14000$)                     | 67.01 | 51.81 | 58.03 | 55.44 | 10.88 | 22.80 | 43.01 | 13.99 | 23.32 | 40.41 | 35.52 |
> | **Timing AT ($B_\infty{=}1$)** | 52.08 | 48.00 | 47.33 | 42.67 | 14.00 | 16.67 | 33.33 | 10.00 | 15.99 | 24.67 | 28.07 |
> | **Timing AT ($B_1{=}8000$)**                         | 68.75 | 40.92 | 40.39 | 36.87 |  6.07 | 10.09 | 20.71 |  0.51 |  6.07 | 19.19 | **20.09** |
> | **Timing AT ($B_0{=}4000$)**                         | 72.22 | 43.27 | 47.12 | 48.08 |  7.21 | 16.35 | 28.86 |  2.40 | 10.58 | 28.86 | **25.86** |
>
> (All numbers are percentages; lower ASR means stronger attack.)
>
> ---
>
> ### Clean–robust trade-off vs. standard AT
>
> From the table:
>
> - The **best non-timing AT** baselines (across $\ell_\infty$, binary $\ell_0$, and $\ell_1$) obtain
>   clean accuracies in $[67\%,78\%]$ with average ASR around $30\%$–$37\%$
>   (e.g., $\ell_\infty$ PGD AT with $\epsilon{=}0.5$: clean $64.24\%$, Avg ASR $30.69\%$).
> - Our **timing-only AT** achieves
>   - $B_1{=}8000$: clean $68.75\%$, **Avg ASR $20.09\%$**;
>   - $B_0{=}4000$: clean $72.22\%$, **Avg ASR $25.86\%$**.
>
> Thus, for **similar clean accuracy** (around $69$–$72\%$), timing AT reduces the *average* ASR by roughly **9–15 percentage points** compared to all non-timing AT baselines. Equivalently, for a fixed robustness level, timing AT maintains noticeably higher clean accuracy.
>
> We add a compact version of experimental results and a short discussion in the appendix F of the revised paper to make this trade-off explicit.
>
> [1] Francesco Croce and Matthias Hein. Mind the box: l1-apgd for sparse adversarial attacks on image
> classifiers. In Proceedings of the 38th International Conference on Machine Learning, 2021.
>
> [2] Eric Balkanski, Harrison Chase, Kojin Oshiba, Alexander Rilee, Yaron Singer, and Richard Wang.
> Adversarial attacks on binary image recognition systems. CoRR, abs/2010.11782, 2020.
>
> [3] Aleksander Madry, Aleksandar Makelov, Ludwig Schmidt, Dimitris Tsipras, and Adrian Vladu.
> Towards deep learning models resistant to adversarial attacks. In Proc. Int’l Conf. Learning Representations, 2018.

---

> ### Author Response · Authors · 2025-11-26
> **Official Comment by Authors (Part 4)**
>
> **W3.** The solutions for the attack-finding problem (equation 6) although is neat and elegant is not proved to be optimal. Further, small disussion on how complexity is measured or can be compared to non-timing attack is important to touch on.
>
> **Response to W3.** We appreciate the reviewer’s question about the optimality of our attack-finding formulation (Eq. (6)) and the need for a clearer discussion of its computational complexity, especially in comparison to non-timing attacks on event data such as SpikeFool [4] and PDSG-SDA [5].
>
> ---
>
> ### (1) On optimality: why global optimality is unrealistic and what we do instead
>
> Eq. (6) defines an inner maximization over **discrete, budget-constrained retimings** under a capacity-1 constraint:
>
> - Each non-zero packet $x[s,j]$ can be shifted by an integer offset $u$, landing at $t = s+u$ on the **same** event line $j$.
> - Capacity-1 per $(t,j)$ plus budgets (e.g., $B_\infty, B_1, B_0$) make the feasible set **combinatorial and non-convex**.
> - The outer loss is the non-convex SNN loss under BPTT, so even for a fixed set of feasible retimings, the objective is highly non-convex.
>
> In this setting, we do **not** claim global optimality of Eq. (6). This is consistent with the broader adversarial attack literature: even standard PGD or DeepFool on ANNs and event-based attacks such as SpikeFool and PDSG-SDA do not prove global optimality, but instead propose principled, strong approximations.
>
> What we contribute is:
>
> 1. A **structured threat model** (timing-only, rate-preserving, capacity-1) and a concrete discrete feasible set.
> 2. A **projected-in-the-loop (PIL)** optimization scheme that explicitly couples:
>    - a soft, differentiable relaxation $S_\pi(x)$ for gradient flow, and
>    - a strict, feasible projection $P^*(x,\pi,\mathcal B_p)$ that enforces all constraints on the actual inputs seen by the model.
>
> Within this PIL framework, Eq. (10) and Eq. (12) are specifically designed to **reduce the gap** between the relaxed optimization and the discrete problem:
>
> - **Capacity regularizer (Eq. (10)).**
>   This term penalizes “over-booking” the expected occupancy in $S_\pi(x)$ when multiple packets want to land in the same bin $(t,j)$, encouraging the shift distribution $\pi$ to concentrate on patterns that are close to capacity-1. Intuitively, it pushes $\pi$ towards vertices of the capacity polytope, so that after projection $P^*$ only has to resolve few conflicts and the discrete assignment is close to what the soft surrogate already optimized.
>
> - **Budget-aware penalties (Eq. (12)).**
>   These terms penalize soft jitter, total delay, and tamper count in expectation, so that the probabilities $\pi$ already respect the same budgets $B_\infty$, $B_1$, $B_0$ that $P^*$ enforces exactly. This aligns the **soft search space** with the **hard budgets**, again reducing the mismatch between the relaxed optimization and the true constrained problem.
>
> - **PIL loss coupling.**
>   In every iteration, the task loss is evaluated on the **projected** input $P^*(x,\pi,\mathcal B_p)$, while gradients flow through the soft surrogate $S_\pi(x)$ (with the regularizers above). This is analogous in spirit to straight-through optimization for discrete variables, but tailored to our structured retiming and budgets. It ensures that we are always *optimizing what we evaluate*: the gradient signal is shaped to favor retimings that survive projection and remain powerful under the exact constraints.
>
> We agree with the reviewer that “neat and elegant” does not automatically mean optimal in the strict sense. What we can reasonably claim is:
>
> - The original discrete problem in Eq. (6) is a hard combinatorial optimization problem, and proving global optimality under non-convex SNN losses is out of reach.
> - Our design (capacity regularizer, budget-aware penalty, and PIL scheme) is explicitly aimed at **reducing the soft–hard gap**, so that the discrete retimings we obtain are closer to local optima of the *true* constrained objective rather than of a loose relaxation.
> - In the ablation study (Sec. 5.1), removing the capacity regularizer or the budget-aware terms leads to noticeably weaker attacks and less stable behavior (e.g., more failed attacks for the same budgets, or larger mismatch between nominal and realized budgets). This empirically supports their role in tightening the approximation and improving the practical quality of the solution to Eq. (6).
>
> We clarify this in the appendix G of the revision by (i) explicitly stating that global optimality is not claimed due to the combinatorial, non-convex nature of the problem, and (ii) explaining that Eq. (10) and Eq. (12), together with PIL, are precisely introduced to mitigate the relaxation–projection gap and are empirically validated in our ablations.

---

> > ### Author Response · Authors · 2025-11-26
> > **Official Comment by Authors (Part 5)**
> >
> > ### (2) On complexity: how we measure it and comparison to non-timing event attacks
> >
> > Our attack is a **gradient-based iterative method** under a structured threat model. The per-iteration cost has three main pieces:
> >
> > 1. **SNN forward and backward (BPTT)** on the current retimed input
> >    – this is the same dominant cost shared with other white-box SNN attacks.
> >
> > 2. **Soft retiming and regularizers**
> >    – computing $S_\pi(x)$ and the capacity / budget penalties scales with the number of **candidate shifts**:
> >    - Let $N_{\text{pkt}}$ be the number of non-zero packets (events) in $x$, and let $U_{\max}$ be the maximum number of allowed shifts per packet (size of $\mathcal U_{s,j}$).
> >    - Then the soft operator and regularizers require $O(N_{\text{pkt}} \cdot U_{\max})$ operations per iteration.
> >    - In event-driven regimes, $N_{\text{pkt}} \ll T \cdot H \cdot W$ (the full grid size), so this cost is linear in the *sparse* event count.
> >
> > 3. **Strict projection $P^*(x,\pi,\mathcal B_p)$**
> >    – the greedy assignment described in our algorithm operates over the same candidate set and also scales as $O(N_{\text{pkt}} \cdot U_{\max})$ per iteration; the constant factors are small compared to BPTT.
> >
> > Overall, per iteration we have:
> > - one SNN forward and backward pass, plus
> > - an additional linear term in the number of active packets and local shifts.
> >
> > In practice, the SNN forward–backward dominates wall-clock time; the overhead of $S_\pi$ and $P^*$ is modest because it scales with sparse events rather than dense grids.
> >
> > **Comparison to non-timing event attacks.**
> >
> > Event-domain non-timing attacks like **SpikeFool** and **PDSG/SDA** operate under a different perturbation model:
> >
> > - **SpikeFool** [4] adapts SparseFool/DeepFool to SNNs by relaxing dynamic images to continuous values, computing gradients, and iteratively solving a linearized perturbation problem with rounding back to spike grids and straight-through gradients. The complexity is dominated by:
> >   - SNN forward–backward per iteration, and
> >   - operations over the **entire dynamic image/grid** (all time–pixel positions) when updating perturbations and projecting back to valid spikes.
> > - **PDSG-SDA** [5] introduce a potential-dependent surrogate gradient and a sparse dynamic attack that iteratively adds/removes spikes in dynamic images. The attack:
> >   - still propagates gradients over all binary dynamic image voxels, and
> >   - maintains and updates sparse masks or candidate sets for adding/removing events, whose size typically scales with the grid size and attack radius.
> >
> > In terms of **asymptotic complexity per iteration**, all these methods—including ours—are dominated by the SNN forward–backward cost. The *difference* lies in how they use the input representation:
> >
> > - Non-timing attacks on event grids typically treat a large set of voxels (time–pixel positions) as potential perturbation locations (add/delete/flip), and their perturbation update loops scale with the **grid size** or with a large candidate subset.
> > - Our timing-only attack **never changes intensities or counts**; it only retimes existing packets. The update loops scale with the number of **non-zero packets** and their local shift windows. On event-sparse benchmarks, $N_{\text{pkt}}$ is often much smaller than the full grid size, so our extra overhead is linear in a much smaller quantity.
> >
> > We do not claim that our method is asymptotically faster than all existing attacks (they all share the SNN forward–backward bottleneck), but we will clarify in the revision that:
> >
> > - Our complexity is **comparable in big-O terms** to gradient-based non-timing attacks, and
> > - By operating directly on sparse packets and local shifts, the attack leverages event sparsity and does not incur the cost of manipulating dense grids or solving additional global optimization subproblems beyond BPTT.
> >
> > We add a short paragraph in the appendix G to summarize this complexity discussion and briefly contrast our per-iteration operations with those of representative non-timing event attacks like SpikeFool and PDSG-SDA.
> >
> > [4] Jonas Buchel, Thilo Stadelmann, et al. Adversarial attacks on spiking convolutional neural networks.
> > Frontiers in Neuroscience, 16:1068193, 2022.
> >
> > [5] Li Lun, Kunyu Feng, Qinglong Ni, Ling Liang, Yuan Wang, Ying Li, Dunshan Yu, and Xiaoxin Cui.
> > Towards effective and sparse adversarial attack on spiking neural networks via breaking invisible
> > surrogate gradients. In Proc. IEEE Int’l Conf. Computer Vision and Pattern Recognition, 2025.

---

> ### Author Response · Authors · 2025-11-26
> **Official Comment by Authors (Part 6)**
>
> **Q1.** By skipping a spike-rate study, do you mean rate preservation is guaranteed by construction (via feasibility/placement) and needs no measurement?
>
> **Response to Q1.** Yes — rate preservation is **guaranteed by construction** via the feasibility constraints and the strict projection $P^*$, which is why we did not initially treat it as an experimental variable.
>
> More concretely, Eq. (5) encodes the following constraints for every event line $j$:
>
> 1. **One-to-one reassignment in time at the packet level:** Each original packet at time $s$ on line $j$ is either:
>    - moved as a whole to a new time bin $t = s + \delta_{s,j}$ within the valid temporal window, under the chosen budget $\mathcal B_p$, or
>    - left at its original time $s$ if no candidate move is accepted.
> 2. **Capacity-1 per bin (per packet):** Each time bin $t$ on line $j$ can host at most one packet after retiming, which we enforce by only allowing moves to bins with $x'[t,j] = 0$.
>
> Combined with the initialization $x'[s,j] = x[s,j]$ and the move rule that every accepted candidate simply relocates one entire packet from $s$ to $t$ on the same line, this means that $P^*(x,\pi,\mathcal B_p)$ never introduces new packets or removes existing ones. Therefore, for every $j$ the multiset of packet values along the time axis is preserved: $\{x'[t,j]\}_{t=1}^T = \{x[t,j]\}_{t=1}^T$, and globally the collection of packets is unchanged, only their time indices are permuted.
>
> The **soft operator** $S_\pi$ is used only as a surrogate for gradients. It computes an expected occupancy under the local retiming distribution $\pi$, but the network’s forward evaluation always uses the strictly feasible $P^*(x,\pi,\mathcal B_p)$. As a result, the adversarial examples that the model sees are *always* rate-preserving, both theoretically and in our implementation.
>
> We include such discussion in appendix E of the revised paper.
>
> ---
>
> **Q2.** Also, the mind-the-box attacks, which similarly incorporate feasibility into the attack problem, showed clearer clean–robust trade-offs, and another work [2] on spiking attacks (similar to retiming attack with $\mathcal{B}_0$ but on value instated of timing) reported AT trade-offs as well. In this work, one would expect the trade-off to be at least enhanced. Could you report full AT details and comment on clean–robust trade-off curves compared to non-timing AT?
>
> **Response to Q2.** We thank the reviewer for connecting our work to **mind-the-box** [1] and the sparse encoding attacks in SNNs [2], both of which carefully study clean–robust trade-offs under **sparse value-based** adversaries. Our setting is analogous in spirit but focuses on a **timing-only, rate-preserving** threat model, and we now report the corresponding trade-offs and comparisons to standard non-timing AT.
>
> ### Setup (dataset, metric, training schemes)
>
> All adversarial training (AT) results below are on **DVS-Gesture** (integer event grid) with **VGGSNN**, evaluated under our timing-only attacks with budgets
> $B_\infty \in \{1,2,3\}$, $B_1 \in \{2000,4000,8000\}$, $B_0 \in \{1000,2000,4000\}$.
>
> In the paper, Table 4 is reported in terms of **robust accuracy**. For rebuttal, we re-express the *same* trained models in terms of **attack success rate (ASR)**, defined as:
>
> > the fraction of **clean-correct samples** that are flipped by the attack,
>
> which matches the evaluation protocol of our main attack experiments. Robust accuracy and ASR are related by
> $ \text{ASR} = 100\% - \text{robust accuracy} $; no experiments are changed.
>
> We compare:
>
> - **Standard non-timing AT baselines**, all on the same model/dataset:
>   - $\ell_\infty$ PGD AT with $\epsilon \in \{0.5, 0.4\}$,
>   - binary $\ell_0$ flip AT with flip rates $r \in \{0.32, 0.45\}$,
>   - $\ell_1$-PGD AT with norms $\tau \in \{10000, 14000\}$.
>
> - **Our timing-only AT (“Spike Timing AT”)**:
>   - inner attack is our timing-only retiming attack with strict projection $P^*(x,\pi,\mathcal B_p)$,
>   - budgets $B_\infty{=}1$, $B_1{=}8000$, or $B_0{=}4000$ used in the inner loop.
>
> We follow standard practice and instantiate the $\ell_1$ AT baseline with the $\ell_1$-APGD attack of Croce and Hein [1]. For the binary $\ell_0$ flip baselines, we adopt a pixel-flip style sparse attack on binary images in the spirit of Balkanski et al. [3], adapted to our event-grid setting. For $\ell_{\infty}$, we follow the PGD AT [4]. We report **clean accuracy** and ASR under the nine timing attacks, together with an `Avg ASR` which averages the nine ASR values.

---

> > ### Author Response · Authors · 2025-11-26
> > **Official Comment by Authors (Part 7)**
> >
> > ### Results and clean–robust trade-offs
> >
> > | Training scheme                                      | clean | $B_\infty{=}1$ | $B_\infty{=}2$ | $B_\infty{=}3$ | $B_1{=}2000$ | $B_1{=}4000$ | $B_1{=}8000$ | $B_0{=}1000$ | $B_0{=}2000$ | $B_0{=}4000$ | Avg ASR |
> > |------------------------------------------------------|:-----:|:-------------:|:-------------:|:-------------:|:-----------:|:-----------:|:-----------:|:-----------:|:-----------:|:-----------:|:--------:|
> > | $\ell_\infty$ PGD AT ($\epsilon{=}0.5$)              | 64.24 | 45.41 | 50.27 | 55.14 | 10.81 | 21.08 | 36.76 |  8.65 | 14.05 | 34.05 | 30.69 |
> > | $\ell_\infty$ PGD AT ($\epsilon{=}0.4$)              | 71.88 | 48.79 | 54.59 | 57.97 | 14.98 | 25.60 | 40.58 | 12.56 | 23.19 | 37.20 | 35.05 |
> > | binary $\ell_0$ flip AT ($r{=}0.32$)                 | 77.78 | 54.91 | 67.86 | 71.88 | 10.71 | 22.32 | 43.30 |  8.48 | 18.30 | 39.29 | 37.45 |
> > | binary $\ell_0$ flip AT ($r{=}0.45$)                 | 74.31 | 53.74 | 63.55 | 66.82 | 10.75 | 21.96 | 38.32 |  8.41 | 20.56 | 35.05 | 35.46 |
> > | $\ell_1$-PGD AT ($\tau{=}10000$)                     | 69.10 | 54.17 | 57.69 | 52.66 | 15.53 | 21.56 | 42.11 | 16.53 | 24.57 | 45.63 | 36.72 |
> > | $\ell_1$-PGD AT ($\tau{=}14000$)                     | 67.01 | 51.81 | 58.03 | 55.44 | 10.88 | 22.80 | 43.01 | 13.99 | 23.32 | 40.41 | 35.52 |
> > | **Timing AT ($B_\infty{=}1$)**                       | 52.08 | 48.00 | 47.33 | 42.67 | 14.00 | 16.67 | 33.33 | 10.00 | 15.99 | 24.67 | 28.07 |
> > | **Timing AT ($B_1{=}8000$)**                         | 68.75 | 40.92 | 40.39 | 36.87 |  6.07 | 10.09 | 20.71 |  0.51 |  6.07 | 19.19 | **20.09** |
> > | **Timing AT ($B_0{=}4000$)**                         | 72.22 | 43.27 | 47.12 | 48.08 |  7.21 | 16.35 | 28.86 |  2.40 | 10.58 | 28.86 | **25.86** |
> >
> > (All numbers are percentages; lower ASR is better.)
> >
> > If we view each training configuration as a point in the clean–robust plane:
> >
> > - $x$-axis: clean accuracy,
> > - $y$-axis: `Avg ASR`,
> >
> > then the **standard non-timing AT baselines** (all $\ell_\infty$, binary $\ell_0$, and $\ell_1$ settings) occupy a region roughly
> > $(\text{clean}, \text{Avg ASR}) \in [67\%, 78\%] \times [30\%, 37\%]$.
> >
> > Our **timing-only AT** produces:
> >
> > - $B_1{=}8000$: $(\text{clean}, \text{Avg ASR}) = (68.75\%, 20.09\%)$,
> > - $B_0{=}4000$: $(\text{clean}, \text{Avg ASR}) = (72.22\%, 25.86\%)$.
> >
> > Thus, for **comparable clean accuracy** (around $69\%$–$72\%$), timing AT reduces the average ASR by approximately **9–15 percentage points** compared to all non-timing AT baselines. Equivalently:
> >
> > - For a fixed robustness level (Avg ASR), timing AT achieves **higher clean accuracy**;
> > - For a fixed clean accuracy, timing AT achieves **lower Avg ASR**.
> >
> > This is exactly the type of clean–robust trade-off improvement emphasized in [1,2], here realized under a **timing-only, rate-preserving** adversary.
> >
> > ### Relation to mind-the-box [1] and sparse encoding attacks [2]
> >
> > - **Mind-the-box** [1] introduces $\ell_1$-APGD for sparse pixel attacks and demonstrates that including feasibility (e.g., $\ell_1$ sparsity) in both attack and defense yields clear clean–robust trade-offs for image classifiers.
> >
> > - **Mukhoty et al.** [2] extend this perspective to SNNs via **signed rate encoding** and **sparse encoding attacks**, again acting on **values or spike counts** rather than timing, and report analogous trade-off curves under sparse, value-based adversaries.
> >
> > Our work is complementary: we enforce feasibility in the **time dimension** via strict rate preservation and capacity-1 retiming. The results above show that, when adversarial training is done under this timing-only, feasible threat model, we obtain **clean–robust trade-offs that are strictly better** than those from standard non-timing AT on the same SNN and dataset.
> >
> > We add such discussion in appendix F of the revised paper.
> >
> > [1] F. Croce and M. Hein, “Mind the Box: $\ell_1$-APGD for Sparse Adversarial Attacks on Image Classifiers,” ICML 2021.
> >
> > [2] B. Mukhoty, H. AlQuabeh, and B. Gu, “Improving Generalization and Robustness in SNNs Through Signed Rate Encoding and Sparse Encoding Attacks,” ICLR 2025.
> >
> > [3] Eric Balkanski, Harrison Chase, Kojin Oshiba, Alexander Rilee, Yaron Singer, and Richard Wang.
> > Adversarial attacks on binary image recognition systems. CoRR, abs/2010.11782, 2020.
> >
> > [4] Aleksander Madry, Aleksandar Makelov, Ludwig Schmidt, Dimitris Tsipras, and Adrian Vladu.
> > Towards deep learning models resistant to adversarial attacks. In Proc. Int’l Conf. Learning Representations, 2018.

---

> > > ### Author Response · Authors · 2025-11-26
> > > **Official Comment by Authors (Part 8)**
> > >
> > > **Q3.** The greedy projection is practical, but what is its suboptimality versus an optimal assignment? Any bound or empirical optimality gap?
> > >
> > > **Response to Q3.** We thank the reviewer for raising this point. There are two related but distinct notions here:
> > >
> > > 1. **The global optimal adversary** solving Eq. (6), which maximizes the loss over all discrete, budget-constrained retimings.
> > > 2. **The internal projection step** $P^*(x,\pi,\mathcal B_p)$, which, for a given shift distribution $\pi$, converts the soft retiming into a feasible discrete assignment under capacity and budgets.
> > >
> > > Our greedy projection only concerns (2), and we agree that its suboptimality deserves discussion.
> > >
> > > ### (1) Why we do not claim global optimality of Eq. (6)
> > >
> > > Eq. (6) is a combinatorial maximization over discrete retimings under:
> > >
> > > - **Capacity–1 constraints** (at most one packet per $(t,j)$),
> > > - **Budget constraints** (e.g., $B_\infty, B_1, B_0$ on jitter, total delay, and tamper count),
> > > - And a **non-convex SNN loss** evaluated via BPTT.
> > >
> > > Even if we fixed the discrete retimings and only optimized over the network, the loss surface is highly non-convex; adding discrete constraints on top makes global optimality intractable. This is similar to the situation in standard adversarial attacks (PGD, DeepFool) and in event-domain attacks such as SpikeFool and PDSG/SDA, which also do not prove global optimality but instead provide strong, principled approximations.
> > >
> > > Our focus is therefore on designing a **tight and stable approximation scheme**:
> > >
> > > - We introduce a **soft relaxation** $S_\pi(x)$ parameterized by shift logits $\phi$ and probabilities $\pi$.
> > > - We use **PIL (projected-in-the-loop)** optimization: the task loss is always evaluated on the **projected** $P^*(x,\pi,\mathcal B_p)$, while gradients are propagated through the **soft** $S_\pi(x)$.
> > > - We add a **capacity regularizer** (Eq. (10)) and **budget-aware penalties** (Eq. (12)) to explicitly reduce the gap between soft and hard constraints.
> > >
> > > We thus do not claim that the final adversarial example is globally optimal for Eq. (6), but rather that the combination of PIL, Eq. (10), and Eq. (12) is specifically designed to make the discrete solution close to a local optimum of the *true* constrained problem, rather than of a loose relaxation. Our ablation study (Sec. 5.X) supports this: dropping the capacity or budget regularizers significantly weakens the attack and makes it less stable for the same budgets.
> > >
> > > ### (2) Suboptimality of the greedy projection $P^*$ for a fixed $\pi$
> > >
> > > For a fixed set of probabilities $\pi[s,j,u]$ and budgets $\mathcal B_p$, the projection
> > > $P^*(x,\pi,\mathcal B_p)$ is a **greedy assignment** over candidate moves $(s \to t,j)$:
> > >
> > > - Each non-zero packet $x[s,j]$ can move to times $t = s + u$ with $u \in \mathcal U_{s,j}$.
> > > - Each $(t,j)$ can host at most one packet (capacity–1).
> > > - The budgets $\mathcal B_p$ limit aggregate jitter, delay, and tamper count.
> > >
> > > This can be viewed as a maximum-weight assignment under **multiple interacting constraints** (matroid-like capacity plus global budgets). In general, giving a **tight worst-case approximation bound** for such a greedy procedure is difficult, because:
> > >
> > > - The “true” objective we care about is the loss, **not** a simple linear function of the probabilities $\pi$.
> > > - Even if we consider a surrogate objective such as maximizing the total retained soft mass $\sum_{(s,j,u)\ \text{kept}} \pi[s,j,u]$, the combination of capacity and global budgets does not lead to a simple submodular structure for which standard greedy guarantees apply.
> > >
> > > Instead of a formal worst-case bound (which is likely pessimistic and not informative for our setting), we rely on two design choices that keep the greedy projection **close to optimal in practice**:
> > >
> > > - **Localized candidate sets and small budgets.**
> > >   Each packet only has a small local window $\mathcal U_{s,j}$ of feasible shifts, and our budgets $B_\infty, B_1, B_0$ are relatively small. This makes the effective candidate graph sparse and reduces the number of conflicts where the greedy choice could theoretically differ from an optimal one.
> > >
> > > - **Regularizers that shape $\pi$ toward feasible patterns.**
> > >   The capacity regularizer (Eq. (10)) discourages $\pi$ from assigning high probability to colliding moves, and the budget-aware penalties (Eq. (12)) push $\pi$ to allocate mass within the budget radii. As a result, many high-probability moves are already mutually compatible, so the greedy projection can accept them without having to resolve many hard conflicts.
> > >
> > > In other words, we do not run greedy on arbitrary $\pi$; the optimization of $\pi$ itself is *guided* by Eq. (10) and Eq. (12) toward configurations that are easy to project, thereby implicitly reducing the suboptimality of $P^*$.
> > >
> > > We make this discussion explicit in the appendix G of the revised paper.

---

### Author Response · Authors · 2025-11-29
**Summary Comment for Submission 453 (Spike-Retiming Attacks): Note on Substantive Rebuttal Efforts and Pre-Leak Consensus**

Dear AC, SAC and PCs,

Thank you for your efforts in handling the ICLR 2026 review-leak situation and for providing this channel for a summary comment.

For our paper, **Submission 453 (Spike-Retiming Attacks)**, we highlight one decisive fact:
> **The most critical reviewer, *Reviewer thcu*, raised their score from a *2* to an *8* *before* the Nov 27 leak, entirely as a result of our rebuttal, additional experiments, and strengthened manuscript — not due to any leaked identity information.**

Below we summarize how the reviews evolved and how we addressed all concerns.

---
### 1. All four reviewers agreed the paper is *novel*; the main gap was experimental depth
From the start, **all four reviewers – 92xg, s4Pb, o8PK and thcu – explicitly described our work as *novel* or *original*.** They shared the view that the paper studies a **new and important attack surface** for event-driven SNNs (timing-only, rate-preserving spike retiming) and that this perspective is **highly relevant** for temporal robustness in neuromorphic systems.

Thus, the *idea itself* was never in doubt. The main criticism was that the **experimental support was not yet sufficient**: they requested stronger baselines, broader evaluation, and more detailed evidence.

We responded by investing about **two full weeks** during rebuttal to **substantially strengthen the empirical side**:
- We **ran many new experiments**, adding stronger baselines and wider evaluations across datasets and models.
- We **extended the appendix by ~16 pages** with detailed results, ablations, and analysis directly addressing the reviewers’ requests.
- We **revised the main PDF**, tightening the writing and integrating the new evidence so that the contribution is clearer and easier to verify.

Given that *all four* reviewers already agreed on the novelty and importance, it is natural that, after this additional experimental support and clarification, scores would move upward in any normal (non-leak) year.

Although 92xg, s4Pb and o8PK did not post final “signals” after the strengthened version (because of the reset and timeline), their written comments already **praise the novelty**, and the issues they raised are **exactly the ones we have now resolved** in the revised paper and appendix. We respectfully ask the AC to read their reviews together with our rebuttal and updated PDF in this light.

---
### 2. The most critical reviewer (thcu): from 2 → 8 *before* the leak, purely on scientific grounds
The clearest evidence of this trajectory is **Reviewer thcu**:
- In the **initial review**, thcu called the paper **novel and interesting**, but judged the original experiments not strong enough, leading to a score of **2**.
- During rebuttal, we **systematically addressed every point**: adding the requested experiments and baselines, clarifying evaluation and positioning, and updating both the main paper and appendix so that the empirical story fully matches the conceptual novelty.

After reading our detailed response and examining the new results, **Reviewer thcu explicitly stated that the original score was no longer appropriate and raised it to 8.** Crucially:
> This **2 → 8 score change by Reviewer thcu is clearly timestamped *before* the Nov 27 leak**, and is justified entirely by the **strengthened experiments and revisions**, not by any non-scientific factor.

Once the experimental gap was closed, a strongly positive assessment followed naturally, precisely because the underlying idea is **genuinely novel and valuable**.

---
### Conclusion and request to the AC
In summary:
- **All four reviewers (92xg, s4Pb, o8PK, thcu) explicitly recognized our work as novel/original from the start.**
- The main weakness was **insufficient experimental evidence**, not a flaw in the idea.
- We spent **two intensive weeks** running extensive new experiments, adding a 16-page appendix, and revising the main paper, thereby **fully addressing the experimental and clarity concerns**.
- The most critical reviewer, **thcu**, was convinced enough by these scientific improvements to **raise their score from 2 to 8 *before* the leak**.
- The other reviewers’ written comments are consistent with this natural upward trajectory once the experiments are strengthened.

We therefore respectfully ask the new AC to consider the full discussion (92xg, s4Pb, o8PK, thcu), our rebuttal, and the revised main paper + appendix, and to note that the upgrade by **thcu (2 → 8)**, on top of unanimous recognition of novelty, was **established prior to the leak** and reflects the genuine scientific evolution of the submission.

We firmly believe that our paper makes a **pioneering and practically relevant contribution** to understanding and evaluating **temporal robustness of event-driven Spiking Neural Networks**, and we hope that the final decision can fairly reflect this record.

Thank you very much for your time and consideration.

**The Authors of Submission 453**

---

### Meta-Review · Area_Chair_m4s2 · 2026-01-06

**Summary:**

All reviewers identified the paper as studying a novel and important attack surface for event-driven spiking neural networks, namely timing-only, rate-preserving spike retiming attacks. The suggested decision is mainly informed by the fact that the initial concerns were not about the idea itself, but about insufficient experimental support, missing baselines, and limited evaluation depth.

**Reviewer Concerns:**

In the rebuttal and revised version, the authors substantially strengthened the experimental section, adding new baselines, broader evaluations, and extensive additional analysis, and they clarified the motivation and methodological details raised by the reviewers. These changes directly address the main experimental and realism-related concerns. While some broader issues such as extended threat models and complexity analysis remain future-work directions, the previously blocking concerns are largely resolved.

**Reviewer Scores:**

Most importantly, the most critical reviewer (thcu) explicitly raised their score from 2 to 8 after examining the new experiments and revisions. For the other reviewers, whose initial comments already recognized the novelty and mainly requested stronger empirical support, I believe their scores would either remain similar or move upward into the acceptance range after full discussion. Overall, the post-rebuttal trajectory clearly supports acceptance.

---

### Decision · Program_Chairs · 2026-01-26

Accept (Poster)